# Incentivizing Truthfulness and Collaborative Fairness in Bayesian Learning

Rachael Hwee Ling Sim [* 1]   Jue Fan [* 1 2]   Xiao Tian [1 2]   Xinyi Xu   Patrick Jaillet [3]   Bryan Kian Hsiang Low [1]

## Abstract

Collaborative machine learning involves training high-quality models using datasets from a number of sources. To incentivize sources to share data, existing data valuation methods fairly reward each source based on its data submitted as is. However, as these methods do not verify nor incentivize data truthfulness, the sources can manipulate their data (e.g., by submitting duplicated or noisy data) to artificially increase their valuations and rewards or prevent others from benefiting. This paper presents the first mechanism that provably ensures (**F**) collaborative fairness and incentivizes (**T**) truthfulness at equilibrium for Bayesian models. Our mechanism combines semivalues (e.g., Shapley value), which ensure fairness, and a truthful data valuation function (DVF) based on a validation set that is unknown to the sources. As semivalues are influenced by others' data, we introduce an additional condition to prove that a source can maximize its expected data values in coalitions and semivalues by submitting a dataset that captures its true knowledge. Additionally, we discuss the implications and suitable relaxations of (**F**) and (**T**) when the mediator has a limited budget for rewards or lacks a validation set. Our theoretical findings are validated on synthetic and real-world datasets.

## 1. Introduction

In recent years, there has been more interest in applying collaborative *machine learning* (ML)[1] to build high-quality ML models (Yang et al., 2019). Although an individual data source may have limited data, multiple sources when combined would have larger and more diverse data. For example, in healthcare, a local hospital with limited data diversity and quantity can draw on data from other hospitals to improve disease prediction (Sheller et al., 2020; Panagopoulos et al., 2022). As another example, financial institutions can share data to build a more accurate fraud classification model (Brundyn et al., 2022). Existing works (Sim et al., 2020; Nguyen et al., 2022; Sim et al., 2023; Wu et al., 2024) have recognized that since data sources incur significant costs for data collection, they require incentives for collaboration. So, these works suggest how to provide incentives, such as (**F**) *collaborative fairness* which ensures that a source receives a higher monetary reward (or a more valuable model as a reward) by contributing more valuable data and enforces that a source's reward should depend on others' contribution (e.g., lower if its data is redundant due to similarity to others) (Sim et al., 2022).

The above existing works share a common limitation: They value the data submitted as is without incentivizing or verifying (**T**) *truthfulness* of the data. This limitation is problematic as untruthfulness can harm the performance of the jointly trained ML model, reduce the overall benefits of collaboration, and disincentivize truthful data sources who are unwilling to benefit harmful untruthful sources. In Sec. 3 and App. C.1, we describe how untruthful data sources can submit duplicated data to artificially increase their valuations and rewards or introduce noise to their data to prevent others from benefiting. This raises the key question: **[I] How can we define the *data valuation function* (DVF, which maps each dataset to a value) to prefer data that contributes to better model performance while incentivizing *truthfulness*?** Intuitively, the untruthful sources can artificially increase their data values/rewards if they had complete information about the DVF. For instance, knowing that only data size/cardinality matters, they can duplicate data to increase their rewards. Similarly, knowing the exact validation set, they can discard the irrelevant part of their datasets to better align with the validation set and increase their rewards (Zheng et al., 2024). To counter such strategies and incentivize (**T**) truthfulness, the DVF must be based on some information that remains unknown to all data sources and assumptions are needed.

---

[*]Equal contribution [1]Department of Computer Science, National University of Singapore, Singapore [2]Agency for Science, Technology and Research (A*STAR), Singapore [3]Department of Electrical Engineering and Computer Science, Massachusetts Institute of Technology, USA. Correspondence to: Bryan Kian Hsiang Low <lowkh@comp.nus.edu.sg>.

*Proceedings of the 43rd International Conference on Machine Learning*, Seoul, South Korea. PMLR 306, 2026. Copyright 2026 by the author(s).

[1]App. B.1 describes how our collaborative ML setting differs from the federated learning and crowd-sourcing settings.

*Table 1.* Comparison of related works on (**F**) collaborative fairness, (**T**) truthfulness, (**O**) considering other incentives or constraints, and (**E**) multi-source experiments. Only **ours** *simultaneously* ensures fairness (Sec. 2), provably guarantees truthfulness (Sec. 3), *and* is empirically verified with multiple sources (Sec. 6). The × marks DVFs that do not satisfy the fairness conditions (Sec. 2) or are proven to disincentivize truthfulness (App. C.1). Blank cells indicate no theoretical guarantees of (**T**) or no consideration of (**O/E**). However, some works may still demonstrate truthfulness empirically. More differences are highlighted in Secs. 1 and 7.

| | (F) | (T) | (O) | (E) |
|---|---|---|---|---|
| **Ours** | ✓ | ✓ | ✓ | ✓ |
| Zheng et al. (2024) | × | ✓ | | |
| Sim et al. (2020; 2023) | ✓ | × | ✓ | ✓ |
| Chen et al. (2020) | × | ✓ | ✓ | |
| Xu et al. (2021b) | ✓ | × | | ✓ |
| Ghorbani & Zou (2019); Jia et al. (2019); Xu et al. (2021a); Wu et al. (2024) | ✓ | | | ✓ |

On the other hand, existing approaches that incentivize truthful data submission may not ensure (**F**) *collaborative fairness*. The approaches of Chen et al. (2020); Zheng et al. (2024) value a source $i$ based on the change in log-likelihood of others' data and of a validation set after observing source $i$'s dataset, respectively. (**T**) Truthful data submission leads to the highest *expected* data value compared to untruthful submissions. However, with only two data sources, peer prediction approaches (e.g., Chen et al. (2020)) may unfairly assign equal values to a smaller, less informative dataset and a more informative one. The approach of Zheng et al. (2024) may unfairly value a redundant source (with data similar to several others) equally with a unique data source that contributes a greater improvement in the log-likelihood after observing *all* sources' datasets (App. B.2). Since these approaches do not enforce that a source's reward depends on others' contribution to achieve collaborative fairness, we pose the second question: **[II] Can *post-processing* additionally ensure *collaborative fairness* while still preserving the truthfulness incentive?** We show that it is possible with an additional assumption to address the influence of others' data (due to ensuring collaborative fairness) on truthfulness.

This paper seeks to fill the gap and achieve both (**T**) *truthfulness* and (**F**) *collaborative fairness* simultaneously (Fig. 4). We address **[I]** and ensure (**T**) by defining our DVF based on the log-likelihood of the validation set for *Bayesian* models. Next, to achieve (**F**), we decide the reward of each source based on the *semivalues* (Dubey et al., 1981) of the cooperative game with all sources and the DVF as the characteristic function. We address **[II]** by proving the existence of a truthful equilibrium where every source maximizes its semivalue/reward by truthful data submission (on the condition that other sources are truthful). Lastly, we address the following question: **[III] is it possible to incentivize (T) and (F) while simultaneously considering other in-**

**centives/constraints, e.g., when removing the need for a validation set?** Our work generalizes that of Zheng et al. (2024) by addressing **[II]** and **[III]**. It also goes beyond existing data valuation methods based on validation accuracy (Ghorbani & Zou, 2019; Jia et al., 2019) and collaborative ML works that ensure (**F**) (Sim et al., 2020) by specifying reasonable assumptions required to theoretically guarantee (**T**). Our contributions are given in Tab. 1 and include:

- Identifying that unknown information is necessary to prevent untruthfulness, a DVF and reasonable assumptions to theoretically guarantee (**T**) truthfulness (Sec. 3);
- Proposing the first mechanism that provably ensures both (**T**) truthfulness and (**F**) collaborative fairness at equilibrium (Sec. 4);
- Identifying the implications and suitable relaxations of (**T**) and (**F**) when the mediator lacks a validation set (Sec. 5); and
- Validating our theoretical findings regarding questions **[I-III]** on synthetic and real-world datasets with multiple sources (Sec. 6).

## 2. Collaborative ML

Our problem setting considers a set $N$ of $n$ data sources collaborating to train an ML model for the same task (e.g., predicting heart disease in the general population). Each source $i \in N$ owns a *true dataset* $\bar{D}_i$ but may choose to share the dataset $D_i$ with the mediator during the collaboration. We define the true dataset $\bar{D}_i$ as the dataset that fully captures the knowledge of source $i$ beyond that of the mediator. For example, instead of the raw collected data, $\bar{D}_i$ may have outliers removed/corrected and include data augmentations (e.g., image translations with the same label when training an image classification model). For each coalition/subset of sources $C \subseteq N$, let $D_C \triangleq \bigcup_{i \in C} D_i$ denote their aggregated dataset. Let $\mathfrak{D} \triangleq (D_i)_{i \in N}$ denote all submitted datasets. As in prior works (Sim et al., 2020; Chen et al., 2020), we consider the setting where a trusted mediator evaluates the value of the data of a source $i$, source $j$, or a coalition $C$ using a common *truthful* data valuation function $v$ (Sec. 3) via $v(D_i)$, $v(D_j)$, or $v(D_C)$, respectively. Based on the coalition values $(v(D_C))_{C \subseteq N}$, the mediator decides the *fair* reward $r_i$ for each source $i \in N$ which still incentivizes truthful submission of $\bar{D}_i$ (Fig. 4).

Next, we provide working definitions of (**F**) collaborative fairness and (**T**) truthfulness, which we will refine in later sections. To satisfy (**F**), *the data valuation function $v$ should assign a higher value to a dataset $D$ that leads to ML models with better performance (e.g., accuracy) on a validation set.* In some existing works, under the assumption of truthful sources, the valuation can also be based on quantities such as diversity of the dataset (Xu et al., 2021b) and information gain on the model parameters (Sim et al., 2020) as these

quantities have been shown to correlate with better model performance. Let $\mathcal{A}$ be the learning algorithm that takes in a dataset and returns a trained ML model. To satisfy **(T)** for every source $i$, *the data valuation function $v$ should assign a higher value to the true dataset (defined above) over datasets that result in different model parameters.* Formally, for any dataset $D_i$ that results in different parameters from $i$'s true dataset $\bar{D}_i$ (i.e., $\mathcal{A}(D_i) \neq \mathcal{A}(\bar{D}_i)$), the valuation (which depends on $\mathcal{A}$) should not increase: $v(D_i) \leq v(\bar{D}_i)$. We assume that each source can only submit one dataset for valuation and cannot untruthfully increase its reward by submitting multiple datasets as different sources. See App. A for an in-depth justification of this and later assumptions.

**(F)** additionally requires that each source $i$'s reward $r_i$ depends on the value of the data contributed by other sources (Sim et al., 2022). A fair reward is often derived by modeling the collaboration as a cooperative game with $n$ players and a characteristic function $\nu_{\mathfrak{D}}^v$ that maps each coalition $C$ to the data value $v(D_C)$; the subscript $\mathfrak{D}$ captures the dependence on the submitted datasets. Let $\mathfrak{D}' \triangleq (D_i')_{i \in N}$ denote another possible combination of submitted datasets. For a data source $i$, we write $\nu_{\mathfrak{D}}^v \succeq_i \nu_{\mathfrak{D}'}^v$ to mean that $i$ is more valuable in $\mathfrak{D}$ and $i$'s contribution to every coalition is at least as large in $\mathfrak{D}$ as in $\mathfrak{D}'$. Formally, this requires $v(D_{C \cup \{i\}}) - v(D_C) \geq v(D'_{C \cup \{i\}}) - v(D'_C)$ for all $C \subseteq N \setminus \{i\}$. This can occur if $D_i$ in $\mathfrak{D}$ is more informative than $D_i'$ in $\mathfrak{D}'$ or if others' data $D_C'$ in $\mathfrak{D}'$ already contain substantial information about $D_i'$. We define the reward values $(r_i)_{i \in N}$ to be *fair* if they satisfy four conditions:

F0 **Uselessness:** If source $i$ contributes no value to all coalitions, it should get no reward. Formally, if $\nu_{\mathfrak{D}}^v(C \cup \{i\}) = \nu_{\mathfrak{D}}^v(C)$ for all $C \subseteq N \setminus \{i\}$, then $r_i = 0$.

F1 **Symmetry**: If sources $i$ and $j$ contribute equally to all coalitions, they should get equal rewards. Formally, if $\nu_{\mathfrak{D}}^v(C \cup \{i\}) = \nu_{\mathfrak{D}}^v(C \cup \{j\})$ for all $C \subseteq N \setminus \{i, j\}$, then $r_i = r_j$.

F2 **Monotonicity**: If source $i$ provides at least as much value in $\mathfrak{D}$ as in $\mathfrak{D}'$, its reward $r_i$ based on $\mathfrak{D}$ should be at least as high as its reward $r_i'$ based on $\mathfrak{D}'$. Formally, if $\nu_{\mathfrak{D}}^v \succeq_i \nu_{\mathfrak{D}'}^v$, then $r_i \geq r_i'$.

F3 **Strict Monotonicity**: If source $i$ provides more value in $\mathfrak{D}$ than $\mathfrak{D}'$, its reward $r_i$ based on $\mathfrak{D}$ should exceed its reward $r_i'$ based on $\mathfrak{D}'$. Formally, if $\nu_{\mathfrak{D}}^v \succeq_i \nu_{\mathfrak{D}'}^v$ and $\nu_{\mathfrak{D}}^v \not\preceq_i \nu_{\mathfrak{D}'}^v$, then $r_i > r_i'$.

Sim et al. (2020); Ghorbani & Zou (2019) and others use semivalues as reward values to satisfy these fair conditions.

**Definition 2.1.** The **semivalue** (Dubey et al., 1981) $\phi_i[\nu]$ of source $i$ is a weighted combination of its marginal contributions to various coalitions, with non-negative coefficients $(w_c)_{c=0}^{n-1}$ satisfying $\sum_{c=0}^{n-1} w_c \binom{n-1}{c} = 1$, i.e., $\phi_i[\nu] \triangleq \sum_{C \subseteq N \setminus \{i\}} w_{|C|}[\nu(C \cup \{i\}) - \nu(C)]$.

Semivalues always satisfy F0 to F2 and fair semivalues also satisfy F3 with $w_c > 0$ for all $c$ (Domenech et al., 2016). *Fair* semivalues, such as the Shapley value (Shapley, 1953) which sets every $w_c \binom{n-1}{c} = 1/n$, are widely used in data valuation (Ghorbani & Zou, 2019; Kwon & Zou, 2021). In App. B.3, we elaborate on the fairness conditions (and their implications such as strict desirability) and give an illustrative numeric example.

## 3. Data Valuation Functions

In this section, we discuss the *data valuation function* (DVF) $v$, which maps a source's or coalition's data to a value that correlates with model performance. DVFs are either validation-set-based or validation-set-free. Validation-set-based DVFs rely on a held-out validation set to evaluate model performance, while validation-set-free DVFs rely on proxies of performance (e.g., dataset diversity or information gain on the model parameters (Xu et al., 2021b; Sim et al., 2020)) assuming truthful sources. In App. C.1, we describe strategies that an untruthful source $i$ can provably use to obtain a $D_i$ that artificially increases its value under different validation-set-free DVFs. For example, duplicating data can increase the information gain and volume. For validation-set-based DVFs with a known validation set, Zheng et al. (2024) have also described how a source can discard some data to better align with the validation set to increase its data value/reward. To counter such strategies and incentivize truthfulness, we suggest that DVFs must depend on some information $\mathcal{O}$ unknown to all sources, such as a held-out validation set. Since $\mathcal{O}$ is unknown, it should be modeled as a random variable for each source. Following the *expected utility hypothesis* (a widely used concept in economics for analyzing decision-making under uncertainty) and Chen et al. (2020), we define the truthfulness incentive **(T)** based on expected data value:

**Definition 3.1** (*i*-truthful). A function $\tau$, which depends on source $i$'s submitted dataset $D_i$ and the dependent random variables (RV) $\mathcal{O}$ unknown to $i$, incentivizes $i$'s *truthfulness* (is *i*-truthful) if given its true dataset $\bar{D}_i$, any other submitted dataset $D_i$ does not increase $i$'s expected value. That is,

$$\mathbb{E}_{\mathcal{O} | \bar{D}_i} \left[ \tau(D_i; \mathcal{O}) \right] \leq \mathbb{E}_{\mathcal{O} | \bar{D}_i} \left[ \tau(\bar{D}_i; \mathcal{O}) \right].$$

The function $\tau$ incentivizes $i$'s *strict truthfulness* (is *i*-s·truthful) if the inequality is strict whenever the dataset $D_i$ results in different model parameters ($\mathcal{A}(D_i) \neq \mathcal{A}(\bar{D}_i)$).

Strict truthfulness is preferred as trivial DVFs (e.g., $\tau(D_i) = 0$) are *i*-truthful. In practice, each source $i$ need not compute expectations, but must know that truthful submission maximizes its value on average. Next, Prop. 3.2 specifies the assumptions and a DVF that satisfies Def. 3.1.

**Proposition 3.2.** *Let $\mathcal{T}$ denote the RV for the mediator's held-out validation set and $T^*$ be the actual held-out vali-*

*dation set unknown to sources. Let $\mathcal{D}_i$ and $\mathcal{D}_C \triangleq \bigcup_{i \in C} \mathcal{D}_i$ denote the RVs representing source $i$'s dataset and coalition $C$'s dataset, respectively. Define*

$$v(D_C) \triangleq \log p(\mathcal{T} = T^* | \mathcal{D}_C = D_C) - \log p(\mathcal{T} = T^*) \,.$$

*Assume that all sources agree*

*A1 on a Bayesian model $\mathcal{A}$ (e.g., Bayesian linear regression), the prior distribution of model parameters $p(\theta)$, the relationship between the validation set and the model parameters (i.e., $p(\mathcal{T}|\theta)$), and the (possibly different) relationships between each source's dataset and the model parameters (e.g., $p(\mathcal{D}_i|\theta)$), and*

*A2 the mediator's observed $T^*$ is drawn from $p(\mathcal{T}|\theta)$.*

*The value $v(D_i)$ of source $i$'s dataset is $i$-s·truthful (Def. 3.1) with $\mathcal{T}$ as the dependent unknown RV. Formally,*

$$\mathbb{E}_{\mathcal{T}|\bar{D}_i}\left[v(D_i)\right] \le \mathbb{E}_{\mathcal{T}|\bar{D}_i}\left[v(\bar{D}_i)\right] \,.$$

*Moreover, if we additionally assume each source $i$*

*A3 believes that every other source $j$'s submitted dataset $D_j$ is drawn from $p(\mathcal{D}_j|\theta)$,*

*the value $v(D_C)$ of coalition $C \ni i$ is $i$-s·truthful with $(\mathcal{T}, \mathcal{D}_{C \setminus \{i\}})$ as the dependent unknown RVs. Formally,*

$$\mathbb{E}_{\mathcal{T},\mathcal{D}_{C \setminus \{i\}}|\bar{D}_i}[v(D_C)] \le \mathbb{E}_{\mathcal{T},\mathcal{D}_{C \setminus \{i\}}|\bar{D}_i}[v(D_{C \setminus \{i\}} \cup \bar{D}_i)] \,.$$

*The inequalities above are strict whenever the dataset $D_i$ and true dataset $\bar{D}_i$ result in different model parameters.*

We prove Prop. 3.2 in App. C.2 and justify why the assumptions (including the necessity of a validation set) are reasonable in App. A. While Zheng et al. (2024) have proven the first part of Prop. 3.2 that the value of source $i$'s dataset is $i$-s·truthful, we further prove that the value $v(D_C)$ of coalition $C$ is also $i$-s·truthful by adding assumption A3. Our result suggests that source $i$ should still be truthful when there are other sources in the coalition contributing data and affecting source $i$'s data value/reward (for **(F)** in Sec. 2). Sec. 6 empirically evaluates if **(T)** holds and the impact of varying validation set size and model misspecifications (such as misspecifying $p(\mathcal{T}|\theta)$). Our DVF $v$ satisfies several desirable properties in addition to **(T)**:

- $v$ is an **effective measure of model performance**. Let $\mathbf{X}^*$ and $\mathbf{y}^*$ denote the input matrix and the output vector of the validation set. For discriminative models, which consider $\mathbf{X}^*$ as known and model $p(\mathcal{Y} = \mathbf{y}^*|\theta, \mathbf{X}^*)$ instead of $p(\mathcal{T} = T^*|\theta)$, $v$ corresponds to a scaled and translated version of the log-likelihood (or log predictive density) of the validation set. The log predictive density is widely used to evaluate Bayesian models and quantify how well the posterior predictive distribution, $p(\mathcal{Y}|\mathbf{X}^*, \mathcal{D}_i = D_i)$, aligns with the observed outputs

$\mathbf{y}^*$.[2] Submitting $D_i$ with noisy data or duplicated data is expected to degrade the log predictive density — the predictive mean $\mathbb{E}[\mathcal{Y}|\mathbf{X}^*, \mathcal{D}_i = D_i]$ may deviate further from $\mathbf{y}^*$ or the predictive uncertainty decreases, respectively. The latter is undesirable as the Bayesian model may assign higher confidence to inaccurate predictions.

- $v$ is easy to compute and numerically stable, with closed form expressions for exponential family models. For more complex models, $v$ can be **efficiently** approximated using Monte Carlo sampling or variational inference (Murphy, 2022), as implemented in probabilistic ML libraries (Phan et al., 2019).The time complexity tends to be linear w.r.t. the number of sampling steps, the number of model parameters and the size of the dataset $D_C$. See (Zheng et al., 2024) for alternative efficient computation methods.

- The expected value of $v(D_C)$ over all possible realizations of $D_C$ **represents the mutual information** $\mathbb{I}(\mathcal{D}_C, \mathcal{T})$, that is, the expected reduction in uncertainty about the validation set $\mathcal{T}$ due to $C$'s data. Due to the "information-never-hurt" property of entropy (Cover & Thomas, 1991), observing more data cannot decrease $\mathbb{I}(\mathcal{D}_C, \mathcal{T})$. Thus, the DVF is expected to be **monotonic**: When $C' \subseteq C$, $\mathbb{E}[v(D_C)] \ge \mathbb{E}[v(D_{C'})]$.

- A source without data will have no value, i.e., $v(\emptyset) = 0$.

*Remark.* Do other validation-set-based DVFs, such as accuracy, satisfy Def. 3.1 when the validation set is unknown? Validation accuracy may not be $i$-s·truthful because a source can perturb its data to result in different model parameters (e.g., add backdoor triggers to its data (Saha et al., 2020)) as long as it does not change the most likely class. The study of non-Bayesian models is left to future work as it is harder to prove Def. 3.1 when their training involves an extra step of loss minimization rather than directly computing probabilities based on the model specifications.

## 4. Fair and Truthful Semivalues

In Sec. 2, we have described how using the semivalues can ensure **(F)** collaborative fairness. In this section, we prove that semivalues $(\phi_i[\nu_{\mathfrak{D}}^v])_{i \in N}$ based on the above DVF are **(T)** $i$-s·truthful too, which has *not* been studied by existing works in Tab. 2. Compared to $v(D_i)$, the semivalue $\phi_i[\nu_{\mathfrak{D}}^v]$ of source $i$ includes additional terms, such as $v(D_{C \cup \{i\}})$ and $-v(D_C)$, which depend on other sources' data. Thus, proving that the semivalue $\phi_i[\nu_{\mathfrak{D}}^v]$ satisfies Def. 3.1 and that its expected value is maximized by source $i$ submitting $\bar{D}_i$ is non-trivial. Next, Prop. 4.1 states that semivalues based on a DVF from Prop. 3.2 also preserve Def. 3.1.

**Proposition 4.1.** *Let $\nu_{\mathfrak{D}, i \to \bar{D}_i}^v$ denote the characteristic*

---

[2]See App. B.4.5 for additional details, e.g., formula for Gaussian distributions.

*function computed using $i$'s true dataset $\bar{D}_i$ and dataset $D_j$ for $j \neq i$. Assuming A1-A3 and using DVF $v$ from Prop. 3.2, the semivalue $\phi_i[\nu^v_{\mathfrak{D}}]$ for source $i$ is $i$-s·truthful (Def. 3.1) with $(\mathcal{T}, \mathcal{D}_{N \setminus \{i\}})$ as the dependent unknown RV. Formally, for any $D_i$, the expected semivalue from submitting $D_i$ is not greater than that from submitting $\bar{D}_i$ with strict inequality when $D_i$ results in different model parameters:*

$$\mathbb{E}_{(\mathcal{T}, \mathcal{D}_{N \setminus \{i\}})|\bar{D}_i} \left[ \phi_i[\nu^v_{\mathfrak{D}}] \right] \leq \mathbb{E}_{(\mathcal{T}, \mathcal{D}_{N \setminus \{i\}})|\bar{D}_i} \left[ \phi_i[\nu^v_{\mathfrak{D}, i \to \bar{D}_i}] \right].$$

We prove Prop. 4.1 in App. D.1 using Def. 2.1 and Prop. 3.2. Prop. 4.1 implies that a truthful Nash equilibrium exists. When source $i$ knows that every other source $j \in N \setminus \{i\}$ submits its true dataset $\bar{D}_j$ (and agrees that it is drawn from $p(\mathcal{D}_j | \theta)$), source $i$ expects that submitting $\bar{D}_i$ maximizes its semivalue $\phi_i[\nu^v_{\mathfrak{D}}]$. **Thus, source $i$ has no incentive to deviate unilaterally from truthful submission.** Moreover, when the mediator uses the Shapley value, no better equilibrium exists: By the group rationality property of the Shapley value (Jia et al., 2019), the total reward $\sum_{i \in N} \phi_i[\nu^v_{\mathfrak{D}}]$ equals $v(D_N)$. Applying Prop. 3.2 to the aggregated dataset $D_N$, the total expected reward $\mathbb{E}[v(D_N)]$ is maximized when the submitted $\mathfrak{D}$ leads to the same inference about the model parameters as the true datasets $(\bar{D}_i)_{i \in N}$. As other equilibria yield less total rewards and lack the natural symmetric Schelling point of honesty, they are difficult for sources to coordinate on (Dorner et al., 2023) and **all sources will prefer the easier truthful Nash equilibrium, justifying assumption A3.** Together, Prop. 3.2 and Prop. 4.1 provably ensure (**F**) collaborative fairness and incentivize (**T**) truthfulness at equilibrium for Bayesian models.

*Remark on semivalue weights' impact on (T) and (F).* (i) When $w_0 = 1, w_{c \neq 0} = 0$, the semivalue $\phi_i[\nu^v_{\mathfrak{D}}] = v(D_i)$ always incentivizes (**T**) regardless of what other sources submit. However, these weights do not satisfy the constraint required for F3 and (**F**). Conversely, (ii) when $w_c > 0$ for all $c$ instead, the semivalues ensure (**F**), but ensure (**T**) *only* under assumption A3. Setting larger weights for smaller coalitions (such as $w_0$) may reduce the influence of other potentially untruthful sources. The impact on fairness and truthfulness is demonstrated in Fig. 14.

*Remark on efficiency.* Computing the semivalue $\phi_i$ exactly requires evaluation of $2^n$ coalitions, which may be costly for large $n$. In our setting, however, the number of sources is typically small (e.g., a few hospitals). When $n$ is larger, efficient unbiased estimators $\hat{\phi}_i$ using only $O(n \log n)$ samples (Jia et al., 2019; Kolpaczki et al., 2024; Li & Yu, 2024) can be used. As the expected value of the unbiased estimator equals the exact semivalue $\phi_i[\nu^v_{\mathfrak{D}}]$, such estimators are also fair and $i$-s·truthful: the expected value of $\hat{\phi}_i[\nu^v_{\mathfrak{D}}]$ (over samples and the unknown RV) is no greater than that of $\hat{\phi}_i[\nu^v_{\mathfrak{D}, i \to \bar{D}_i}]$. In Tab. 3, we empirically verify this on 20 sources using unbiased estimators.

In data valuation (Ghorbani & Zou, 2019; Sim et al., 2022), we often care about the ranking of the semivalues instead. For example, the mediator may only use data from sources with the largest semivalues. Prop. 4.2 (proven in App. D.2) explains why if source $i$'s expected semivalue is less than $k$'s when it submits $\bar{D}_i$, **source $i$ cannot improve its semivalue ranking (over $k$) by submitting an alternative dataset** as it may increase the expected shortfall instead.

**Proposition 4.2.** *From each source $i$'s perspective, the expected decrease in its own semivalue from submitting alternative dataset $D_i$ instead of $\bar{D}_i$ is not smaller than the decrease in the semivalue of any other source $k \neq i$:*

$$\mathbb{E}_{(\mathcal{T}, \mathcal{D}_{N \setminus \{i\}})|\bar{D}_i} \left[ \phi_i[\nu^v_{\mathfrak{D}, i \to \bar{D}_i}] - \phi_i[\nu^v_{\mathfrak{D}}] \right]$$
$$\geq \mathbb{E}_{(\mathcal{T}, \mathcal{D}_{N \setminus \{i\}})|\bar{D}_i} \left[ \phi_k[\nu^v_{\mathfrak{D}, i \to \bar{D}_i}] - \phi_k[\nu^v_{\mathfrak{D}}] \right].$$

## 5. Other Incentives or Constraints

### 5.1. Limited Budget

We first consider the constraint where the mediator has a limited reward budget to distribute among data sources: When a mediator gives *monetary* rewards, the budget is often limited (e.g., profits generated by the ML model) (Jia et al., 2019; Chen et al., 2020); when a mediator rewards each source *with an ML model*, the model performance (and the value of data from all sources) is inherently bounded (Sim et al., 2020). To ensure the *feasibility* or *budget balance* incentive, can we transform each source's semivalue $\phi_i[\nu^v_{\mathfrak{D}}]$ so that the reward $r_i$ does not exceed the budget $B$, i.e., $r_i \leq B$? A natural solution is to scale the semivalue by a factor $a$, i.e., $r_i \triangleq \phi_i[\nu^v_{\mathfrak{D}}]/a$.

When the factor $a$ depends on source $i$'s submission (e.g., the RV $A \triangleq \max_{k \in N} \phi_k[\nu^v_{\mathfrak{D}}]$), it is infeasible to prove that $r_i$ is $i$-truthful because $\phi_i[\nu^v_{\mathfrak{D}}]/A$ becomes non-linear w.r.t. $v(D_{C \cup \{i\}})$ and cannot be simplified (e.g., $\mathbb{E}[\phi_i[\nu^v_{\mathfrak{D}}]/A]$ is not bounded by $\mathbb{E}[\phi_i[\nu^v_{\mathfrak{D}}]]/\mathbb{E}[A]$). Thus, to provably guarantee that $r_i$ is $i$-truthful, the mediator must decide the factor $a$ in $r_i$ *beforehand*, for example, by using historical data and discussion with sources. As the proposed DVF in Sec. 3 is unbounded, the scaled semivalue $\phi_i[\nu^v_{\mathfrak{D}}]/a$ might exceed $B$. This is corrected by using a bounded *feasible* reward value $r_i \triangleq \min(\phi_i[\nu^v_{\mathfrak{D}}]/a, B)$ that is $i$-s·truthful only when $\phi_i[\nu^v_{\mathfrak{D}}]/a \leq B$, and $i$-truthful otherwise. In the otherwise case, source $i$ can report any dataset that results in different model parameters as long as its semivalue exceeds $aB$.

### 5.2. Removing Validation Set

In some privacy-sensitive applications, the mediator may find it hard to procure a validation set (Sim et al., 2023). Is it possible to ensure collaborative fairness *and* ensure truthful submission maximizes expected rewards *without*

a held-out validation set? As the DVF must still depend on other information unknown to source $i$ to incentivize truthfulness (Sec. 3), each source $i$'s data must be evaluated using other sources' data as the validation set.

In Sec. 2, the definition of (**F**) collaborative fairness considers a **common** DVF that correlates with model performance and reward values that satisfy four fairness conditions. Chen et al. (2020) incentivize strict truthfulness of source $i$ by directly rewarding with $v^{-i}(D_i) \triangleq \log p(\mathcal{D}_{N\setminus\{i\}} = D_{N\setminus\{i\}}|\mathcal{D}_i = D_i) - \log p(\mathcal{D}_{N\setminus\{i\}} = D_{N\setminus\{i\}})$. This approach does not ensure (**F**) since each source $i \in N$'s reward (a) is decided by a different DVF (e.g., $v^{-i}(D) \neq v^{-j}(D)$) and (b) does not depend on how the dataset $D_{N\setminus\{i\}}$ is partitioned among the other sources. (a) matters as the difference in data values/rewards should reflect the differences from the dataset $D$ instead of from the DVF. (b) matters as the partition influences whether source $i$ provides as much value in $\mathfrak{D}$ as in $\mathfrak{D}'$ in every coalition, so setting the same reward $v^{-i}(D_i)$ in both cases may violate fairness condition F3.

To correct (a), we use the same set of multiple DVFs to decide the rewards of all sources. To correct (b), we decide the reward values based on semivalues and use each DVF to evaluate the data of all coalitions that can be formed by $n$ sources. As the validation set and training data should be disjoint, for each source $j$, we randomly split its data into a validation set $T_j$ and a remaining dataset $D_j$ (e.g., 50% each). We consider $n$ cooperative games where, in the $j$-th game, $T_j$ serves as the validation set with $\mathcal{T}_j$ denoting the RV. The valuation is based on the remaining $\mathfrak{D} = (D_j)_{j\in N}$. The characteristic function $\nu_{\mathfrak{D}}^{T_j}$ is defined as

$$\nu_{\mathfrak{D}}^{T_j}(C) \triangleq \log p(\mathcal{T}_j = T_j|\mathcal{D}_C = D_C) - \log p(\mathcal{T}_j = T_j).$$

By our earlier results (Prop. 4.1), for any source $i \neq j$, truthfully submitting $\bar{D}_i$ maximizes its expected data values and semivalue. Specifically, for any $C \ni i$,

$$\mathbb{E}_{\mathcal{D}_{N\setminus\{i\}}, \mathcal{T}_j|\bar{D}_i}\left[\nu_{\mathfrak{D}}^{T_j}(C)\right] \leq \mathbb{E}_{\mathcal{D}_{N\setminus\{i\}}, \mathcal{T}_j|\bar{D}_i}\left[\nu_{\mathfrak{D}, i\to\bar{D}_i}^{T_j}(C)\right],$$

$$\mathbb{E}_{\mathcal{D}_{N\setminus\{i\}}, \mathcal{T}_j|\bar{D}_i}\left[\phi_i[\nu_{\mathfrak{D}}^{T_j}]\right] \leq \mathbb{E}_{\mathcal{D}_{N\setminus\{i\}}, \mathcal{T}_j|\bar{D}_i}\left[\phi_i[\nu_{\mathfrak{D}, i\to\bar{D}_i}^{T_j}]\right].$$

At first glance, we can correct (a) and (b) by setting the reward value as $\dot{r}_i \triangleq \sum_{j\in N}\phi_i[\nu_{\mathfrak{D}}^{T_j}]$ where source $i$'s data is also evaluated on its own split validation set. However, Prop. 4.1 does not hold for $j = i$ and our experiments (e.g., Fig. 16a) show that (**T**) truthfulness no longer holds as each source $i$ can artificially increase its own value $\nu_{\mathfrak{D}}^{T_i}(D_i)$, semivalue $\phi_i[\nu_{\mathfrak{D}}^{T_i}]$, and the reward $\dot{r}_i$ by ensuring that its remaining dataset $D_i$ predicts its split validation set $T_i$ well.

Thus, to ensure (**T**) truthfulness, the reward value of source $i$ must exclude $\phi_i[\nu_{\mathfrak{D}}^{T_i}]$. More precisely, we set the reward value for source $i$ as $\check{r}_i \triangleq \sum_{j\in N\setminus\{i\}}\phi_i[\nu_{\mathfrak{D}}^{T_j}]$ (see Fig. 5) which corrects (b) and largely corrects (a) (as $\check{r}_i$

and $\check{r}_j$ depend on the same overlapping subset of DVFs based on the other $n - 2$ sources). The following conditions hold: F0, (i) **Modified Symmetry:** For two sources $i$ and $j$ who contribute equally to every coalition under every validation set (i.e., $\nu_{\mathfrak{D}}^{T_j}(C \cup \{i\}) = \nu_{\mathfrak{D}}^{T_j}(C \cup \{j\})$ for all $C \subseteq N \setminus \{i,j\}$), they receive equal rewards $\check{r}_i = \check{r}_j$ only if $\phi_i[\nu_{\mathfrak{D}}^{T_j}] = \phi_j[\nu_{\mathfrak{D}}^{T_i}]$. This may occur when both sources have identical datasets and the mediator uses the same random seed to split them into validation and remaining datasets. (ii) **monotonicity** (F2) and (iii) **strict monotonicity** (F3) still hold based on the new validation sets. For example, source $i$ should receive a higher reward $r_i$ if its data provides at least as much value in $\mathfrak{D}$ as in $\mathfrak{D}'$ when evaluated on every other source's validation set, i.e., $(\forall j \neq i\ \nu_{\mathfrak{D}}^{T_j} \succeq_i \nu_{\mathfrak{D}'}^{T_j}) \implies \check{r}_i \geq \check{r}_i'$. (ii) and (iii) are nontrivial as they are not satisfied by existing solution (Chen et al., 2020). The relaxation in (i) is needed as while setting the reward value as $\dot{r}_i$ or a constant would satisfy F1, they are not $i$-s·truthful. In App. E.3, we consider strictly enforcing (**F**) and explain why (**T**) may not hold.

# 6. Experiments

This section empirically validates **[I]** whether our DVF incentivizes truthfulness under different validation sets, **[II]** whether the corresponding semivalues incentivize truthfulness and collaborative fairness, and **[III]** how these incentives are impacted when the mediator has limited budget or lacks a validation set. As comparisons with DVFs that are provably untruthful (e.g., information gain, volume; see App. C.1) or unfair by design (Sec. 5.2) are less meaningful, we compare against the closest theoretical baseline (Zheng et al., 2024) in Fig. 13. We also do not claim to outperform accuracy- or similarity-based DVFs without theoretical truthfulness guarantees (Ghorbani & Zou, 2019; Xu et al., 2021a).

When truthfulness is incentivized, source 0 submitting a dataset that fully captures its knowledge should lead to higher value and reward over alternative strategies that degrade model performance or artificially inflate knowledge. We compare the [**T**] truthful strategy with five untruthful strategies: [**S**] sharing only a subset of data (50%), [**N**] adding noise to outputs **y** (mislabeling a fraction of the output labels in classification datasets; adding Gaussian noise to outputs of regression datasets), [**D**] duplicating the dataset (concatenating 3 copies), [**I**] injecting mislabeled synthetic data outside the observed input domain, or [**P**] adding Gaussian noise to each feature in each input. We evaluate four Bayesian models and datasets with 3 sources:[3]

---

[3]Sim et al. (2020) also use 3 sources to demonstrate the validity. More experiments (e.g., with 9/20 sources) and experimental details are given in App. F.

- **Gaussian process regression with the Friedman dataset (GP-FR):** We generate $400, 300,$ and $300$ data points for the respective sources 0, 1, and 2 using the synthetic Friedman function with standard Gaussian noise: $\mathbf{y} = 10 \sin(\pi \mathbf{X}_{[:,0]} \mathbf{X}_{[:,1]}) + 20(\mathbf{X}_{[:,2]} - 0.5)^2 + 10 \mathbf{X}_{[:,3]} + 5 \mathbf{X}_{[:,4]} + 0 \mathbf{X}_{[:,5]} + \mathcal{N}(0, 1)$. Each input in $\mathbf{X}$ has 6 features independently drawn from the standard uniform distribution. We standardize the output $\mathbf{y}$ and train a GP model with a squared exponential kernel and different lengthscales for each feature.

- **Bayesian binary logistic regression with the Heart disease dataset (LO-HE)** (Bharti et al., 2021): Healthcare firms aim to predict whether a patient has heart disease based on features like resting blood pressure. After preprocessing (outlier removal and standardization), we split 990 data points among sources 0-2 and the mediator's validation set in the ratio $[.3, .2, .2, .3]$.

- **Bayesian neural network regression with the Cycle Power Plant dataset (NN-CY)** (Tfekci & Kaya, 2014): Power plants aim to predict net electrical energy output based on four features, such as temperature and ambient pressure. The dataset contains 9,568 points, of which $25\%$ are reserved for validation. The remaining data is split among sources 0-2 in the ratio $[.4, .3, .3]$. We train a Bayesian neural network with one ReLU hidden layer of 100 units. The output is perturbed with Gaussian noise (mean 0, variance drawn from an inverse-gamma prior with shape and scale parameters $(3, 1)$).

- **Bayesian multinomial logistic regression with the Blood MNIST dataset (LO-BL)** (Yang et al., 2021) in App. F: We extract image embeddings from the BloodMNIST dataset using a pretrained ResNet-18 model and select the top 50 features via principal component analysis (PCA). Healthcare firms seek to classify blood cells into eight types. After preprocessing, there are 10,927 and 3,007 images in the training and validation sets, respectively. Sources 0-2 have different cell types distribution; for example, source 0 has fewer class 0 samples.

In all non-GP models, we use a standard Gaussian prior for the weights and estimate the data value using posterior samples generated via the No-U-Turn Sampler (NUTS) (Hoffman & Gelman, 2014), running four chains with 1,000 burn-in and 2,000 sampling iterations per chain. NUTS makes no assumptions about the posterior distribution, avoids extra approximations to compute the validation set log-likelihood, and provides asymptotically exact estimates.

## 6.1. DVF

In Fig. 1a-c, we plot the value $v(D_0)$ of source 0 when evaluated on validation sets of different sizes. Across all datasets, it can be observed that source 0's strategy T consistently leads to its highest value compared to other untruthful strategies, regardless of the validation set size. Larger validation

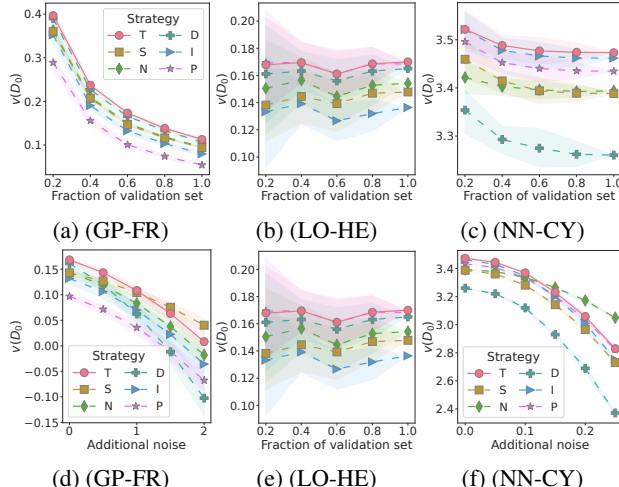

(a) (GP-FR)  (b) (LO-HE)  (c) (NN-CY)

(d) (GP-FR)  (e) (LO-HE)  (f) (NN-CY)

*Figure 1.* Graphs of value of source 0 (with 95% CI shaded across 20 sets) under its different strategies when evaluated on validation sets of (a-c) increasing sizes and (d-f) with increasing noise added to the outputs for various datasets.

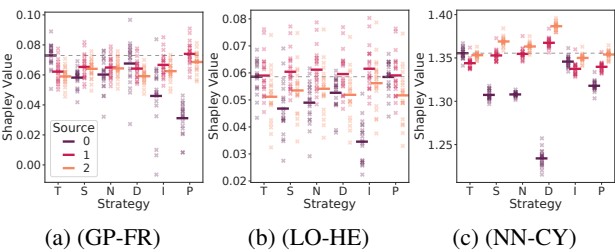

(a) (GP-FR)  (b) (LO-HE)  (c) (NN-CY)

*Figure 2.* Graphs of all sources' Shapley values (mean across 20 validation sets plotted as straight bars) when source 0 uses different strategies for various datasets (a)–(c).

sets result in smaller variance in the data value.

Next, we empirically evaluate if truthfulness is still incentivized under Bayesian model misspecifications. In Fig. 1d-f, we plot the value $v(D_0)$ of source 0 with increasing levels of noise added to the validation set outputs. Across all datasets, source 0's strategy T achieves its highest value compared to other untruthful strategies when the noise is low. However, as noise increases and the validation set likelihood $p(\mathcal{T}|\theta)$ in A2 becomes more misspecified, $v(D_0)$ may not be empirically $i$-s·truthful for source 0. Untruthful strategies, such as submitting a subset (S in Fig. 1d) or a noisy dataset (N in Fig. 1e-f), may result in a higher log-likelihood of the noisy validation set than strategy T as they produce a higher predictive uncertainty. In App. F.2, it can be similarly observed that under minor misspecifications of $p(\mathcal{T}|\theta)$ (by generating the validation set with modified parameters in the Friedman function) and $p(\mathcal{D}_1|\theta)$, or a validation set with different input distribution, $v(D_0)$ and $v(D_{\{0,1\}})$ are still $i$-s·truthful for source 0. Thus, truthfulness still holds under minor violations of assumptions A2 and A3.

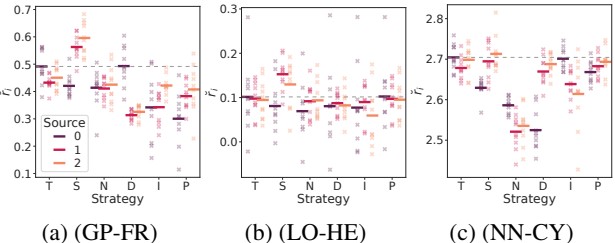

(a) (GP-FR)   (b) (LO-HE)   (c) (NN-CY)

*Figure 3.* Graphs of all sources' reward values $\check{r}_i$ (mean across 10 training and validation splits plotted as straight bars) when source 0 uses different strategies for various datasets (a)–(c).

## 6.2. Truthful Semivalues

In Fig. 2, we consider source 0 using various strategies while all other sources submit true datasets. We report the Shapley values computed for 20 different subsets of the validation set. Across all datasets, source 0's strategy T consistently leads to its highest Shapley value. Moreover, collaborative fairness is evident: The Shapley values of other sources are higher when source 0 submits a less valuable untruthful dataset (e.g., with strategies S and N) than their Shapley values when source 0 is truthful. This reflects that other sources' data become more informative and important for accurate predictions when source 0 is untruthful. In App. F.3, Fig. 14 shows that with larger weight on smaller coalitions, there is less fairness as sources 1's and 2's rewards become less influenced by source 0's strategies. Even when source 1 untruthfully mislabels part of its dataset $D_1$, source 0's semivalue is still usually maximized by its strategy T.

## 6.3. Other Constraints

Firstly, we consider the scenario with a **limited budget**. We reuse Fig. 2a and consider $a = 1$, $B = .06$ (Sec. 5.1), and $r_i = \min(\phi_0[\nu_{\mathfrak{D}}^v], .06)$. As strategy D results in $\phi_0[\nu_{\mathfrak{D}}^v] > .06$, it can also result in the same maximum reward .06 for source 0 as strategy T. Thus, $r_i$ is only $i$-truthful.

Next, when the mediator **does not have a validation set**, the mediator partitions each source's dataset into training and validation sets in a $75\% - 25\%$ split and computes the semivalue of each source on all resulting validation sets, as described in Sec. 5.2. The reward value for each source $i$ is computed as $\check{r}_i = \sum_{j \in N \setminus \{i\}} \phi_i[\nu_{\mathfrak{D}}^{T_j}]$, making it independent of its own validation set. In Fig. 3, we consider source 0 using various strategies when other sources submit their true datasets. We report the reward values $\check{r}_i$ computed for 10 different training and validation set splits. Across all datasets, it can be observed that source 0's strategy T consistently leads to its highest reward value.

Unlike in Sec. 6.2, the reward values of other sources do not always increase when source 0 uses an untruthful strategy. In Fig. 3a-c, source 0's strategy I decreases the reward of sources 1 and 2 compared to source 0's strategy T. This

difference arises because the log-likelihood of observing source 0's untruthful validation set $T_0$ may be significantly lower than that of observing $\bar{T}_0$. Thus, for any source $j \neq 0$, $\phi_j[\nu_{\mathfrak{D}}^{T_0}] < \phi_j[\nu_{\mathfrak{D}, 0 \to \bar{D}_0}^{\bar{T}_0}]$, leading to a much lower $\check{r}_j$ when source 0 submits an untruthful $D_0$. This observation does not violate fairness — the change in the validation set may violate the premise of (ii) monotonicity. App. F.5 investigates another $50\% - 50\%$ train-validation set split and shows that if source $i$'s reward includes $\phi_i[\nu_{\mathfrak{D}}^{T_i}]$, $i$ can use untruthful strategies to artificially increase its reward and make its remaining dataset predict its validation set well.

## 7. Related Works

**Collaborative Fairness (F).** Many existing works ensure **(F)** by deciding reward values using the Shapley value or other semivalues. Ghorbani & Zou (2019); Jia et al. (2019); Kwon & Zou (2021) have proposed validation accuracy as the DVF, while Xu et al. (2021b); Sim et al. (2020; 2023) have proposed validation-set-free DVFs, such as dataset diversity/volume, information gain, or Bayesian surprise the data elicits about the model parameters.[4] But, these works do not incentivize or verify the truthfulness of data submitted. Validation-set-free DVFs may incentivize untruthful submissions as sources can increase their rewards with low-effort strategies like data duplication (App. C.1). In contrast, we propose a DVF that satisfies our truthfulness definition and specify the necessary assumptions (Sec. 3).

**Incentivizing Truthful Data Submission.** Faltings (2022) give an overview of game-theoretic mechanisms for eliciting accurate information. In this work, we focus on mechanisms to elicit true data from sources jointly training an ML model for a common prediction task with a shared ground truth (e.g., heart disease in a population). A mechanism is *truthful* if sources expect the highest reward by reporting the data they believe are correct. Peer consistency mechanisms (Miller et al., 2005; Jurca & Faltings, 2011; Faltings et al., 2017; Dorner et al., 2023; Chen et al., 2020) evaluate a source based on the agreement between its submission and those of randomly selected peers. but may fail to ensure **(F)** as rewards are not based on the same DVFs and all sources' data values. To address this, we assume access to a ground truth validation set, as in Richardson et al. (2020); Zheng et al. (2024). Zheng et al. (2024) also use Bayesian models and pointwise mutual information between each source's data and the mediator's held-out validation set. Our DVF in Prop. 3.2 additionally considers each *coalition's* data value as it is required for computing semivalues. Unlike these prior works, our work ensure **(F)** conditions (Sec. 2) and prove the existence of a truthful equilibrium where every source maximizes its semivalue reward (Sec. 4).

---

[4]App. B.1 discusses why incentivization mechanisms in federated learning are less suitable comparisons.

# 8. Conclusion

This paper proposes the first mechanism that provably ensures *both* collaborative fairness and incentivizes truthfulness, and comprehensively evaluates the mechanism on multiple datasets. The novelty lies in identifying the reasonable assumptions and analysis needed for the proof and suitable prior works to build upon — works that achieved truthfulness or fairness alone, such as pointwise mutual information (Zheng et al., 2024). Apps. G and H, respectively, address some limitations and some questions a reader may have.

## Impact Statement

This paper presents work whose goal is to advance the field of Machine Learning. There are many potential societal consequences of our work, none of which we feel must be specifically highlighted here.

## Acknowledgments

This research is supported by the National Research Foundation Singapore and the Singapore Ministry of Digital Development and Innovation, National AI Group under the AI Visiting Professorship Programme (award number AIVP-2024-001). Jue Fan and Xiao Tian are supported by Agency for Science, Technology and Research (A*STAR) Graduate Academy. The authors would also like to thank the anonymous reviewers and AC for their helpful feedback.

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

# A. Assumptions

**Assumption A.1** (No cloning). Each source can only submit *one* dataset for valuation. Thus, a source cannot increase its reward by submitting multiple datasets as different sources.

Assum. A.1 is realistic as the source's identity can be verified (e.g., through a business registration ID) and has been similarly assumed by Chen et al. (2020); Zheng et al. (2024). Additionally, other works have addressed the replication robustness issue alone. For example, Han et al. (2022) propose a robust variant of the Shapley value which ensures that the total reward of a source from cloning and multiple submissions is lower than the reward without cloning.

**Assumption A.2** (Agreement on probabilistic models). All sources agree on (or trust the mediator to decide) a Bayesian model $\mathcal{A}$ (e.g., Bayesian logistic regression), the prior model parameters $p(\theta)$, the relationship between the validation set and the model parameters (i.e., $p(\mathcal{O}|\theta)$), and the (possibly different) relationships between each source's dataset and the model parameters (e.g., $p(\mathcal{D}_i|\theta)$).

We focus on Bayesian models as they naturally produce probabilistic predictions that account for both prior knowledge and the observed data. Such probabilistic predictions are essential for Def. 3.1 and applying proper scoring rules (see App. B.5). The focus is not overly restrictive as Bayesian models can model complex relationships. In App. B.4, we provide a background on Bayesian models.

Assum. A.2 is essential as the mediator uses these quantities to compute the posterior of the model parameters and the validation set (i.e., $p(\mathcal{T} = T^*|\mathcal{D}_i = D_i)$) during data valuation.

Assum. A.2 is reasonable as the sources can discuss and agree on the model specifications and relationships based on historical, public, or similar data at the start of the collaboration. The sources can also use majority voting to select the best common prior belief. Furthermore, our approach is consistent with other existing Bayesian data valuation methods (Sim et al., 2020; Chen et al., 2020), which also assume a known and agreed-upon prior.

Notably, Assum. A.2 does *not* require all sources' datasets to share the same relationship with the model parameters. For example, in a Bayesian regression model with weights $\theta$ to predict housing prices, source $i$ can inform the mediator if its output (housing prices) has a larger noise variance (in $p(y_i|\theta, X_i)$) or differs from others by a constant due to different pricing strategies (e.g., $y_i = X_i\theta + \text{constant}$). Hence, source $i$ does not need to alter its dataset to match the dataset distribution of others.

**Assumption A.3** (Validation Set). All sources believe that the mediator's held-out validation set $T^*$ is drawn from $p(\mathcal{T}|\theta)$.

Assum. A.3 is needed so that each source would hypothetically consider $p(\mathcal{T}|\theta)$ when computing the expectation in the truthfulness definition. If a source suspects that $T^*$ was from another random/noisy distribution instead, it may not truthfully submit its data.

Assum. A.3 is sensible as the mediator can be trusted to obtain a validation set from actual evaluations and to suggest an appropriate model. Moreover, the mediator can use outlier detection techniques, such as residual analysis (Murphy, 2022), to remove data that do not fit the model specifications and are unlikely to be from $p(\mathcal{T}|\theta)$. Using a validation set is a common practice in data valuation works (Jia et al., 2019; Ghorbani & Zou, 2019), and it serves as a foundation for ensuring truthfulness (Richardson et al., 2020).

To our knowledge, works that remove the ground truth assumption can only incentivize and ensure truthfulness in limited settings (e.g., multiple independent and homogeneous tasks, agents' signals on a task being conditionally independent given the ground truth, discrete predictions, or knowing the misclassification rate for the noisy ground truth in binary classification (Liu et al., 2023)). Thus, removing this validation set assumption is highly non-trivial.

**Assumption A.4** (Other Sources). Every source $i$ believes that every other source $j$'s submitted dataset $D_j$ is drawn from $p(\mathcal{D}_j|\theta)$.

Assum. A.4 can be satisfied if every source believes that every other source $j$ is truthful and submits its true dataset $\bar{D}_j$ and believes that $\bar{D}_j$ is drawn from $p(\mathcal{D}_j|\theta)$.

Assum. A.4 is used to show the existence of a truthful equilibrium — when other sources are truthful, it is optimal for source $i$ to also be truthful. This assumption is reasonable, as sources may believe that converging to a truthful equilibrium is easier. It is commonly used in existing works (Chen et al., 2020; Liu & Wei, 2020; Richardson et al., 2020; Dorner et al., 2023).

Additionally, in practice, a source $j$ can improve its true dataset $\bar{D}_j$ to align with its model specifications $p(\mathcal{D}_j|\theta)$ by removing outliers from its raw data. This can be done using outlier detection techniques, such as residual analysis (Murphy, 2022).

## B. Background

### B.1. Collaborative Machine Learning, Federated Learning and Data Valuation

In this section, we highlight the differences between collaborative ML, federated learning, and data valuation.

Data valuation is concerned with how much each source's data contributes to an ML model performance. A source's data can first be independently valued using a performance metric (e.g., accuracy) before being valued relative to the data contributed by others (e.g., semivalues) (Sim et al., 2022). Intuitively, a source should receive a higher monetary reward (or a more valuable model as a reward) for contributing more valuable data. For fairness, a source's reward should depend on others' contributions (e.g., lower if their data is redundant due to similarity to others) (Sim et al., 2022).

**Our work involves data valuation, but as we consider the collaborative ML setting described by (Sim et al., 2020), the goal is to elicit larger truthful datasets from *a few* sources (e.g., companies, hospitals) instead of datum from individual owners.** Thus, we do not compare against works (Jia et al., 2019; Ghorbani & Zou, 2019) that consider a crowd-sourcing or interpretable ML setting where a value is assigned to each datum. For those works, scalability may pose a greater challenge.

**Collaborative ML (CML) is a more abstract and thus more general setting than federated learning (FL).**

In FL (McMahan et al., 2017), a model is trained on data from multiple sources without centralizing them. Instead, sources (*clients*) share only updates to improve the model. As the data-processing inequality guarantees that updates cannot contain more information than the raw training data, FL can partially allay sources' privacy concerns. Additionally, FL algorithms also seek to reduce communication costs and address data heterogeneity across sources (McMahan et al., 2017). *Federated averaging* is a popular FL algorithm for training neural networks. In each round, every selected source trains its local neural network by running a few steps of gradient descent on its own data and sends the parameters update to the server. The server aggregates the updates from all selected sources. In the next round, the server broadcasts the aggregated parameters to selected sources for updating their local neural networks.

In contrast, in CML, a model can be trained on data from multiple sources in other ways that may involve centralization. For example, each source can share sufficient statistics (Caragea et al., 2004; Sim et al., 2023) or raw data via secure data sharing platforms (that prevent unauthorized access) such as X-road (Sim et al., 2020). Motivating examples are given in Sec. 1. In CML, one research area focuses on *incentives that sources need to share data (App. E.2) and ways to achieve these incentives*. As any source would have incurred a nontrivial cost to collect its data, they would not altruistically share their data with others. For example, a firm with significant data may be concerned about losing its competitive edge if competitors get the same collaboratively trained model. These sources will be motivated to share their data when given enough incentives and rewards (Sim et al., 2020). Works that incentivize CML fall into two categories: (i) monetary rewards or (ii) non-monetary rewards such as ML models and data derivatives. We briefly overview CML incentivization works that use non-monetary rewards (ii), which we further break down to (ii.a) and (ii.b). In (ii.a), such as Sim et al. (2020), the mediator decides the target reward values (satisfying incentives) and generates model rewards to realize them (e.g., by adding noise to the model parameters). In (ii.b), the mediator decides some quantity that correlates with model quality instead of directly controlling the reward values. For example, in each round of FL, Xu et al. (2021a) fairly decide the number of non-zero components in the gradient rewards. Lin et al. (2023) fairly decide the probability that source $i$ is selected to receive the model updates in each round. Wu et al. (2024) fairly decide the proportions of local model updates that each source receives in each round of FL. (We also note that these works assume and may not incentivize truthfulness. For instance, a source can increase its cosine gradient Shapley value (Xu et al., 2021a) by only submitting data of the majority class and removing its unique data of minority class.)

**Comparison.** This work focuses on (i), as deciding monetary reward values is more straightforward and facilitates theoretical analysis. Deciding how to give out model rewards in (ii) may lead to a conflict in incentives, especially with strict truthfulness (see Sec. 5.1 for a discussion about the issue with limited budget or models with maximum attainable performance). We also focus on **CML rather than FL**, as decentralized and iterative settings affect the valuation and model rewards in (ii) and make strict truthfulness harder to guarantee. Thus, we consider our CML setting in Sec. 2 as a first step and leave

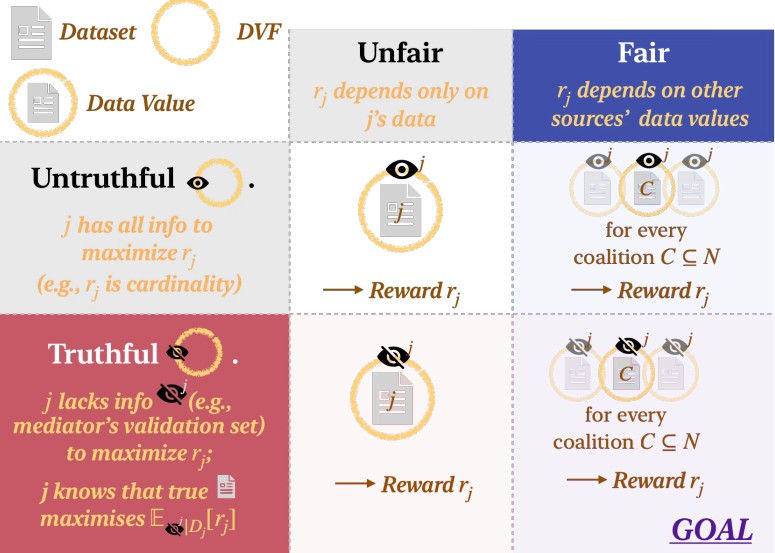

*Figure 4.* The goal is to determine *fair* and *truthful* reward values $r_j$ for each source $j \in N$. The *data valuation function* (DVF, denoted by a circle) evaluates the data value of a source $j$ or coalition $C$ (within). A reward $r_j$ is *unfair* if it depends solely on $j$'s data and not on others', and *untruthful* if $j$ has full information (eye) to optimize $r_j$. Definitions of (**F**) collaborative fairness and (**T**) truthfulness are in Secs. 2 and 3, respectively. We provably achieve this goal in Sec. 4 by considering a DVF with information unknown to $j$ and data values from subsets of $N$.

truthfulness in the FL setting as future work. We compare only with works that value data (to decide reward values) (Sim et al., 2020), instead of works that focus the FL setting (Xu et al., 2021a; Lin et al., 2023; Wu et al., 2024).

### B.2. Problem with Existing Methods

2 **sources.** The reward based on Chen et al. (2020) will always be equal regardless of what $\mathcal{D}$ and $D$ are used because of the following mathematical relationship:

$$
\begin{aligned}
r_i &= \log p(\mathcal{D}_j = D_j \mid \mathcal{D}_i = D_i) - \log p(\mathcal{D}_j = D_j) \\
&= \log \frac{p(\mathcal{D}_j = D_j, \mathcal{D}_i = D_i)}{p(\mathcal{D}_j = D_j)p(\mathcal{D}_i = D_i)} \\
&= \log p(\mathcal{D}_i = D_i \mid \mathcal{D}_j = D_j) - \log p(\mathcal{D}_i = D_i) \\
&= r_j.
\end{aligned}
$$

We consider the GP Friedman dataset where source 0 and 1 have 400 and 300 data points, respectively. Our method solves the problem as source 0 fairly has a Shapley value of 0.098 that is larger than source 1's 0.066.

3 **sources, source** 1 **with more unique contribution.** Zheng et al. (2024) reward source $i$ with $v(D_i)$ instead of the fair semivalue $\phi_i[\nu_{\mathcal{D}}^v]$.

We consider the GP Friedman dataset where source 0 has 400 data points, source 1 has another 300 unique data points while source 2 has 250 data points that are a subset of $D_0$. We observe that based on Zheng et al. (2024), source 2 gets slightly higher valuation with $v(D_2) = .1346$ vs $v(D_1) = 0.1306$ despite having less unique data and contributing a less improvement after observing source 0's data. However, source 2 gets a slightly lower Shapley value $\phi_2[\nu_{\mathcal{D}}^v] = 0.042$ vs $\phi_1[\nu_{\mathcal{D}}^v] = 0.049$, thus solving the problem. This is because source 2 contributes less to improving predictions when source 0 is present, e.g., $v(D_{0,2}) = 0.1469$ vs $v(D_{0,1}) = 0.1647$.

## B.3. Semivalues

When collaborative ML is modeled as a cooperative game with $n$ players and a characteristic function $\nu$ that maps coalition $C$ to its value, the semivalues, characterized by their weighting coefficients $(w_c)_{c=0}^{n-1}$, are defined as

$$\phi_i[\nu] = \sum_{C \subseteq N \setminus \{i\}} w_{|C|}[\nu(C \cup \{i\}) - \nu(C)] \tag{1}$$

where the weights are non-negative and $\sum_{c=0}^{n-1} w_c \binom{n-1}{c} = 1$.

We define $\nu \succeq_i \nu'$ if $i$'s contribution to every coalition using $\nu$ is as large as that when using another characteristic function $\nu'$, i.e., $\forall C \subseteq N \setminus \{i\}$, $\nu(C \cup \{i\}) - \nu(C) \geq \nu'(C \cup \{i\}) - \nu'(C)$. We also define $i \succeq j$ if the value of the coalition with $i$ is never less than with player $j$, i.e., $\forall C \subseteq N \setminus \{i, j\}$, $\nu(C \cup \{i\}) \geq \nu(C \cup \{j\})$. Semivalues satisfy the following desirable properties.

- **Dummy.** A dummy player $i$ always contributes the same value to every coalition, i.e., for any coalition $C \subseteq N \setminus \{i\}$, $\nu(C \cup \{i\}) - \nu(C) = \nu(\{i\})$. If player $i \in N$ is a dummy, then $\phi_i[\nu] = \nu(\{i\})$. For example, a null player, whose contribution to any coalition is $0$, should always get $0$ value.
- **Symmetry.** If the value of every coalition remains unchanged when $i$ is replaced with $j$ (i.e., $(i \succeq j) \wedge (j \succeq i)$), then the semivalues of players $i$ and $j$ are equal, i.e., $\phi_i[\nu] = \phi_j[\nu]$.
- **Monotonicity.** If $\nu \succeq_i \nu'$, then the semivalue of player $i$ based on $\nu$ should be at least the semivalue based on $\nu'$, i.e., $\phi_i[\nu] \geq \phi_i[\nu']$.
- **Desirability.** If $i \succeq j$, then the semivalue of player $i$ is at least that of player $j$, i.e., $\phi_i[\nu] \geq \phi_j[\nu]$.

Additionally, when $w_c > 0$ for all $c$, the semivalues also satisfy:

- **Strict Monotonicity.** If $\nu \succeq_i \nu'$ and $\nu \npreceq_i \nu'$, then the semivalue of player $i$ based on $\nu$ should be greater than the semivalue based on $\nu'$, i.e., $\phi_i[\nu] > \phi_i[\nu']$.
- **Strict Desirability.** If player $i$ adds more value to some coalition $C' \subseteq N \setminus \{i, j\}$ than $j$ (i.e., $\nu(C' \cup \{i\}) > \nu(C' \cup \{j\})$) and at least the same value to other coalitions $C \subseteq N \setminus \{i, j\}$, it follows that $(i \succeq j) \wedge (j \nsucceq i)$. In such a scenario, player $i$ will have a strictly larger semivalue than player $j$, i.e., $\phi_i[\nu] > \phi_j[\nu]$.

**Example.** Consider the following characteristic functions in Tab. 2. The players $i$ and $j$ are *symmetric* based on $\nu$. Player $j$ deserves a higher reward value under $\nu$ than $\nu'$ due to *strict monotonicity* as we observe that $\nu \succeq_j \nu'$ (since $\nu(C) = \nu'(C)$ when $C \not\ni j$ and $\nu(C) > \nu'(C)$ when $C \ni j$). However, player $i$ does not get a strictly higher reward value under $\nu$ than $\nu'$ as $\nu \succeq_i \nu'$ and $\nu \preceq_i \nu'$ (for $C = \emptyset, \{j\}, \{k\}, \{j, k\}$, $\nu(C \cup \{i\}) - \nu(C) = \nu'(C \cup \{i\}) - \nu'(C)$). Under $\nu'$, player $i \succeq j$, thus by desirability, $i$ should get a higher reward value than $j$.

*Table 2.* The value of the different coalitions ($\rightarrow$) under different characteristic functions ($\downarrow$) with 3 players.

|  | $\{i\}$ | $\{j\}$ | $\{k\}$ | $\{i, j\}$ | $\{i, k\}$ | $\{j, k\}$ | $\{i, j, k\}$ |
|---|---|---|---|---|---|---|---|
| $\nu$ | $\nu(\{i\}) = 3$ | 3 | 1 | 5 | 4 | 4 | 6 |
| $\nu'$ | $\nu'(\{i\}) = 3$ | 2 | 1 | 4 | 4 | 3 | 5 |

These properties align with our intuition of **(F)** collaborative fairness. The dummy property ensures that a null player (freerider) has no value. The symmetry property ensures that two players with equal values are treated equally. However, while two players may have equal value when alone $\nu(\{i\}) = \nu(\{j\})$, $i$ may still contribute more when others are present (e.g., in $\nu(N)$), thus, it is important to consider other coalitions. Sim et al. (2020) have also highlighted these desirable fairness properties for data valuation and collaborative machine learning.

*Remark on desirability.* The symmetry and (strict) monotonicity properties imply (strict) desirability. The former ensures $\phi_i[\nu] = \phi_j[\nu]$ when $i$ and $j$ are symmetric while the latter ensures that $\phi_j[\nu] > \phi_j[\nu']$ when $j$'s contribution decreases to smaller than $i$ in $\nu'$. Thus, $\phi_i[\nu'] > \phi_j[\nu']$. The monotonicity and desirability properties are also referred to as sensitivity and dominance property in the Cooperative Game Theory literature (Domenech et al., 2016).

## B.4. Bayesian Models

When applying a Bayesian learning algorithm $\mathcal{A}$, we encode our prior belief about the likely values of the model parameters $\theta$ through a distribution $p(\theta)$. After observing the dataset $D_i$, the output $\mathcal{A}(D_i)$ is the posterior distribution $p(\theta|D_i)$, which

is computed using Bayes' rule:

$$p(\theta|D_i) = \frac{p(D_i|\theta)p(\theta)}{p(D_i)}.$$

Intuitively, the posterior becomes more concentrated around parameter values $\theta$ that have a higher likelihood $p(D_i|\theta)$ of generating the dataset $D_i$.

For some Bayesian models, the posterior $p(\theta|D_i)$ can be computed using *sufficient statistics* instead of the dataset $D_i$. We define statistics and sufficient statistics as follows:

**Definition B.1** (Statistic (Singh & Koutra, 2015)). A statistic $g(D_i)$ is a function of only the data $D_i$. The statistic $g(D_i)$ and the model parameters $\theta$ are conditionally independent given $D_i$, i.e., $p(g(D_i)|D_i, \theta) = p(g(D_i)|D_i)$.

**Definition B.2** (Sufficient Statistic (SS) (Singh & Koutra, 2015; Titsias, 2009)). The statistic $g(D_i)$ is a SS for the dataset $D_i$ if the model parameters $\theta$ and dataset $D_i$ are conditionally independent given $g(D_i)$, i.e., $p(\theta|g(D_i), D_i) = p(\theta|g(D_i))$.

**Definition B.3** (Sufficiency Principle (Casella & Berger, 2021)). The statistic $g(D_i)$ is a sufficient statistic for the dataset $D_i$ if for any datasets $D_i, D_j$ such that $g(D_i) = g(D_j)$, the inference about $\theta$ should be the same regardless of whether $D_i$ or $D_j$ is observed.

Intuitively, the SS definition enforces that knowing the dataset $D_i$ should not provide extra information about $\theta$ beyond $g(D_i)$. The posterior $\mathcal{A}(D_i) \neq \mathcal{A}(\bar{D}_i)$ if and only if the datasets have different sufficient statistics.

We briefly discuss a few Bayesian models and methods for evaluating them.

### B.4.1. EXPONENTIAL FAMILY

**Definition B.4** (Exponential Family (Murphy, 2022)). A likelihood function $p(d|\theta)$ belongs to the *exponential family* if it is of the form

$$p(d|\theta) = h(d) \exp\left[\xi(\theta)^\top g(d) - A(\xi(\theta))\right],$$

where $\xi(\theta)$ is the vector of *natural parameters*, and $A(\xi(\theta))$ is the log partition function ensuring that $\sum_d p(d|\theta) = 1$.

The univariate Gaussian (with mean $\mu$ and variance $\sigma^2$) belongs to the exponential family, as we can define the sufficient statistics $g(x) = (x, x^2)^\top$ in

$$p(x|\mu, \sigma^2) = \frac{1}{\sqrt{2\pi\sigma^2}} \exp\left[\frac{-1}{2\sigma^2}x^2 + \frac{\mu}{\sigma^2}x - \frac{\mu^2}{2\sigma^2}\right].$$

When the data $D = \{d_l\}_{l=1}^c$ are drawn independently from the same distribution, the joint probability $p(D|\theta)$ of observing $\{d_l\}_{l=1}^c$ can be expressed as:

$$\left(\prod_{l=1}^c h(d_l)\right) \exp\left[\xi(\theta)^\top \sum_{l=1}^c g(d_l) - cA(\xi(\theta))\right].$$

The sufficient statistic $g(D)$ for the dataset $D$ is the sum of the sufficient statistics of individual data points, i.e., $g(D) = \sum_{l=1}^c g(d_l)$.

If the posterior $p(\theta|D)$ and the prior $p(\theta)$ belong to the same distribution family, they are *conjugate distributions*, and $p(\theta)$ is the conjugate prior for the likelihood function.

**Definition B.5** (Conjugate Prior (Murphy, 2022)). Let the likelihood $p(D|\theta)$ belong to the exponential family with natural parameters $\xi(\theta)$ and log partition function $A(\xi(\theta))$. The (natural) conjugate prior has the form

$$p(\theta|\nu_0, \varsigma_0) \propto \exp\left[\xi(\theta)^\top (\nu_0 \varsigma_0) - \nu_0 A(\xi(\theta))\right] = \exp\left[\eta^\top \zeta(\theta)\right].$$

The prior's natural parameters $\eta \triangleq \left[\nu_0, \varsigma_0^\top\right]^\top$ consist of the size $\nu_0$ and the mean sufficient statistics $\varsigma_0$ of the prior pseudo-data. The sufficient statistic $\zeta(\theta)$ is $\left[\xi(\theta)^\top, -A(\xi(\theta))\right]^\top$.

The main advantage of using conjugate priors is that they allow for closed-form computation of the posterior. Given the sufficient statistics $g(D)$ for $D$ with $c$ data points, the posterior natural parameters are updated as:

$$p(\theta|D) = p\left(\theta|\nu_0 + c, \frac{\nu_0 \varsigma_0 + g(D)}{\nu_0 + c}\right). \tag{2}$$

### B.4.2. BAYESIAN LINEAR REGRESSION

In linear regression, each datum consists of the input $\mathbf{x} \in \mathbb{R}^w$ and the output variable $y \in \mathbb{R}$. Let $D$ denote the dataset with $c$ data points, and let $\mathbf{y}$ and $\mathbf{X}$ denote the concatenated output vector and design matrix, respectively, in $\mathbb{R}^{c \times w}$. Bayesian linear regression models the relationship as $\mathbf{y} = \mathbf{X}\mathbf{w} + \mathcal{N}(0, \sigma^2 \mathbf{I})$ where the model parameters $\theta$ consist of the weight parameters $\mathbf{w} \in \mathbb{R}^w$ and the noise variance $\sigma^2$. The likelihood is given by:

$$p(\mathbf{y}|\mathbf{X}, \mathbf{w}, \sigma^2) = (2\pi\sigma^2)^{-\frac{c}{2}} \exp\left[\frac{-1}{2\sigma^2}\mathbf{y}^\top\mathbf{y} + \frac{1}{\sigma^2}\mathbf{w}^\top\mathbf{X}^\top\mathbf{y} - \frac{1}{2\sigma^2}\mathbf{w}^\top\mathbf{X}^\top\mathbf{X}\mathbf{w}\right].$$

This likelihood depends on the data through the sufficient statistics $g(D) = (\mathbf{y}^\top\mathbf{y}, \mathbf{X}^\top\mathbf{y}, \mathbf{X}^\top\mathbf{X})$. Bayesian linear regression can also be written in exponential family form.

### B.4.3. GAUSSIAN PROCESSES

A Gaussian process (GP) is a collection of random variables $\{f(\mathbf{x})\}_{\mathbf{x} \in \mathfrak{X}}$ such that any finite subset follows a multivariate Gaussian distribution (Rasmussen & Williams, 2006). A GP is fully specified by its prior mean function $\mathbb{E}[f(\mathbf{x})]$ and its covariance function (kernel) $k_{\mathbf{x}\mathbf{x}'} \triangleq \text{cov}[f(\mathbf{x}), f(\mathbf{x}')]$ for all $\mathbf{x}, \mathbf{x}' \in \mathfrak{X}$.

We assume each observation $y(\mathbf{x}_i)$ is the function $f$ evaluated at input $\mathbf{x}_i$ corrupted by zero-mean Gaussian noise $\epsilon_i$ with variance $\sigma^2$, i.e., $y(\mathbf{x}_i) = f(\mathbf{x}_i) + \epsilon_i$. Given a training set of observations $D = \{(\mathbf{x}_1, y(\mathbf{x}_1)), \ldots, (\mathbf{x}_t, y(\mathbf{x}_t))\}$, the posterior belief about the function $f$ evaluated at new inputs $\mathbf{X}^*$ is given by the posterior mean vector and covariance matrix:

$$\mu_{\mathbf{X}^*|D} = \mathbf{K}_{D\mathbf{X}^*}^\top (\mathbf{K}_D + \sigma^2\mathbf{I})^{-1}\mathbf{y},$$
$$\Sigma_{\mathbf{X}^*|D} = \mathbf{K}_{\mathbf{X}^*\mathbf{X}^*} - \mathbf{K}_{D\mathbf{X}^*}^\top (\mathbf{K}_D + \sigma^2\mathbf{I})^{-1}\mathbf{K}_{D\mathbf{X}^*}.$$

The matrices $\mathbf{K}_{D\mathbf{X}^*}$ and $\mathbf{K}_{\mathbf{X}^*\mathbf{X}^*}$ store the pairwise similarities between elements in the training set $D$ and the validation set $\mathbf{X}^*$, and within the validation set $\mathbf{X}^*$, respectively.

### B.4.4. MORE COMPLEX BAYESIAN MODELS

Bayesian neural networks extend neural networks by placing distributions over their weights (Jospin et al., 2022). Gaussian Processes with deep kernels (Wilson et al., 2016) extend Gaussian Processes by leveraging deep neural networks to learn flexible feature representations.

### B.4.5. EVALUATION METRICS

**Negative Log Predictive Density (NLPD).** The NLPD on a validation dataset $T^*$ with inputs $\mathbf{X}^*$ and outputs $\mathbf{y}^*$ given the dataset $D_i$ is:

$$\text{NLPD} \triangleq -\log p(\mathcal{Y} = \mathbf{y}^*|\mathbf{X}^*, D_i).$$

The NLPD loss decreases when the model's posterior predictive distribution $p(\mathcal{Y}|\mathbf{X}^*, D_i)$ assigns higher probability to the observed outputs $\mathbf{y}^*$, indicating that the predictive mean $\mathbb{E}[\mathcal{Y}|\mathbf{X}^*, \mathcal{D}_i = D_i]$ is close to $\mathbf{y}^*$ (accurate predictions) and that predictive uncertainty is well-calibrated (i.e., the model is more certain about accurate predictions). The NLPD loss increases if a model is uncertain about accurate predictions or overconfident about inaccurate ones.

**Mean Negative Log Probability (MNLP).** The MNLP on a validation dataset $T^*$ given the dataset $D_i$ is:

$$\text{MNLP} \triangleq \frac{1}{|T^*|} \sum_{(\mathbf{x}^*, y^*) \in T^*} -\log p(y^*|\mathbf{x}^*, D_i).$$

The MNLP loss increases if a model is uncertain about accurate predictions or overconfident about inaccurate ones, making it a useful metric for capturing both prediction accuracy and uncertainty.

Both NLPD and MNLP are widely used for evaluating Bayesian models. Adding noise or withholding data untruthfully may lead to more inaccurate predictions and duplicating data can cause overconfident predictions, increasing both losses.

**Regression (Gaussian Case).**   When the predictive distribution $p(\mathbf{y}^*|\mathbf{X}^*, D_i)$ is multivariate Gaussian with predictive mean vector $\widehat{\boldsymbol{\mu}}(\mathbf{X}^*)$ and predictive covariance matrix $\widehat{\boldsymbol{\Sigma}}(\mathbf{X}^*)$, the NLPD becomes:

$$\text{NLPD} \triangleq \frac{1}{2} \left( \log \det(2\pi\widehat{\boldsymbol{\Sigma}}(\mathbf{X}^*)) + (\mathbf{y}^* - \widehat{\boldsymbol{\mu}}(\mathbf{X}^*))^\top \widehat{\boldsymbol{\Sigma}}^{-1}(\mathbf{X}^*)(\mathbf{y}^* - \widehat{\boldsymbol{\mu}}(\mathbf{X}^*)) \right).$$

The log-determinant term penalizes models that predict high uncertainty, while the Mahalanobis distance penalizes inaccurate predictions, particularly when the predictive covariance is small (i.e., overconfident predictions).

Similarly for MNLP which considers each validation data point $(\mathbf{x}^*, y^*)$ separately, the predictive distribution $p(y^*|\mathbf{x}^*, D_i)$ is Gaussian with predictive mean $\widehat{\mu}(\mathbf{x}^*)$ and variance $\widehat{\sigma^2}(\mathbf{x}^*)$. The MNLP becomes:

$$\text{MNLP} \triangleq \frac{1}{|T^*|} \sum_{(\mathbf{x}^*, y^*) \in T^*} \frac{1}{2} \left( \log(2\pi\widehat{\sigma^2}(\mathbf{x}^*)) + \frac{(\widehat{\mu}(\mathbf{x}^*) - y^*)^2}{\widehat{\sigma^2}(\mathbf{x}^*)} \right).$$

The first term penalizes large predictive variance (underconfidence), and the second term penalizes inaccurate predictions more when the predictive variance is small (overconfidence).

**Binary Classification.**   For binary classification, the predictive probability $p(y^*|\mathbf{x}^*, D_i)$ is typically estimated using Monte Carlo samples from $p(\theta|D_i)$. Specifically, $p(y = 1|\mathbf{x}^*, D_i) \approx \frac{1}{m} \sum_{j=1}^{m} \text{sigmoid}(\theta_j^\top \mathbf{x}^*)$ , where $\theta_j$ are the Monte Carlo samples. In this case, the MNLP reduces to the cross-entropy loss.

Both MNLP and NLPD penalize overconfidence in inaccurate predictions and underconfidence in accurate ones, encouraging models to balance predictive accuracy and uncertainty. Thus, these metrics also discourage dataset inflation (e.g., duplication) or withholding/corruption of data.

## B.5. Proper Scoring Rules

Let $x'$ denote the input and $y'$ the corresponding output. In Def. B.6, we consider discriminative models and assume that $x'$ is given. We can also consider generative models by replacing the conditional distributions $p(y|x')$ and $q(y|x')$ in Def. B.6 by the joint distributions $p(x', y')$ and $q(x', y')$.

**Definition B.6** (Strictly Proper Scoring Rule).  Let $p(x', y')$ denote the true (unknown) joint distribution of observing $(x', y')$. Given $x'$, a model or source makes a probabilistic prediction $q(y|x')$, representing a distribution over possible outputs $y$ conditioned on $x'$.

When $(x', y')$ is observed, a scoring rule $V(q(y|x'); (x', y'))$ assigns a score to the predictive distribution $q(y|x')$, based on how well it agrees with the actual output $y'$. The scoring rule rewards predictions that are both accurate and well-calibrated with respect to the true conditional distribution $p(y|x')$.

The expected score of the prediction $q$ under the true joint distribution $p$ is defined as:

$$V_{\mathbb{E}}(q; p) := \int p(x') \int p(y'|x') V(q(y|x'); (x', y')) \, dy' \, dx',$$

which represents the average score obtained by the model's prediction $q$ over all possible realizations of $(x', y')$ where $y'$ is drawn from $p(y'|x')$.

A scoring rule $V$ is said to be *proper* if the expected score is maximized when the predictive distribution $q(y|x')$ equals the true conditional distribution $p(y|x')$. Formally, for any predictive distribution $q(y|x')$, we require:

$$V_{\mathbb{E}}(q; p) \leq V_{\mathbb{E}}(p; p),$$

with equality if and only if $q(y|x') = p(y|x')$. If this equality holds *only* when $q(y|x') = p(y|x')$, then the scoring rule is said to be *strictly proper*.

Strictly proper scoring rules are widely used in machine learning for models that output probabilistic predictions. These rules ensure that the model's predictive probabilities are consistent with the true underlying distribution. Examples of strictly proper scoring rules include:

- The **logarithmic scoring rule**, which corresponds to the cross-entropy loss used in classification tasks, is given by:

$$V(q(y|x'); (x', y')) = \log q(y'|x'),$$

and it is maximized when $q(y|x') = p(y|x')$.

- The **Brier score**, another strictly proper scoring rule, measures the squared error between the predictive probabilities and the true outputs. In the case of classification where $y'$ is represented as a one-hot encoded vector, the Brier score is:

$$V(q(y|x'); (x', y')) = \sum_k \left(q_k(y|x') - \mathbb{I}(y' = k)\right)^2,$$

where $q_k(y|x')$ denotes the predicted probability of class $k$ and $\mathbb{I}(y' = k)$ is the indicator function.

**Classification accuracy** is *not* a strictly proper scoring rule because accuracy only evaluates the predicted label, ignoring how well-calibrated the predictive probabilities are. As a result, probabilistic predictions can be perturbed without affecting the accuracy score, as long as the predicted labels remain the same.

### B.6. Incentivizing Truthful Data Submission

Faltings (2022) gives an overview of game-theoretic mechanisms for eliciting accurate information. In this work, we focus on mechanisms to elicit true data from sources jointly training an ML model for a common prediction task with a shared ground truth (e.g., heart disease in a population) [5]. A mechanism is *truthful* if sources expect the highest reward by reporting the data they believe are correct. Peer consistency mechanisms (Miller et al., 2005; Jurca & Faltings, 2011; Faltings et al., 2017) evaluate a source based on the agreement between its submission and those of randomly selected peers. Dorner et al. (2023) incentivizes truthfulness at the Nash equilibrium by penalizing disagreement between a source's reported and its peers' model parameters in mean estimation and stochastic gradient descent tasks. Chen et al. (2020) values a source $i$ by the change in log-likelihood of others' data after a Bayesian model is updated by source $i$'s data. However, these peer consistency mechanisms may fail to ensure (**F**) as rewards are not based on the same DVFs and all sources' data values.

As an alternative, Richardson et al. (2020); Zheng et al. (2024) assume access to a ground truth validation set. Richardson et al. (2020) allows for multiple models and loss functions but assumes that all data points are drawn from the true distribution and that more data cannot worsen test loss. Zheng et al. (2024) also uses Bayesian models and pointwise mutual information between each source's data and the mediator's held-out validation set.

## C. Data Valuation Functions

### C.1. Strategies to increase valuation for untruthful data valuation functions

In this section, we will examine how sources can untruthfully increase (or maintain) their valuation when using the data valuation functions from previous works. We will first examine data valuation functions that do not use a validation set.

**Cardinality**  Let $v(D_i) = |D_i|$. Data source $i$ can increase its valuation by untruthfully submitting more data points. For example, duplicating every data point in $\bar{D}_i$ to form $D_i$ would artificially increase the valuation.

**Divergence from the prior**  Consider exponential family models where each dataset $D_i$ has sufficient statistics $s(D_i)$. Sim et al. (2023) propose to value a dataset by the divergence of the posterior model parameters from the prior, i.e., $v(D_i) = \mathbb{D}_{KL}(p(\theta|D_i); p(\theta))$. In their Appendix E.2, Sim et al. (2023) have shown that submitting the dataset with large sufficient statistics (i.e., $\kappa s(\bar{D}_i)$ where $\kappa > 1$) would increase the valuation (i.e., $v(D_i) > v(\bar{D}_i)$).

*Proof.* The KL divergence between two members of the same exponential family with natural parameters $\eta$ and $\eta'$, and log partition function $B(\cdot)$ is given by $(\eta - \eta')^\top \nabla B(\eta) - B(\eta) + B(\eta')$. Let source $i$'s corresponding data sufficient statistics and cardinality be $\mathbf{s}_i$ and $c_i$, respectively. Let $\eta'$ and $\eta$ be the natural parameters of the prior and the power posterior distribution (used to generate a model reward with value $r_i$), respectively. Then, $\eta = \eta' + \kappa_i \left[\mathbf{s}_i^\top, c_i\right]^\top$. For

---

[5]There is a single task and objective ground truth. Thus, mechanisms that need statistics taken over multiple tasks are unsuitable.

$\kappa_i \geq 0$, the derivative of valuation/reward $r_i$ w.r.t. $\kappa_i$ is non-negative. Thus increasing $\kappa_i$ would increase the reward:

$$\frac{\mathbf{d}r_i}{\mathbf{d}\kappa_i} = \frac{\partial r_i}{\partial \eta} \frac{\partial \eta}{\partial \kappa_i}$$

$$= \left((\eta - \eta')^\top \nabla^2 B(\eta) + \nabla B(\eta) - \nabla B(\eta)\right) \left[\mathbf{s}_i^\top, \ c_i\right]^\top$$

$$= \left[\kappa_i \mathbf{s}_i^\top, \ \kappa_i c_i\right] \nabla^2 B(\eta) \left[\mathbf{s}_i^\top, \ c_i\right]^\top = \kappa_i \left[\mathbf{s}_i^\top, \ c_i\right] \nabla^2 B(\eta) \left[\mathbf{s}_i^\top, \ c_i\right]^\top \geq 0 \,.$$

The last inequality is because the second derivative is positive semi-definite.

Furthermore, the source does not need to submit a dataset that scales the true sufficient statistics. Any dataset that results in posterior model parameters significantly diverging from the prior (e.g., concatenate multiple perturbed copies of $D_i$) can untruthfully increase the valuation.

**Information gain** Sim et al. (2020) propose to value data based on the information gain on Bayesian model parameters $\theta$ or the Gaussian process function given the data. Formally, $v(D_i) = \mathbb{I}(\theta; D_i) = .5 \log \det(\mathbf{I} + \mathbf{K}_{D_i}/\sigma^2)$ where $\mathbf{K}_{D_i}$ is the kernel matrix that records the pairwise similarity between all points in $D_i$. The data source can untruthfully increase its valuation by duplicating data or arbitrarily adding more data.

*Proof* Let $D_i' \subset D_i$. The information gain is the reduction in entropy of the model parameters $\theta$, $\mathbb{I}(\theta; D_i) = \mathbb{H}(\theta) - \mathbb{H}(\theta|D_i)$. The difference $\mathbb{I}(\theta; D_i) - \mathbb{I}(\theta; D_i') = \mathbb{H}(\theta|D_i') - \mathbb{H}(\theta|D_i) \geq 0$ due to the 'information never hurts" property of entropy (Cover & Thomas, 1991). Thus, adding data would increase the information gain.

Note that it is not straightforward to compute the information gain for other models, such as Bayesian logistic regression. However, the information gain would involve taking expectation over the data and may hence not incentivize strict truthfulness.

**Volume** Each source $i$'s dataset $D_i$ consists of an input matrix $\mathbf{X}_i$ (with size $|D_i| \times f$ where $f$ is the number of features) and an output vector. Xu et al. (2021b) propose to value data based on the volume (i.e., square root of the determinant) of the left Gram matrix: $v(S) = \sqrt{|\mathbf{X}_i^\top \mathbf{X}_i|}$. Since the volume of a dataset does not decrease with the addition of data points, source $i$ can untruthfully increase its valuation by arbitrarily adding more diverse data. For example, duplicating each data point $k$ times would increase $v(S)$ by a factor of $\sqrt{k}$.

Moreover, the data valuation functions based on information gain and volume depend solely on the input matrix and not on the observed output values (such as housing prices). As a result, source $i$ can untruthfully submit incorrect output values in $D_i$, increasing its privacy while reducing the model performance for others.

These examples demonstrate Goodhart's law: "When a measure becomes a target, it ceases to be a good measure". When data sources can fully observe the valuation function $v$, it becomes a target, and sources will untruthfully optimize their datasets to maximize their values. In Sec. 3, we avoid Goodhart's law by ensuring the data valuation function depends on some random variables $\mathcal{O}$ unknown to source $i$ (e.g., a held-out validation set or datasets from others).

## C.2. Proofs that DVFs ensure truthfulness

### C.2.1. PROOF OF PROP. 3.2

**Part I.** The value of source $i$'s dataset $v(D_i) \triangleq \log p(\mathcal{T} = T^* | \mathcal{D}_i = D_i) - \log p(\mathcal{T} = T^*)$ is $i$-s·truthful.

*Proof.* Source $i$ does not know the held-out validation set $T^*$. Given assumptions A.2 and A.3, source $i$ will consider possible values $T$ based on its true dataset $\bar{D}_i$, i.e., the distribution $p(\mathcal{T} = T | \mathcal{D}_i = \bar{D}_i)$. We abbreviate $p(\mathcal{T} = T | \mathcal{D}_i = \bar{D}_i)$ as $p(\mathcal{T} = T | \bar{D}_i)$. The difference in expectation from submitting $\bar{D}_i$ instead of $D_i$ is

$$\mathbb{E}_{\mathcal{T} | \bar{D}_i}[v(\bar{D}_i)] - \mathbb{E}_{\mathcal{T} | \bar{D}_i}[v(D_i)]$$

$$= \int_T p(\mathcal{T} = T | \bar{D}_i) \log p(\mathcal{T} = T | \bar{D}_i) \ \mathrm{d}T - \int_T p(\mathcal{T} = T | \bar{D}_i) \log p(\mathcal{T} = T | D_i) \ \mathrm{d}T$$

$$= \mathrm{KL}\left[p(\mathcal{T} = T | \bar{D}_i) \,\|\, p(\mathcal{T} = T | D_i)\right] \geq 0.$$

Since the Kullback-Leibler (KL) divergence is always non-negative, we conclude that

$$\mathbb{E}_{\mathcal{T} | \bar{D}_i}[v(\bar{D}_i)] \geq \mathbb{E}_{\mathcal{T} | \bar{D}_i}[v(D_i)] \,.$$

Moreover, the KL divergence is positive whenever $p(\mathcal{T} = T|D_i) \neq p(\mathcal{T} = T|\bar{D}_i)$. Thus, any dataset $D_i$ leading to different inferences about the model parameters (i.e., with different sufficient statistics) would have strictly lower valuation in expectation, i.e., $\mathbb{E}_{\mathcal{T}|\bar{D}_i}[v(\bar{D}_i)] > \mathbb{E}_{\mathcal{T}|\bar{D}_i}[v(D_i)]$. □

An alternative proof makes use of the result that the logarithmic scoring rule is strictly proper. The definitions of scoring rules are given in Sec. B.5 while the proof is elaborated in Sec. C.2.2.

**Part II.** For any $i$ in coalition $C$, the value of the coalition $C$'s dataset $v(D_C) \triangleq \log p(\mathcal{T} = T^*|\mathcal{D}_C = D_C) - \log p(\mathcal{T} = T^*)$ is $i$-s·truthful.

*Proof.* As source $i$ does not know the held-out validation set $T^*$ or the datasets of others $C \setminus \{i\} = C \setminus \{i\}$, source $i$ should take expectations over $\mathcal{T}$ and $\mathcal{D}_{C\setminus\{i\}}$. Given assumptions A.2-A.4, source $i$ will consider the possible values based on the distribution $p(\mathcal{T} = T, \mathcal{D}_{C\setminus\{i\}} = D_{C\setminus\{i\}}|\mathcal{D}_i = \bar{D}_i)$. We abbreviate $p(\mathcal{T} = T|\mathcal{D}_i = D_i, \mathcal{D}_{C\setminus\{i\}} = D_{C\setminus\{i\}})$ as $p(\mathcal{T} = T|D_i, D_{C\setminus\{i\}})$.

Now, the difference in expectation from submitting $\bar{D}_i$ instead of $D_i$ is

$$
\mathbb{E}_{(\mathcal{T},\mathcal{D}_{C\setminus\{i\}})|\bar{D}_i}\left[v(\bar{D}_i \cup D_{C\setminus\{i\}}))\right] - \mathbb{E}_{(\mathcal{T},\mathcal{D}_{C\setminus\{i\}})|\bar{D}_i}\left[v(D_i \cup D_{C\setminus\{i\}}))\right]
$$

$$
= \int_{D_{C\setminus\{i\}}} \int_T p(\mathcal{T} = T, \mathcal{D}_{C\setminus\{i\}} = D_{C\setminus\{i\}}|\bar{D}_i) \log p(\mathcal{T} = T|\bar{D}_i, D_{C\setminus\{i\}}) \; \mathrm{d}T \; \mathrm{d}D_{C\setminus\{i\}}
$$

$$
\quad - \int_{D_{C\setminus\{i\}}} \int_T p(\mathcal{T} = T, \mathcal{D}_{C\setminus\{i\}} = D_{C\setminus\{i\}}|\bar{D}_i) \log p(\mathcal{T} = T|D_i, D_{C\setminus\{i\}}) \; \mathrm{d}T \; \mathrm{d}D_{C\setminus\{i\}}
$$

$$
= \int_{D_{C\setminus\{i\}}} p(\mathcal{D}_{C\setminus\{i\}} = D_{C\setminus\{i\}}|\bar{D}_i) \int_T p(\mathcal{T} = T|\bar{D}_i, D_{C\setminus\{i\}}) \log \frac{p(\mathcal{T} = T|\bar{D}_i, D_{C\setminus\{i\}})}{p(\mathcal{T} = T|D_i, D_{C\setminus\{i\}})} \; \mathrm{d}T \; \mathrm{d}D_{C\setminus\{i\}}
$$

$$
= \int_{D_{C\setminus\{i\}}} p(\mathcal{D}_{C\setminus\{i\}} = D_{C\setminus\{i\}}|\bar{D}_i) \cdot \mathrm{KL}\left(p(\mathcal{T}|\bar{D}_i, D_{C\setminus\{i\}}) \parallel p(\mathcal{T}|D_i, D_{C\setminus\{i\}})\right) \; \mathrm{d}D_{C\setminus\{i\}}
$$

$$
\geq 0.
$$

Since the KL divergence and probabilities are always non-negative, we conclude that

$$
\mathbb{E}_{(\mathcal{T},\mathcal{D}_{C\setminus\{i\}})|\bar{D}_i}\left[v(\bar{D}_i \cup D_{C\setminus\{i\}}))\right] \geq \mathbb{E}_{(\mathcal{T},\mathcal{D}_{C\setminus\{i\}})|\bar{D}_i}\left[v(D_i \cup D_{C\setminus\{i\}}))\right] .
$$

Moreover, the KL divergence is positive whenever the distributions differ. Thus, any dataset $D_i$ leading to different inferences about the model parameters (i.e., with different sufficient statistics) would have strictly lower valuation in expectation. □

*Remark.* If source $i$ suspects that some source $j$ in $C \setminus \{i\}$ is untruthful and not submitting data that are drawn from $p(\mathcal{D}_j|\theta)$, source $i$ would not be convinced by the proof, which takes expectation over $p(\mathcal{T} = T, \mathcal{D}_{C\setminus\{i\}} = D_{C\setminus\{i\}}|\bar{D}_i)$. For example, source $i$ might attempt to counteract the errors introduced by other sources in $D_{C\setminus\{i\}}$.

### C.2.2. STRICTLY PROPER SCORING RULES

Let $\mathbf{X}^*$ denote the input matrix and $\mathbf{y}^*$ the output vector of the validation set. We consider discriminative models that treat $\mathbf{X}^*$ as known, thus the random variable $\mathcal{T}$ only corresponds to the output vector $\mathcal{Y}$.

Let $V$ be a strictly proper scoring rule. By definition, the expected value $\mathbb{E}_{\mathcal{Y}}\left[V(q_i(\mathcal{Y}|\mathbf{X}^*); \mathbf{X}^*, \mathbf{y}^*\right]$ is only maximized when $q_i(\mathcal{Y}|\mathbf{X}^*)$ matches the true posterior predictive distribution $p(\mathcal{Y} = \mathbf{y}^*|\mathbf{X}^*, \bar{D}_i)$. Hence for any constant $b$ and dataset $D_i$ that leads to different inference about model parameters and predictions,

$$
\mathbb{E}_{\mathcal{Y}}\left[V(p(\mathcal{Y} = \mathbf{y}^*|\mathbf{X}^*, D_i)); \mathbf{X}^*, \mathbf{y}^*\right] - b < \mathbb{E}_{\mathcal{Y}}\left[V(p(\mathcal{Y} = \mathbf{y}^*|\mathbf{X}^*, \bar{D}_i)); \mathbf{X}^*, \mathbf{y}^*\right] - b .
$$

Thus, the DVF

$$
v(D_i) = V(p(\mathcal{Y}|\mathbf{X}^*, D_i); \mathbf{X}^*, \mathbf{y}^*) - b.
$$

is $i$-s·truthful and satisfies Def. 3.1.

For any coalition $C$, the expected value $\mathbb{E}_{\mathcal{Y}}\left[V(q_C(\mathcal{Y}|\mathbf{X}^*); \mathbf{X}^*, \mathbf{y}^*\right]$ is also only maximized when $q_C(\mathcal{Y}|\mathbf{X}^*)$ matches the true posterior predictive distribution $p(\mathcal{Y} = \mathbf{y}^*|\mathbf{X}^*, \bigcup_{j \in C} \bar{D}_j)$. Hence for any constant $b$ and dataset $D_C$ that leads to different inference about model parameters and predictions,

$$\mathbb{E}_{\mathcal{Y}}\left[V(p(\mathcal{Y} = \mathbf{y}^*|\mathbf{X}^*, D_C); \mathbf{X}^*, \mathbf{y}^*)\right] - b < \mathbb{E}_{\mathcal{Y}}\left[V(p(\mathcal{Y} = \mathbf{y}^*|\mathbf{X}^*, \bigcup_{j \in C} \bar{D}_j)); \mathbf{X}^*, \mathbf{y}^*\right] - b \,.$$

Specifically, if every other source $j \in C \setminus \{i\}$ submits $\bar{D}_j$, source $i$ must also submit $\bar{D}_i$ to produce the same inference about model parameters and predictions. That is,

$$\mathbb{E}_{\mathcal{Y}}\left[V(p(\mathcal{Y} = \mathbf{y}^*|\mathbf{X}^*, \bigcup_{j \in C \setminus \{i\}} \bar{D}_j \cup D_i); \mathbf{X}^*, \mathbf{y}^*)\right] - b < \mathbb{E}_{\mathcal{Y}}\left[V(p(\mathcal{Y} = \mathbf{y}^*|\mathbf{X}^*, \bigcup_{j \in C} \bar{D}_j); \mathbf{X}^*, \mathbf{y}^*)\right] - b \,.$$

Hence, for any source $i \in C$, the DVF

$$v(D_C) = V(p(\mathcal{Y}|\mathbf{X}^*, D_C); \mathbf{X}^*, \mathbf{y}^*) - b$$

is $i$-s·truthful when every other source $j \in C \setminus \{i\}$ submits $\bar{D}_j$.

### C.2.3. MEAN LOG-LIKELIHOOD

Note that Prop. 3.2 also holds true when the validation set consists of any single data point $t$ represented by the random variable $\omega_t$.

Thus, when $T^*$ consists of multiple data points, setting $v(D_i) = \frac{1}{|T^*|} \sum_{t \in T^*} \log p(\omega_t = t|\mathcal{D}_i = D_i) - \log p(\omega_t = t)$ still satisfies Def. 3.1.

When considering discriminative models, this redefined $v$ inversely corresponds to the MNLP of the validation data $\sum_{t=(x^*, y^*) \in T^*} -\log p(y^*|\theta, x^*)$.

## D. Truthful Semivalues

### D.1. Proof of Prop. 4.1

*Proof.* Let $\nu^v_{\mathfrak{D}, i \to \bar{D}_i}$ denote the characteristic function computed using the datasets $D_j$ for $j \in N \setminus \{i\}$ and the true dataset $\bar{D}_i$ for $j = i$. Given

$$\phi_i[\nu^v_{\mathfrak{D}}] = \sum_{C \subseteq N \setminus \{i\}} w_{|C|} \left[v(D_C \cup D_i) - v(D_C)\right] \,,$$

we can compute the expected semivalue,

$$\begin{aligned}
\mathbb{E}_{(\mathcal{T}, \mathcal{D}_{N \setminus \{i\}})|\bar{D}_i}[\nu^v_{\mathfrak{D}, i \to \bar{D}_i}] &= \sum_{C \subseteq N \setminus \{i\}} w_{|C|} \left[\mathbb{E}_{(\mathcal{T}, \mathcal{D}_C)|\bar{D}_i}[v(D_C \cup \bar{D}_i)] - \mathbb{E}_{(\mathcal{T}, \mathcal{D}_C)|\bar{D}_i}[v(D_C)]\right] \\
&\geq \sum_{C \subseteq N \setminus \{i\}} w_{|C|} \left[\mathbb{E}_{(\mathcal{T}, \mathcal{D}_C)|\bar{D}_i}[v(D_C \cup D_i)] - \mathbb{E}_{(\mathcal{T}, \mathcal{D}_C)|\bar{D}_i}[v(D_C)]\right] \\
&= \mathbb{E}_{(\mathcal{T}, \mathcal{D}_{N \setminus \{i\}})|\bar{D}_i}[\phi_i[\nu^v_{\mathfrak{D}}]].
\end{aligned}$$

The first equality is due to the linearity of expectation. The second inequality makes use of the fact that the weights are non-negative and Prop. 3.2.

The semivalue $\phi_i[\nu^v_{\mathfrak{D}}]$ is also $i$-s·truthful as for every $C$ such that $w_{|C|} > 0$, data that results in different inference about the model parameters (i.e., $p(\mathcal{T}|\bar{D}_i, D_C) \neq p(\mathcal{T}|D_i, D_C)$), would result in a positive KL divergence and strict inequality. $\square$

### D.2. Proof of Prop. 4.2

*Proof.* Recall that the semivalue for a source $i$ is given by:

$$\phi_i[\nu^v_{\mathfrak{D}}] = \sum_{C \subseteq N \setminus \{i\}} w_{|C|} \left[v(D_i \cup D_C) - v(D_C)\right].$$

For any other source $k \neq i$, its semivalue is:

$$
\begin{aligned}
\phi_k[\nu_{\mathfrak{D}}^v] &= \sum_{C \subseteq N \setminus \{k\}} w_{|C|} \left[ v(D_k \cup D_C) - v(D_C) \right] \\
&= \sum_{C \subseteq N \setminus \{k,i\}} w_{|C|+1} \left[ v(D_k \cup D_i \cup D_C) - v(D_i \cup D_C) \right] \\
&\quad + \sum_{C \subseteq N \setminus \{k,i\}} w_{|C|} \left[ v(D_k \cup D_C) - v(D_C) \right].
\end{aligned}
$$

Now, consider source $i$ submitting a dataset $D_i$ instead of $\bar{D}_i$. The change in its semivalue, denoted by $\Delta_i^i$, is:

$$
\Delta_i^i = \sum_{C \subseteq N \setminus \{i\}} w_{|C|} \left[ v(D_i \cup D_C) - v(\bar{D}_i \cup D_C) \right].
$$

Next, we consider the change in source $k$'s semivalue

$$
\begin{aligned}
\Delta_k^i &= \sum_{C \subseteq N \setminus \{k,i\}} w_{|C|+1} \left[ v(D_i \cup D_k \cup D_C) - v(D_i \cup D_C) \right] - \sum_{C \subseteq N \setminus \{k,i\}} w_{|C|+1} \left[ v(\bar{D}_i \cup D_k \cup D_C) - v(\bar{D}_i \cup D_C) \right] \\
&= \sum_{C \subseteq N \setminus \{k,i\}} w_{|C|+1} \left[ v(D_i \cup D_{C \cup \{k\}}) - v(\bar{D}_i \cup D_{C \cup \{k\}}) \right] - \sum_{C \subseteq N \setminus \{k,i\}} w_{|C|+1} \left[ v(D_i \cup D_C) - v(\bar{D}_i \cup D_C) \right].
\end{aligned}
$$

Prop. 3.2 states that $\mathbb{E}_{(\mathcal{T}, \mathcal{D}_{N \setminus \{i\}}) | \bar{D}_i} [v(D_i \cup D_C) - v(\bar{D}_i \cup D_C)] \leq 0$ for any coalition $C$. While $\mathbb{E}_{(\mathcal{T}, \mathcal{D}_{N \setminus \{i\}}) | \bar{D}_i} [\Delta_i^i]$ is a positive weighted sum of $\mathbb{E}_{(\mathcal{T}, \mathcal{D}_{N \setminus \{i\}}) | \bar{D}_i} [v(D_i \cup D_C) - v(\bar{D}_i \cup D_C)]$, $\mathbb{E}_{(\mathcal{T}, \mathcal{D}_{N \setminus \{i\}}) | \bar{D}_i} [\Delta_k^i]$ consists of both positive and negative weights. Thus,

$$
\begin{aligned}
\mathbb{E}_{(\mathcal{T}, \mathcal{D}_{N \setminus \{i\}}) | \bar{D}_i} \left[ \phi_i[\nu_{\mathfrak{D}}^v] - \phi_i[\nu_{\mathfrak{D}, i \to \bar{D}_i}^v] \right] &= \mathbb{E}_{(\mathcal{T}, \mathcal{D}_{N \setminus \{i\}}) | \bar{D}_i} [\Delta_i^i] \\
&\leq \mathbb{E}_{(\mathcal{T}, \mathcal{D}_{N \setminus \{i\}}) | \bar{D}_i} [\Delta_k^i] \\
&= \mathbb{E}_{(\mathcal{T}, \mathcal{D}_{N \setminus \{i\}}) | \bar{D}_i} \left[ \phi_k[\nu_{\mathfrak{D}}^v] - \phi_k[\nu_{\mathfrak{D}, i \to \bar{D}_i}^v] \right] \\
\mathbb{E}_{(\mathcal{T}, \mathcal{D}_{N \setminus \{i\}}) | \bar{D}_i} \left[ \phi_i[\nu_{\mathfrak{D}, i \to \bar{D}_i}^v] - \phi_i[\nu_{\mathfrak{D}}^v] \right] &\geq \mathbb{E}_{(\mathcal{T}, \mathcal{D}_{N \setminus \{i\}}) | \bar{D}_i} \left[ \phi_k[\nu_{\mathfrak{D}, i \to \bar{D}_i}^v] - \phi_k[\nu_{\mathfrak{D}}^v] \right].
\end{aligned}
$$

$\square$

# E. Considering Other Constraints

## E.1. Weaker Truthfulness Guarantees when denominator depends on data submissions

In this section, we explore the theoretical guarantees when $a = \max_k \phi_k[\nu_{\mathfrak{D}}^v] + \gamma$.

**Proposition E.1.** *Let $B > 0$ be the limited budget per source. Let $\gamma \geq 0$ be such that $\max_k \phi_k[\nu_{\mathfrak{D}}^v] + \underline{\gamma} > 0$. We define the scaled semivalue as $r_i \triangleq B \cdot \frac{\phi_i[\nu_{\mathfrak{D}}^v]}{\max_k \phi_k[\nu_{\mathfrak{D}}^v] + \gamma}$. Then, for each source $i$, the scaled semivalue fits within the limited budget, i.e., $r_i \leq B$.*

*Under assumptions A1-A3 for any $\gamma > \underline{\gamma}$ and alternative dataset $D_i$ in $\mathfrak{D}$ with $\mathbb{E}[\phi_i[\nu_{\mathfrak{D}, i \to \bar{D}_i}^v]] > 0$,[6] we have*

$$
\frac{B \cdot \mathbb{E}[\phi_i[\nu_{\mathfrak{D}}^v]]}{\mathbb{E}[\max_k \phi_k[\nu_{\mathfrak{D}}^v]] + \gamma} \leq \frac{B \cdot \mathbb{E}[\phi_i[\nu_{\mathfrak{D}, i \to \bar{D}_i}]]}{\mathbb{E}[\max_k \phi_k[\nu_{\mathfrak{D}, i \to \bar{D}_i}]] + \gamma},
$$

*where the expectation $\mathbb{E}$ is taken over $(\mathcal{T}, \mathcal{D}_{N \setminus \{i\}}) | \bar{D}_i$. The inequality is strict when $\mathcal{A}(D_i) \neq \mathcal{A}(\bar{D}_i)$. From source $i$'s perspective, the ratio of expected values is maximized by submitting $\bar{D}_i$.*

---

[6]If the expectation is negative, source $i$ would not submit data.

The key intuition of the following proof is that, by Prop. 4.2, any untruthful submission $D_i$ is expected to reduce the numerator more than the denominator.

*Proof.* Let $B > 0$ be the limited budget per source, and let $\gamma \geq 0$ be such that $\max_k \phi_k[\nu_{\mathfrak{D}}^v] + \gamma > 0$. We define the scaled semivalue:

$$r_i \triangleq B \cdot \frac{\phi_i[\nu_{\mathfrak{D}}^v]}{\max_k \phi_k[\nu_{\mathfrak{D}}^v] + \gamma}.$$

Since $\phi_i[\nu_{\mathfrak{D}}^v] \leq \max_k \phi_k[\nu_{\mathfrak{D}}^v]$, it follows that:

$$\frac{\phi_i[\nu_{\mathfrak{D}}^v]}{\max_k \phi_k[\nu_{\mathfrak{D}}^v] + \gamma} \leq 1 \quad \text{and hence} \quad r_i \leq B.$$

Next, we make assumptions A1-A3 to use the past propositions. For any alternative dataset $D_i$ in $\mathfrak{D}$, let $\Delta_i^i = \phi_i[\nu_{\mathfrak{D}}^v] - \phi_i[\nu_{\mathfrak{D}, i \to \bar{D}_i}^v]$. Let source $j = argmax_k \phi_k[\nu_{\mathfrak{D}, i \to \bar{D}_i}^v]$ and $\Delta_j^i = \phi_j[\nu_{\mathfrak{D}}^v] - \phi_j[\nu_{\mathfrak{D}, i \to \bar{D}_i}^v]$. We denote $\Delta_{max}^i = \max_k \phi_k[\nu_{\mathfrak{D}}^v] - \max_j \phi_j[\nu_{\mathfrak{D}, i \to \bar{D}_i}^v]$.

$$\mathbb{E}_{(\mathcal{T}, \mathcal{D}_{N \setminus \{i\}}) | \bar{D}_i} \left[ \Delta_i^i \right] \leq \mathbb{E}_{(\mathcal{T}, \mathcal{D}_{N \setminus \{i\}}) | \bar{D}_i} \left[ \Delta_j^i \right] \tag{3}$$

$$\leq \mathbb{E}_{(\mathcal{T}, \mathcal{D}_{N \setminus \{i\}}) | \bar{D}_i} \left[ \max_k \phi_k[\nu_{\mathfrak{D}}^v] - \max_j \phi_j[\nu_{\mathfrak{D}, i \to \bar{D}_i}^v] \right] \tag{4}$$

$$= \mathbb{E}_{(\mathcal{T}, \mathcal{D}_{N \setminus \{i\}}) | \bar{D}_i} \left[ \Delta_{max}^i \right] .$$

Step 3 is due to Prop. 4.2. Step 4 is because $\phi_j[\nu_{\mathfrak{D}}^v] \leq \max_k \phi_k[\nu_{\mathfrak{D}}^v]$.

For ease of notation, we use $\mathbb{E}$ to abbreviate $\mathbb{E}_{(\mathcal{T}, \mathcal{D}_{N \setminus \{i\}}) | \bar{D}_i}$. We abbreviate $\mathbb{E}[\max_k \phi_k[\nu_{\mathfrak{D}, i \to \bar{D}_i}]]$ as $\bar{\varphi}_{max}$ and $\mathbb{E}[\phi_i[\nu_{\mathfrak{D}, i \to \bar{D}_i}]]$ as $\bar{\varphi}_i$.

$$\mathbb{E}[\Delta_i^i] \leq \mathbb{E}[\Delta_{max}^i]$$

$$\mathbb{E}[\Delta_i^i] \cdot (\bar{\varphi}_{max} + \gamma) \leq \bar{\varphi}_i \cdot (\mathbb{E}[\Delta_{max}^i]) \tag{5}$$

$$(\bar{\varphi}_i + \mathbb{E}[\Delta_i^i]) \cdot (\bar{\varphi}_{max} + \gamma) \leq \bar{\varphi}_i \cdot (\bar{\varphi}_{max} + \mathbb{E}[\Delta_{max}^i] + \gamma)$$

$$\frac{B \cdot \mathbb{E}[\phi_i[\nu_{\mathfrak{D}}^v]]}{\mathbb{E}[\max_k \phi_k[\nu_{\mathfrak{D}}^v]] + \gamma} \leq \frac{B \cdot \mathbb{E}[\phi_i[\nu_{\mathfrak{D}, i \to \bar{D}_i}]]}{\mathbb{E}[\max_k \phi_k[\nu_{\mathfrak{D}, i \to \bar{D}_i}]] + \gamma} . \tag{6}$$

Step 5 is because $\mathbb{E}\left[ \Delta_i^i \right] \leq 0$ (Prop. 4.1) and $\bar{\varphi}_{max} \geq \bar{\varphi}_i \geq 0$.

Step 6 assumes that $(\max_k \phi_k[\nu_{\mathfrak{D}}^v]] + \gamma), (\mathbb{E}[\max_k \phi_k[\nu_{\mathfrak{D}, i \to \bar{D}_i}]] + \gamma), B > 0$.

$\square$

## E.2. Interaction between Truthfulness and other incentives in CML

I1 **Feasibility.** The reward to each source $i$ must not exceed some value $B$. For example, $B$ might be the monetary budget available or it might be the maximum value of the model reward (Sim et al., 2020).

I2 **Efficiency.** To maximize the benefits from the collaboration, there is at least one source that gets the maximum value $B$.

I3 **Individual Rationality.** The reward to each source $i$ must exceed the value each source can get from working alone or not participating in the collaboration. For model rewards, the value of the model reward must exceed the value of the model trained only on source $i$'s data (Sim et al., 2020). For monetary reward, the value of the reward must exceed the cost which may be assumed to be 0.

I4 **Collaborative Fairness.** We define $i \succeq j$ if the value of the coalition with $i$'s data is never less than that with $j$'s, i.e., $\forall C \subseteq N \setminus \{i, j\}$, $v(D_{C \cup \{i\}}) \geq v(D_{C \cup \{j\}})$. The reward values $(r_i)_{i \in N}$ are *fair* if they reflect $i$'s desirability: (i)

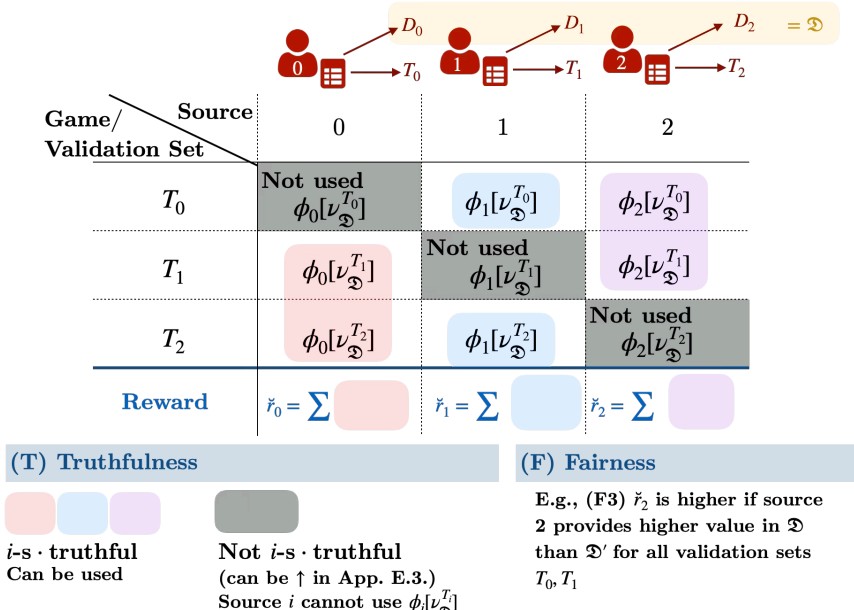

*Figure 5.* When there is no validation set, the reward for each source are computed as follows to ensure truthfulness and modified fairness.

$(i \succeq j) \implies (r_i \geq r_j)$, and (ii) for strict desirability, $(i \succeq j) \land (j \not\succeq i) \implies (r_i > r_j)$. Notably, fairness goes beyond considering whether $v(D_i) = v(D_j)$ as source $i$ may still contribute more to improving model performance (e.g., in $v(D_N)$) than $j$ if its data is less redundant and similar to others' in $N \setminus \{i, j\}$. Condition (i) also implies that two sources with identical value in all coalitions and identical datasets should get equal rewards.

I5 **Privacy.** Some sources may want differential privacy guarantees to prevent others from inferring about their data. Moreover, stronger differential privacy guarantees should lead to lower rewards (Sim et al., 2023).

I6 **No Validation Set.** When it is hard to obtain and agree on a validation set, data valuation must be done without a validation set.

The *strict* truthfulness incentive cannot be simultaneously satisfied with I2. If a source is guaranteed to receive the maximum reward $B$, the reward value does not depend on the level of its contribution. For example, if the source contributing the most informative dataset will get the maximum reward value $B$, each source should only contribute just enough to become the maximum contributor.

The truthfulness incentive may not be satisfied simultaneously with I3. If a source is guaranteed to receive a monetary reward more than its cost (e.g., $0$), sources with no data will submit random datasets to try to get a positive reward and the expected reward from such untruthfulness is non-negative.

Truthfulness differs from I5 as the former is concerned with submitting data that matches the distribution that is agreed on by the collaboration while the latter may change the distribution (e.g., more noise for stronger privacy).

### E.3. Removing the Validation Set

Now, we start from the alternative perspective and enforce (**F**) collaborative fairness from Sec. 2 using at least one common DVF $v^T$ across all sources. Without loss of generality, we assume that the DVF $v^T$ depends on the validation set $T_i$ generated from source $i$ (and optionally other sources $T_{N \setminus \{i\}}$). Would source $i$ still be incentivized to truthfully submit its dataset $\bar{D}_i$ (when the submitted $D_i$ is used to generate the validation set $T_i$ and remaining set $R_i$)?

**Truthful DVF.** Suppose that there exists a DVF $v^T$ that is $i$-truthful. Without loss of generality, we assume that the DVF $v^T$ depends on the validation set $T_i$ generated from source $i$ (and optionally other sources $T_{N \setminus \{i\}}$). The semivalue $\phi_i[v^T]$ defined based on the characteristic function $\nu^T$ (such that $\nu^T(C) = v^T(D_C)$) may not be $i$-s·truthful. This is because in the proof of Prop. 4.1 in App. D.1, we have implicitly assumed that the subtracted value of coalition $v(D_C)$ for $C \not\ni i$ would not change. However, $v^T(D_C)$ changes when source $i$ submits a different dataset $D_i$, thus the inequality in the proof may

not hold. Thus, combining a truthful DVF (if it exists) and fair semivalue may not ensure both truthfulness and fairness.

**Log scoring rule based DVF.** We consider $v^T(D_i) = \log p(\mathcal{T}_i = T_i, \mathcal{T}_{N\setminus\{i\}} = T_{N\setminus\{i\}} | \mathcal{R}_i = R_i) - \log p(\mathcal{T}_i = T_i, \mathcal{T}_{N\setminus\{i\}} = T_{N\setminus\{i\}})$. Since $\mathcal{T}_{N\setminus\{i\}}$ is unknown to source $i$, $i$ should consider the expectation over its known full dataset (i.e., given $\mathcal{D}_i = \bar{D}_i$). We note that source $i$ can indirectly control $T_i$ and $R_i$ as they are split from the submitted $D_i$ (which may not be the true dataset). Thus, the expectation should also be taken given $D_i$. Hence,

$$
\begin{aligned}
&\mathbb{E}_{\mathcal{T}_{N\setminus\{i\}}, \mathcal{T}_i, \mathcal{R}_i | \bar{D}_i, D_i}[v^T(i)] \\
&= \mathbb{E}_{\mathcal{T}_{N\setminus\{i\}} | \bar{D}_i}[\log p(\mathcal{T}_{N\setminus\{i\}} = T_{N\setminus\{i\}} | \mathcal{D}_i = D_i) - \log p(\mathcal{T}_{N\setminus\{i\}} = T_{N\setminus\{i\}})] \\
&\quad + \mathbb{E}_{\mathcal{T}_i, \mathcal{R}_i | D_i}[\log p(\mathcal{T}_i = T_i | \mathcal{R}_i = R_i) - \log p(\mathcal{T}_i = T_i)] \,.
\end{aligned}
$$

The first term is maximized in expectation by truthfully submitting $D_i = \bar{D}_i$. However, source $i$ knows the splitting algorithm (e.g., random 20% is set as the validation set) and has enough information to maximize the second term by submitting a different $D_i$ (e.g., with duplicates).

As a more specific example, consider the Bayesian model where there is an unknown mean $\mu$ and each source draws data from a Gaussian distribution centered at $\mu$. Suppose that source $i$ has 2 data points and after splitting, it will have 1 point each in the validation and remaining set. To increase $\nu^T(i)$, source $i$ can submit two identical points (at the mean of its original data) as it would increase the second term.

## F. Experiments

### F.1. Additional Details

**Compute.** The experiments are run on a machine with Ubuntu 22.04.3 LTS, 2 x Intel Xeon Silver 4116 (2.1 GHz), and NVIDIA Titan RTX GPU (Cuda 12.2). The software environments used are Miniconda and Python. Please refer to the `environment.yml` file attached for the full list of Python packages. For reference, it takes about an hour to evaluate the DVF for multiple coalitions and compute the Shapley value for one strategy of the slowest (NN-CY) experiment in Fig. 2.

Let source $i$'s dataset $D_i$ be broken down into the input matrix $\mathbf{X}_i$ and output vector $\mathbf{y}_i$. We provide more details about the various experiments below.

**(GP-FR), (NN-CY), (GP-FR-9).** For (GP-FR-9), we assign source 0 200 data points and 100 to 8 other sources.

- Strategy N: Source 0 adds noise sampled from $\mathcal{N}(0, .2)$ to the standardized $\mathbf{y}_0$,
- Strategy I: Source 0 generates synthetic data $\mathbf{X}_0^+$ outside its input domain by sampling 10% of its data and setting the first two features (i.e., $\mathbf{X}_{0\ [:,(0,1)]}^+$) to values .1 less than the minimum of each column. Source 0 artificially sets the corresponding $\mathbf{y}_{0\ [:]}^+ = 0$. The synthetic dataset $(\mathbf{X}_0^+, \mathbf{y}_0^+)$ is added to the original $D_0$.
- Strategy P: Source 0 adds zero mean Gaussian noise with standard deviation .05 (GP-FR)/ .1 (NN-CY) to each input feature in its input matrix.
- When source 1 is untruthful: Source 1 sets a random 10% of $\mathbf{y}_1$ to 0.

**(LO-HE).**

- Strategy N: Source 0 flips the class label of each of its data point independently with probability 5%
- Strategy I: Source 0 generates synthetic data $\mathbf{X}_0^+$ outside its input domain by sampling 10% of its data and setting the first two features (i.e., $\mathbf{X}_{0\ [:,(0,1)]}^+$) to values .1 less than the minimum of each column. Source 0 artificially sets the corresponding $\mathbf{y}_0^+$ to the mode, class 1. The synthetic dataset $(\mathbf{X}_0^+, \mathbf{y}_0^+)$ is added to the original $D_0$.
- Strategy P: Source 0 adds zero mean Gaussian noise with standard deviation .2 to each input feature in its input matrix.
- When source 1 is untruthful: Source 1 flips the class label of each of its data points independently with probability 5%

**(LO-BL).** We split the training data among sources 0-2 in the ratio $[.4, .3, .3]$. The training data (from 8 classes) are ordered such that there are more of class 2 for source 0 and more of class 0 and 5 for later sources, resulting in different input distribution.

- Strategy N: Source 0 submits an incorrect class label of each of its data point independently with probability 5%
- Strategy I: Source 0 generates synthetic data $\mathbf{X}_0^+$ outside its input domain by sampling 10% of its data and setting the first two features (i.e., $\mathbf{X}_{0\ [:,(0,1)]}^+$) to values .1 less than the minimum of each column. Source 0 artificially sets the

corresponding $\mathbf{y}_0^+$ to class 0. The synthetic dataset $(\mathbf{X}_0^+, \mathbf{y}_0^+)$ is added to the original $D_0$.

- Strategy P: Source 0 adds zero mean Gaussian noise with standard deviation .2 to each input feature in its input matrix.
- When source 1 is untruthful: Source 1 submits an incorrect class label of each of its data points independently with probability 5%

### F.2. DVF

In this section, we empirically evaluate if truthfulness is still incentivized under different validation set sizes and Bayesian model misspecifications.

**Different validation set sizes.** In Fig. 6, we plot the value $v(D_0)$ of source 0 when evaluated on validation set of different sizes. Across all datasets except (LO-BL), it can be observed that source 0's strategy T consistently leads to its highest value compared to other untruthful strategies, regardless of the validation set size. Larger validation sets result in smaller variance in the data value. (LO-BL) is an exception: data duplication slightly increases the DVF by reducing the high uncertainty of our prior, which may be beneficial when the validation set is well-predicted.

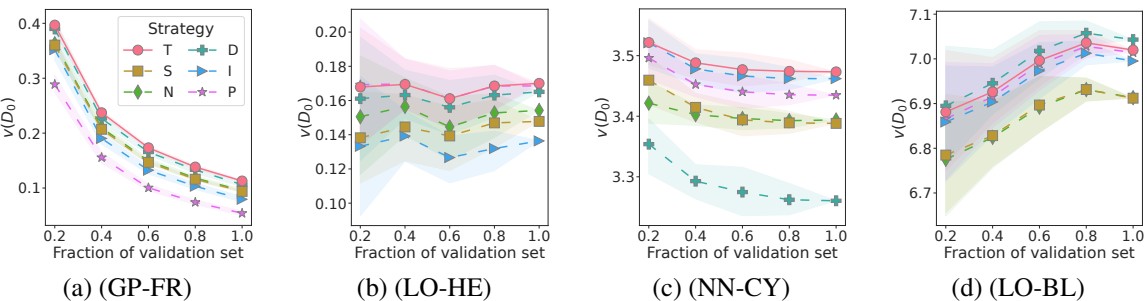

| (a) (GP-FR) | (b) (LO-HE) | (c) (NN-CY) | (d) (LO-BL) |

*Figure 6.* Graphs of value of source 0 (with 95% CI shaded across 20 sets) under its different strategies when evaluated on validation set of increasing sizes for various datasets (a)-(d).

**Increasing noise in** $p(\mathcal{T}|\theta)$**.** In Fig. 7, we plot the value $v(D_0)$ of source 0 when using validation set with increasing levels of noise added to the validation set outputs. Across all datasets, source 0's strategy T still achieves its highest value compared to other untruthful strategies when noise is low. However, as noise increases and the validation set likelihood $p(\mathcal{T}|\theta)$ in A2 becomes more misspecified, $v(D_0)$ may not be empirically $i$-s·truthful for source 0. Untruthful strategies, such as submitting a subset (S in Fig. 7a) or a noisy dataset (N in Fig. 7b-d) may result in a higher log-likelihood of the noisy validation set than the truthful strategy as they produce higher predictive uncertainty.

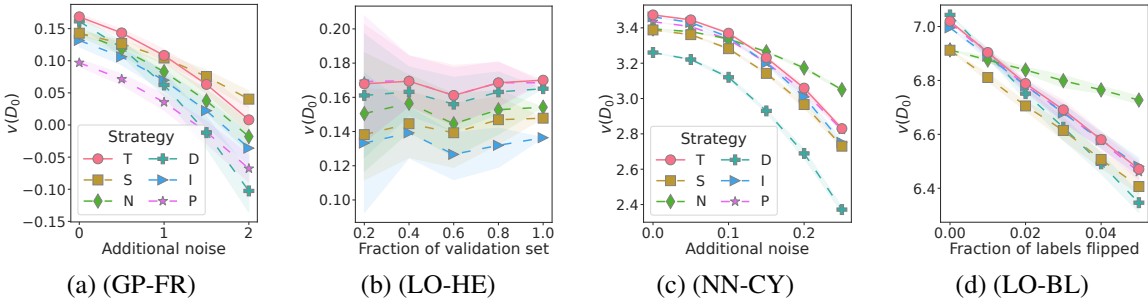

| (a) (GP-FR) | (b) (LO-HE) | (c) (NN-CY) | (d) (LO-BL) |

*Figure 7.* Graphs of value of source 0 (with 95% CI shaded across 20 sets) under its different strategies when evaluated on validation set with increasing levels of noise added to the outputs for various datasets (a)-(d).

**Different validation set input distributions.** We sort the data in the validation set using a random permutation of columns in $\mathbf{X}^*$ and evaluate source 0's strategies on the first $k$ fraction of the sorted validation set. A smaller $k$ would result in a validation set with an input distribution that differs more significantly from source 0's input distribution. In Fig. 8, across all datasets except (LO-BL), source 0's strategy T still achieves its highest value. (LO-BL) is an exception: data duplication slightly increases the DVF by reducing the high uncertainty of our prior, which may be beneficial when the validation set is

well-predicted.

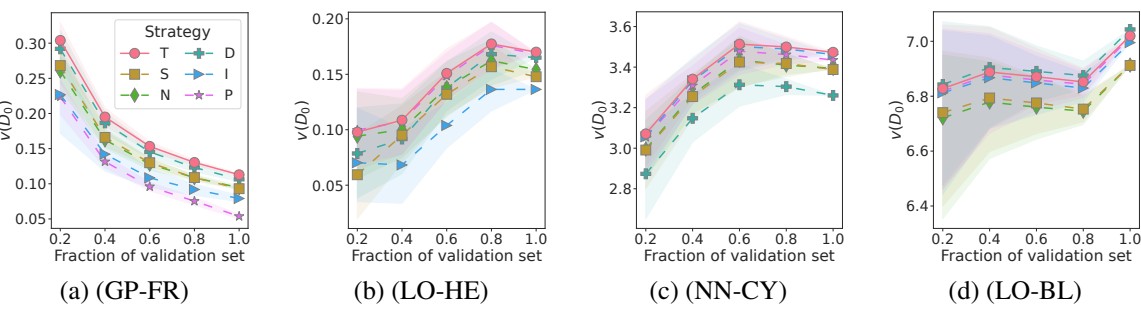

*Figure 8.* Graphs of value of source 0 (with 95% CI shaded across 20 sets) under its different strategies when evaluated on validation set of increasing sizes for various datasets (a)-(d). As the validation set is sorted, a smaller fraction corresponds to a more different input distributions.

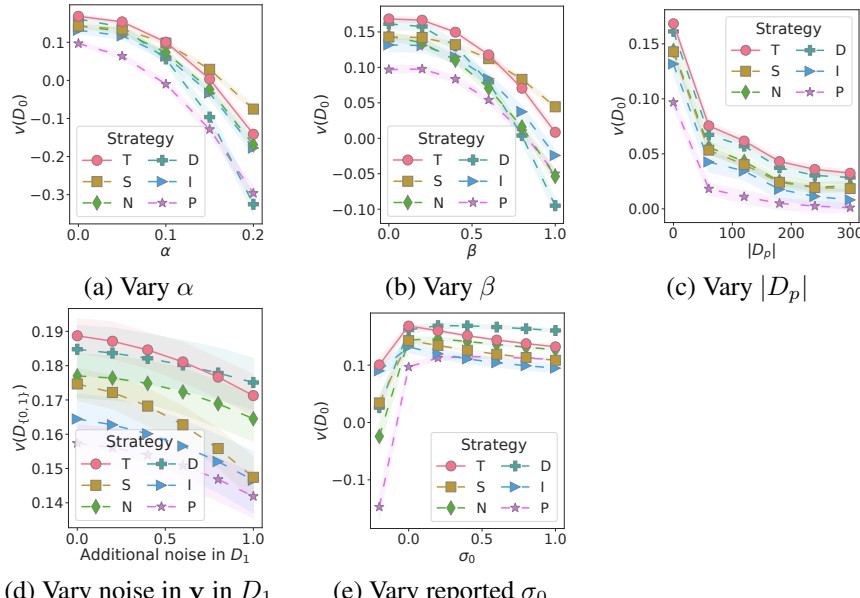

*Figure 9.* Graphs of value of source 0 (with 95% CI shaded across 20 sets) under its different strategies when evaluated on (a)-(b) validation sets generated with modified parameters, (c) a prior that has seen $|D_p|$ data for the (GP-FR) dataset, (d) source 1 adds more noise, and (e) when source 0 reports different noise in $D_0$.

**Model misspecification of $p(\mathcal{T}|\theta)$ in A2.** We consider generating the mediator's validation set $T^*$ with modified parameters of the synthetic Friedman function:

$$(A) : \mathbf{y} = 10\sin((1+\alpha)\pi\mathbf{X}_{[:,0]}\mathbf{X}_{[:,1]}) + 20(\mathbf{X}_{[:,2]} - 0.5)^2 + 10\mathbf{X}_{[:,3]} + 5\mathbf{X}_{[:,4]} + 0\mathbf{X}_{[:,5]} + \mathcal{N}(0,1) ,$$

$$(B) : \mathbf{y} = 10\sin(\pi\mathbf{X}_{[:,0]}\mathbf{X}_{[:,1]}) + 20(\mathbf{X}_{[:,2]} - 0.5)^2 + 10\mathbf{X}_{[:,3]} + 5\mathbf{X}_{[:,4]} + 0\mathbf{X}_{[:,5]} + \beta + \mathcal{N}(0,1) .$$

When $\alpha$ or $\beta$ equals 0, the original synthetic Friedman function is recovered. Larger values of $\alpha$ or $\beta$ mean greater model misspecification. As shown in Fig. 9a-b, when $\alpha$ or $\beta$ is small, source 0 submitting the true $\bar{D}_0$ achieves its highest value. However, as $\alpha$ or $\beta$ increases, and the validation set likelihood $p(\mathcal{T}|\theta)$ in A2 becomes more misspecified, $v(D_0)$ may not be empirically $i$-s·truthful for source 0. Source 0's untruthful strategies, such as [S] submitting a subset or [I] injecting mislabeled synthetic data, may result in a higher log-likelihood of the noisy validation set than the truthful strategy [T] as they produce higher predictive uncertainty.

**Informative prior in GP-FR.** Next, we consider the scenario where the prior distribution of model parameters $p(\theta)$ has been updated by some true data $D_p$. That is, $v(D_i) \triangleq \log p(\mathcal{T} = T^*|\mathcal{D}_i = D_i, \mathcal{D}_p = D_p) - \log p(\mathcal{T} = T^*|\mathcal{D}_p = D_p)$. In

Fig. 9c, we vary the size of $D_p$ and observe that across all sizes and datasets, source 0's strategy T still achieves its highest value. As the size of $D_p$ increases, the data value of source 0 decreases. When we interpret $D_p$ as the data from the coalition $C$, we can also conclude that source 0 adds less value to coalition with more data.

For models in the exponential family (App. B.4.1), this observation can be explained by the formula for the posterior natural parameters in equation 2. As $v_0$ increases (from more prior observations), the relative impact of dataset $D_i$ with $c_i$ data points and sufficient statistics decreases.

**Model misspecification of $p(\mathcal{D}_C|\theta)$ in A3.** In Fig. 9d, we plot $v(D_{\{0,1\}})$ as source 0 uses different strategies (plotted as different lines) while source 1 untruthfully increases the noise in its data (plotted on the horizontal axis). When source 1 adds a small amount of noise, source 0's strategy T still achieves its highest value as compared to other strategies. However, when source 1 increases the additional noise, source 0's untruthful strategies, such as S, may result in a higher $v(D_{\{0,1\}})$ as they produce higher predictive uncertainty.

**Misspecification of $p(\mathcal{D}_0|\theta)$.** In Fig. 9e, we plot $v(D_0)$ as source 0 uses different strategies (plotted as different lines) and reports different noise ($\mathcal{N}(0, 1 + \sigma_0)$ in its $\mathbf{y}_0$). It can be observed that as $\sigma_0$ deviates further from 0, the value $v(D_0)$ decreases. Thus, source 0 cannot artificially increase its value by reporting lower noise. This experiment also shows that source 0 can have different relationships with the model parameters as other sources (here, different noise level in $p(\mathcal{D}_i|\theta)$).

### F.3. Truthful Semivalues

**Truthful Shapley Values.** In Fig. 10, we consider source 0 using various strategies while all other sources submit their true datasets. We report the Shapley values computed for 20 different subsets of the validation set. Across all datasets except (LO-BL), it can be observed that source 0's strategy T consistently leads to its highest Shapley value. (LO-BL) is an exception: data duplication slightly increases source 0's Shapley value by reducing the high uncertainty of our prior, which may be beneficial when the validation set is well-predicted. Moreover, collaborative fairness is always evident: the Shapley values of other sources are higher when source 0 submits a less valuable untruthful dataset (e.g., with strategies S and N) than their Shapley values when source 0 is truthful. This reflects that data from other sources become more informative and important for accurate predictions when source 0 is untruthful.

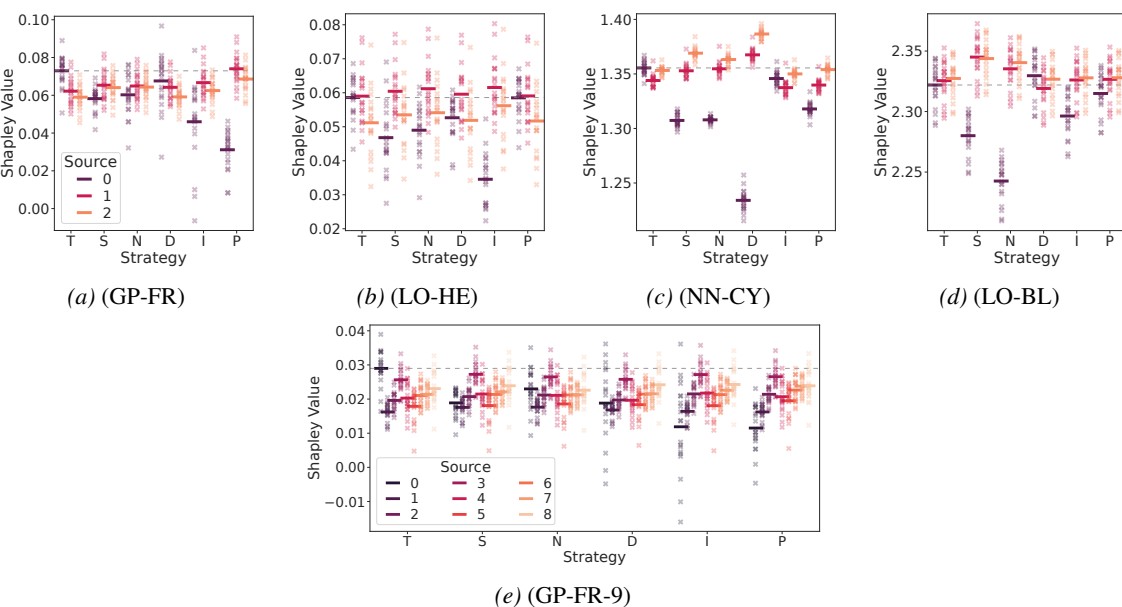

*Figure 10.* Graphs of all sources' Shapley values (mean across 20 validation sets plotted as straight bars) when source 0 uses different strategies for various datasets (a)-(e).

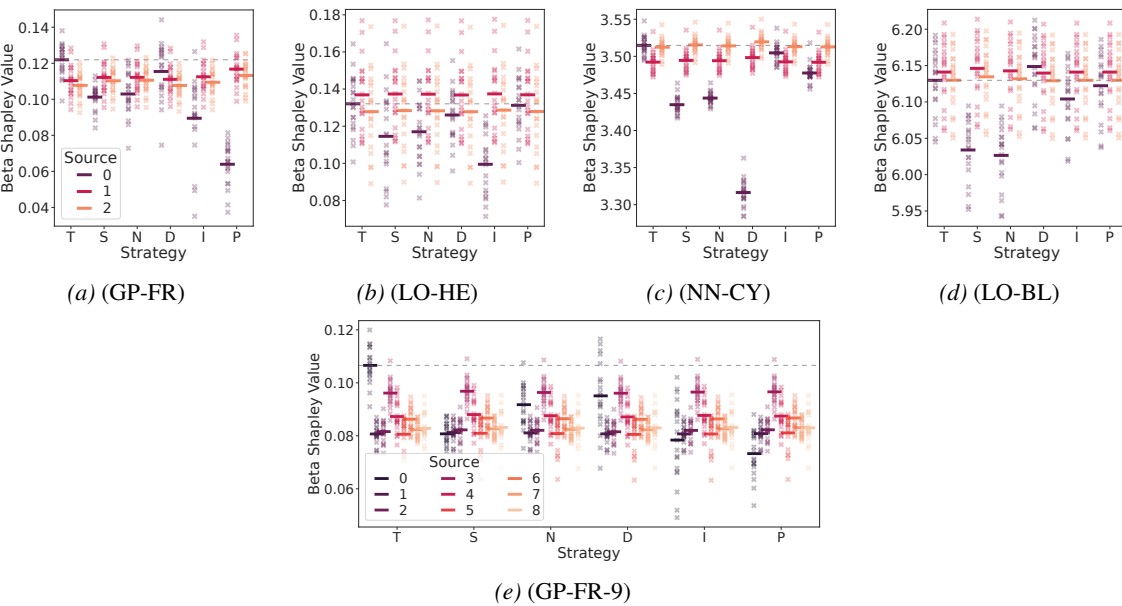

*Figure 11.* Graphs of all sources' `Beta`(16, 1) Shapley values (mean across 20 validation sets plotted as straight bars) when source 0 uses different strategies for various datasets (a)-(e).

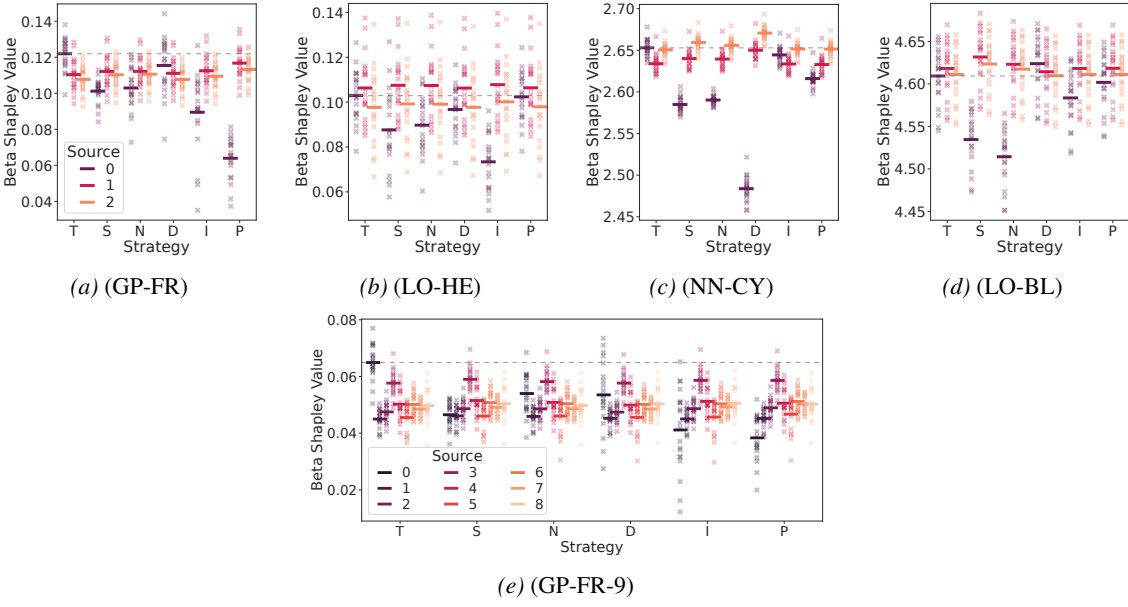

*Figure 12.* Graphs of all sources' `Beta`(4, 1) Shapley values (mean across 20 validation sets plotted as straight bars) when source 0 uses different strategies for various datasets (a)-(e).

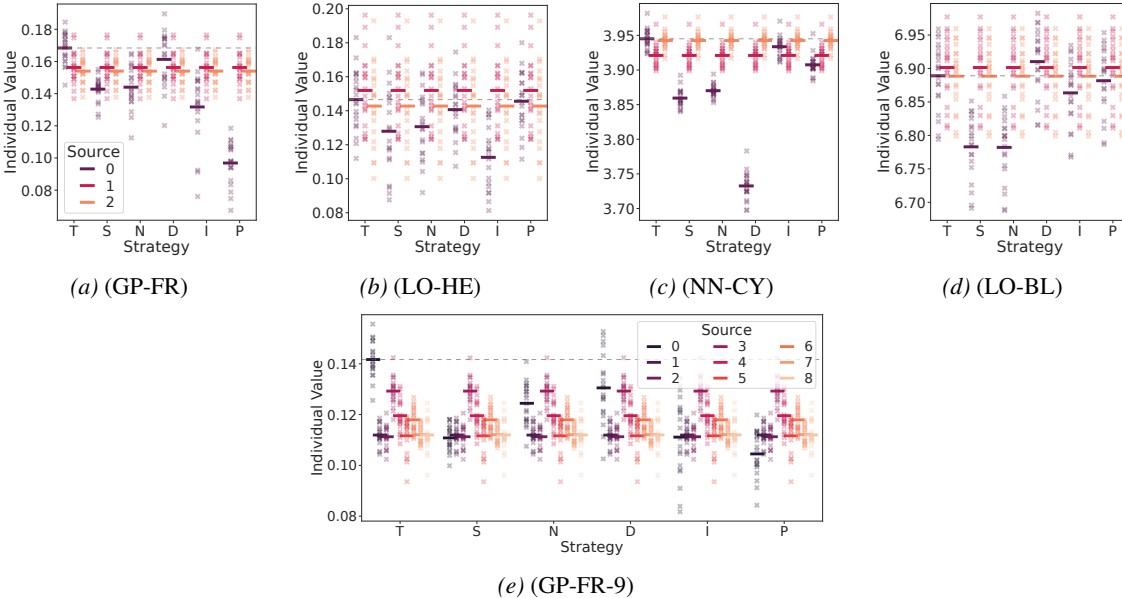

*Figure 13.* Graphs of all sources' individual value, corresponding to (Zheng et al., 2024) (mean across 20 validation sets plotted as straight bars) when source 0 uses different strategies for various datasets (a)-(e).

**Larger weights for smaller coalitions.** We consider using the Beta(16, 1) and Beta(4, 1) Shapley value (Kwon & Zou, 2021) and the individual value to place more weights on smaller coalitions. From Figs. 11,12 and 13, it can be observed that source 0 using strategy T consistently leads to higher Shapley and individual value compared to other untruthful strategies across all datasets. Additionally, it can be observed that the Beta(16, 1) (and Beta(4, 1)) Shapley value of sources 1 and 2 are (more) similar across all source 0's strategies and varying $D_0$. This is close to the extreme where $w_0 = 1$ and $w_{c \neq 0} = 0$ and the semivalues of sources 1 and 2 are independent of source 0's submission (i.e., there is no collaborative fairness in Fig. 13). We also observe that each source's Beta(16, 1) Shapley value is higher than their Shapley value in Fig. 10.

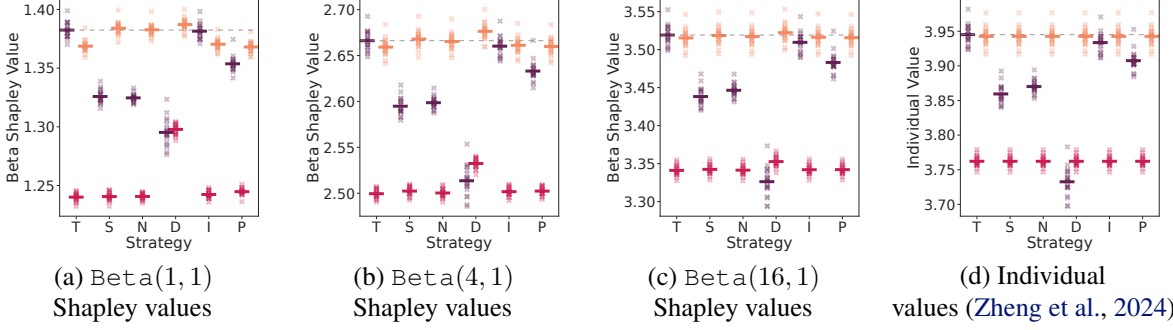

*Figure 14.* Graphs of all sources' Beta Shapley or individual values (mean across 20 validation sets plotted as straight bars) when source 0 uses different strategies and source 1 adds noise to $\mathbf{y}_1$ in the (NN-CY) experiment. From (a) to (d), the weight on smaller coalitions increases.

Fig. 14 considers the (NN-CY) experiment with an untruthful source 1. From Fig. 14a-d, as the weight on smaller coalitions increases, there is (i) **less fairness** as sources 1 and 2 reward become less influenced by source 0's strategies. Source 0's reward is also (ii) less influenced by the untruthful source 1. Unlike in Fig. 14a, in Fig. 14d, source 0's individual value from using untruthful strategy I is clearly lower than from using strategy T and source 0's semivalue when using strategy D becomes clearly lower than source 1's semivalue.

**Untruthful source 1.** In Sec. 6.2, we consider the scenario where all other sources are truthful and assumptions A1-A3 are correct. If source 1 is untruthful and the belief in assumption A3 is wrong (i.e., $D_1$ is not drawn from $p(\mathcal{D}_1|\theta)$), is the semivalue $\phi_0[\nu_{\mathfrak{D}}^v]$ still $i$-s·truthful? In Fig. 15, it can be observed that source 0's semivalue is still usually maximized

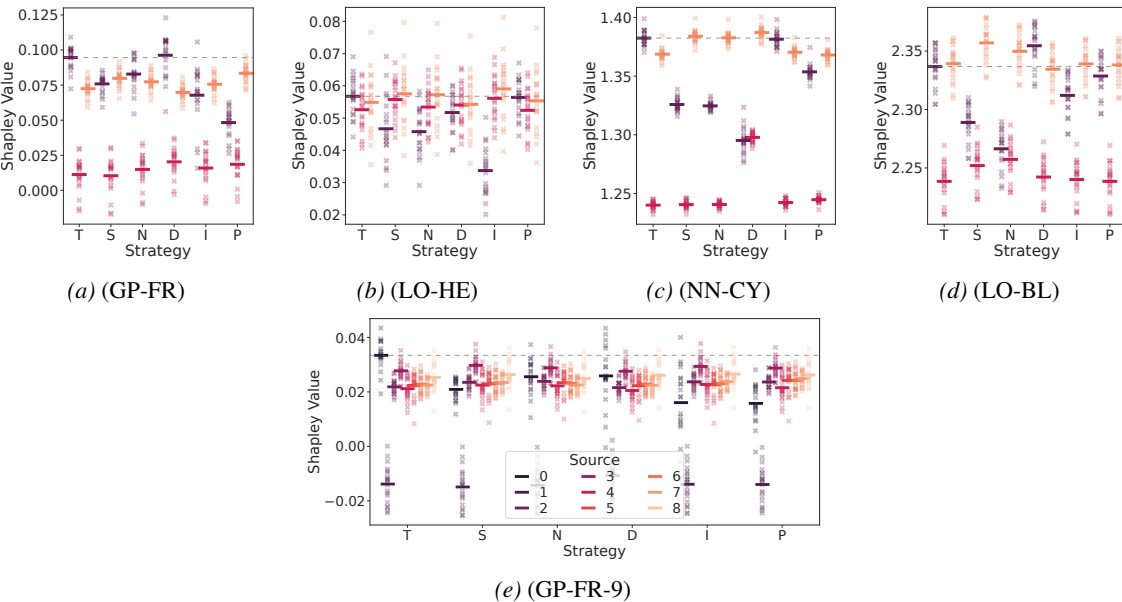

*(a)* (GP-FR)    *(b)* (LO-HE)    *(c)* (NN-CY)    *(d)* (LO-BL)

*(e)* (GP-FR-9)

*Figure 15.* Graphs of all sources' Shapley values (mean across 20 validation sets plotted as straight bars) when source 0 uses different strategies for various datasets (a)-(e). Source 1 adds noise to $\mathbf{y}_1$ in its dataset $D_1$.

by strategy T across all datasets. When source 1's dataset is very noisy (e.g., $10\%$ with the wrong output $y$ in (GP-FR), $6\%$ relabeled in (LO-BL) source 0), source 0 duplicating its dataset (strategy D) can increase $v(D_N)$ and $\phi_0[\nu_{\mathfrak{D}}^v]$ by outweighing/counteracting the wrong $D_1$.

### F.4. More Sources

*Table 3.* Table of the mean and standard deviation of the approximate Shapley values (Kolpaczki et al., 2024) based on 3000 samples across 20 runs for all 20 sources (rows) on the (GP-FR) dataset with 2000 data points. Source 0 tries different strategies (columns). Source 0's strategy T leads to the highest value for itself and lowest value for others as compared to other strategies.

|  | T | S | N | D | I | P |
|---|---|---|---|---|---|---|
| 0 | $0.01608 \pm 0.00266$ | $0.01099 \pm 0.00208$ | $0.01137 \pm 0.00322$ | $0.01605 \pm 0.00611$ | $-0.00578 \pm 0.01563$ | $0.00620 \pm 0.00395$ |
| 1 | $0.01397 \pm 0.00283$ | $0.01404 \pm 0.00282$ | $0.01414 \pm 0.00282$ | $0.01448 \pm 0.00270$ | $0.01436 \pm 0.00284$ | $0.01458 \pm 0.00284$ |
| 2 | $0.01344 \pm 0.00452$ | $0.01428 \pm 0.00460$ | $0.01360 \pm 0.00452$ | $0.01321 \pm 0.00430$ | $0.01418 \pm 0.00454$ | $0.01387 \pm 0.00449$ |
| 3 | $0.01950 \pm 0.00476$ | $0.01966 \pm 0.00484$ | $0.02002 \pm 0.00475$ | $0.01928 \pm 0.00452$ | $0.02088 \pm 0.00505$ | $0.01976 \pm 0.00479$ |
| 4 | $0.00910 \pm 0.00234$ | $0.00899 \pm 0.00235$ | $0.00910 \pm 0.00236$ | $0.00882 \pm 0.00228$ | $0.00875 \pm 0.00232$ | $0.00945 \pm 0.00232$ |
| 5 | $0.01126 \pm 0.00192$ | $0.01109 \pm 0.00195$ | $0.01134 \pm 0.00192$ | $0.01125 \pm 0.00185$ | $0.01041 \pm 0.00196$ | $0.01062 \pm 0.00189$ |
| 6 | $0.00730 \pm 0.00198$ | $0.00782 \pm 0.00203$ | $0.00766 \pm 0.00199$ | $0.00714 \pm 0.00191$ | $0.00721 \pm 0.00194$ | $0.00758 \pm 0.00197$ |
| 7 | $0.00725 \pm 0.00248$ | $0.00748 \pm 0.00252$ | $0.00715 \pm 0.00246$ | $0.00702 \pm 0.00235$ | $0.00866 \pm 0.00245$ | $0.00751 \pm 0.00246$ |
| 8 | $0.01373 \pm 0.00244$ | $0.01377 \pm 0.00242$ | $0.01413 \pm 0.00242$ | $0.01357 \pm 0.00242$ | $0.01462 \pm 0.00276$ | $0.01398 \pm 0.00249$ |
| 9 | $0.01386 \pm 0.00186$ | $0.01440 \pm 0.00185$ | $0.01393 \pm 0.00185$ | $0.01329 \pm 0.00187$ | $0.01376 \pm 0.00186$ | $0.01421 \pm 0.00185$ |
| 10 | $0.01162 \pm 0.00279$ | $0.01171 \pm 0.00289$ | $0.01151 \pm 0.00277$ | $0.01171 \pm 0.00262$ | $0.01191 \pm 0.00284$ | $0.01126 \pm 0.00278$ |
| 11 | $0.00931 \pm 0.00188$ | $0.00937 \pm 0.00191$ | $0.00927 \pm 0.00184$ | $0.00946 \pm 0.00180$ | $0.01041 \pm 0.00194$ | $0.00924 \pm 0.00189$ |
| 12 | $0.00985 \pm 0.00203$ | $0.01011 \pm 0.00199$ | $0.00999 \pm 0.00200$ | $0.00990 \pm 0.00203$ | $0.01075 \pm 0.00212$ | $0.01049 \pm 0.00208$ |
| 13 | $0.00990 \pm 0.00276$ | $0.00975 \pm 0.00282$ | $0.01008 \pm 0.00278$ | $0.00980 \pm 0.00262$ | $0.00951 \pm 0.00294$ | $0.00987 \pm 0.00276$ |
| 14 | $0.00625 \pm 0.00258$ | $0.00628 \pm 0.00267$ | $0.00614 \pm 0.00260$ | $0.00611 \pm 0.00243$ | $0.00575 \pm 0.00258$ | $0.00619 \pm 0.00260$ |
| 15 | $0.01332 \pm 0.00322$ | $0.01388 \pm 0.00333$ | $0.01320 \pm 0.00321$ | $0.01324 \pm 0.00301$ | $0.01296 \pm 0.00329$ | $0.01361 \pm 0.00324$ |
| 16 | $0.01055 \pm 0.00260$ | $0.01050 \pm 0.00266$ | $0.01062 \pm 0.00256$ | $0.01057 \pm 0.00245$ | $0.01048 \pm 0.00259$ | $0.01090 \pm 0.00265$ |
| 17 | $0.01029 \pm 0.00227$ | $0.01076 \pm 0.00234$ | $0.01036 \pm 0.00223$ | $0.01021 \pm 0.00213$ | $0.01138 \pm 0.00241$ | $0.01066 \pm 0.00231$ |
| 18 | $0.00946 \pm 0.00215$ | $0.00961 \pm 0.00218$ | $0.00995 \pm 0.00219$ | $0.00928 \pm 0.00206$ | $0.01027 \pm 0.00248$ | $0.00981 \pm 0.00219$ |
| 19 | $0.00968 \pm 0.00241$ | $0.00971 \pm 0.00245$ | $0.00977 \pm 0.00239$ | $0.00996 \pm 0.00230$ | $0.01004 \pm 0.00241$ | $0.00997 \pm 0.00241$ |

*Table 4.* Table of the mean and standard deviation of the exact Shapley values for 6 sources (rows) on the (LO-BL) dataset. We split the training data among sources in the ratio $[.2, .2, .1, .1, .2, .2]$. Source 0 tries different strategies (columns). Source 0's strategy T leads to the highest value for itself and lowest value for others as compared to other strategies.

|  | T | S | N | D | I | P |
|---|---|---|---|---|---|---|
| 0 | $1.15996 \pm 0.00843$ | $1.11632 \pm 0.00815$ | $1.12150 \pm 0.00848$ | $1.16659 \pm 0.00909$ | $1.14006 \pm 0.00863$ | $1.15340 \pm 0.00851$ |
| 1 | $1.16837 \pm 0.00824$ | $1.17778 \pm 0.00828$ | $1.17207 \pm 0.00812$ | $1.16602 \pm 0.00814$ | $1.16907 \pm 0.00816$ | $1.16976 \pm 0.00817$ |
| 2 | $1.13464 \pm 0.00846$ | $1.14159 \pm 0.00854$ | $1.13528 \pm 0.00836$ | $1.13292 \pm 0.00840$ | $1.13542 \pm 0.00847$ | $1.13570 \pm 0.00841$ |
| 3 | $1.14289 \pm 0.00772$ | $1.15127 \pm 0.00776$ | $1.14542 \pm 0.00770$ | $1.14016 \pm 0.00761$ | $1.14314 \pm 0.00765$ | $1.14358 \pm 0.00748$ |
| 4 | $1.18589 \pm 0.00735$ | $1.19497 \pm 0.00764$ | $1.18884 \pm 0.00743$ | $1.18436 \pm 0.00741$ | $1.18705 \pm 0.00742$ | $1.18725 \pm 0.00744$ |
| 5 | $1.18307 \pm 0.00827$ | $1.19114 \pm 0.00834$ | $1.18467 \pm 0.00819$ | $1.18437 \pm 0.00825$ | $1.18086 \pm 0.00803$ | $1.18380 \pm 0.00829$ |

## F.5. Removing the Validation Set

**Sources are evaluated on their own validation set.** In Fig. 16, we decide the reward value $\hat{r}_i = \sum_{j \in N} \phi_i[\nu_{\mathfrak{D}}^{T_j}]$. As source $i$ is evaluated on its own split validation set, source 0 can increase its reward $\hat{r}_i$ by injecting synthetic data that can only be predicted well by itself (strategy I vs T in Fig. 16a).

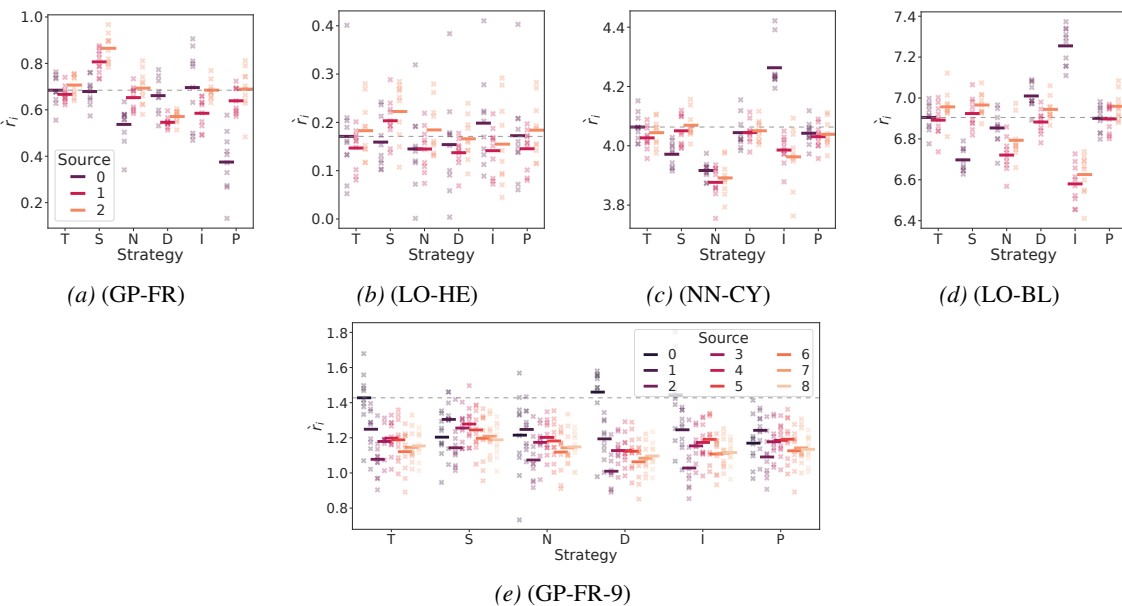

*(a)* (GP-FR)  *(b)* (LO-HE)  *(c)* (NN-CY)  *(d)* (LO-BL)

*(e)* (GP-FR-9)

*Figure 16.* Graphs of all sources' reward values $\hat{r}_i = \sum_{j \in N} \phi_i[\nu_{\mathfrak{D}}^{T_j}]$ (mean across 10 $75\% - 25\%$ training and validation splits plotted as straight bars) when source 0 uses different strategies for various datasets (a)-(e) and other sources are truthful. Source $i$ is evaluated on its own validation set.

**Sources are not evaluated on their own validation set.** In Figs. 17 and 18, we decide the reward value $\check{r}_i = \sum_{j \in N \setminus \{i\}} \phi_i[\nu_{\mathfrak{D}}^{T_j}]$. As source 0 is not evaluated on its own split validation set, we observe a similar result to Sec. 6.3. Across all datasets, source 0's strategy T usually leads to the highest reward value compared to untruthful strategies. There is an exception for (GP-FR) in Fig. 18: Strategy D, duplication, increases the log-likelihood of the validation set from others. This may be because source 0's full knowledge is captured by $D_0$ but the mediator only makes use of $50\%$ of this knowledge (thus the original knowledge may be "recovered" by duplication). Thus, to disincentivize duplication, we recommend using small validation sets.

**Reward value of other sources.** Without the mediator's validation set, the reward values of other sources do not always increase when source 0 uses an untruthful strategy. For example, in Figs. 3,16,18,19, source 0's strategy I decreases the reward of sources 1 and 2 when compared against strategy T. This difference arises because the log-likelihood of observing source 0's untruthful validation set $T_0$ may be significantly lower than that of observing $\bar{T}_0$. As a result, for any source $j \neq 0$, $\phi_j[\nu_{\mathfrak{D}}^{T_0}] < \phi_j[\nu_{\mathfrak{D}, 0 \to \bar{D}_0}^{\bar{T}_0}]$, leading to much lower $\check{r}_j$ and $\hat{r}_j$ when source 0 submits an untruthful dataset. Without the mediator's validation set, the reward values across sources and strategies may not follow the same trends as with a constant validation set.

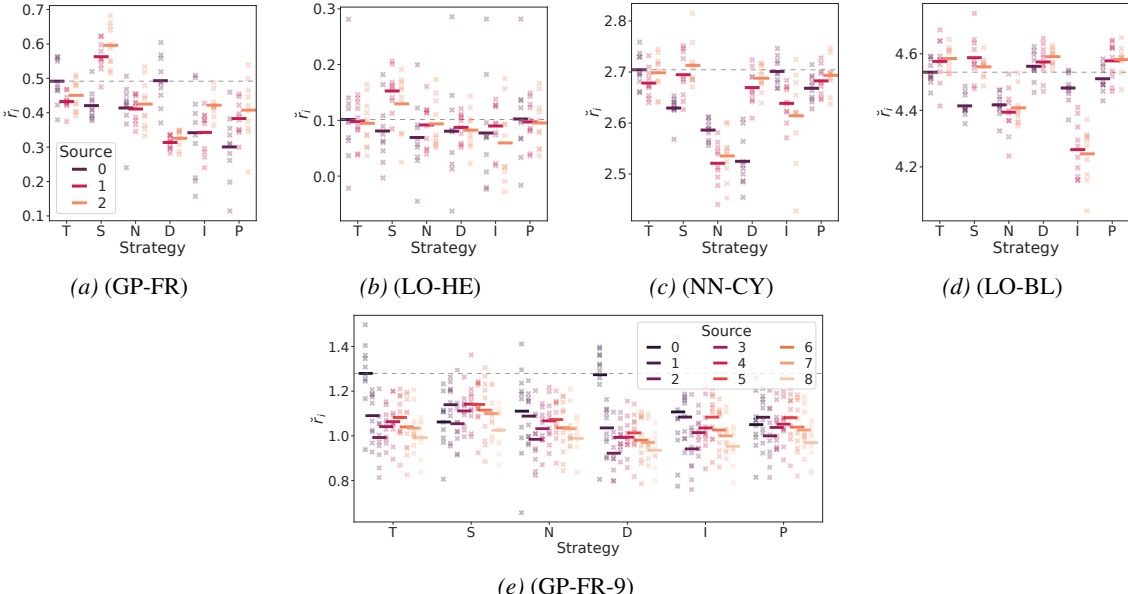

*Figure 17.* Graphs of all sources' reward values $\breve{r}_i$ (mean across 10 training and validation splits plotted as straight bars) when source 0 uses different strategies for various datasets (a)-(e).

## G. Limitations

In this section, we thoroughly discuss the limitations of our work. Some of these limitations arise from assumptions needed for theoretical guarantees and it may be non-trivial for future work to address and remove each of them.

**Reliance on a high-quality validation set.** In Secs. 3 and 4, our theoretical results rely on a high-quality validation set as we need the value to depend on some information that is unknown to all sources.

The assumption of existence is **reasonable** (App. A) when the mediator can be trusted to obtain a validation set from actual evaluations and to suggest an appropriate model. Moreover, the mediator can use outlier detection techniques, such as residual analysis (Murphy, 2022), to remove data that do not fit the model specifications and are unlikely to be from $p(\mathcal{T}|\theta)$.

The assumption of existence is **not unique to this work** and thus the community views it as a sensible and necessary starting point for research. Using a validation set is a common practice in data valuation works (Jia et al., 2019; Ghorbani & Zou, 2019), and it serves as a foundation for ensuring truthfulness (Richardson et al., 2020). To our knowledge, works that remove the ground truth assumption can only incentivize and ensure truthfulness in limited settings (e.g., multiple independent and homogeneous tasks, agents' signals on a task being conditionally independent given the ground truth, discrete predictions, or knowing the misclassification rate for the noisy ground truth in binary classification (Liu et al., 2023)). Thus, removing this validation set assumption is highly non-trivial.

Lastly, the assumption is **not too strict or restrictive**. We analyze what happens when the validation set is of smaller sizes and of lower quality (e.g., noisy) in Sec. 6 and App. F.2. We verify that truthfulness is still incentivized when the misspecification (e.g., noise) is low.

**Currently only works for Bayesian models.** We motivated the need for Bayesian models in App. A. We focus on Bayesian models as they naturally produce probabilistic predictions that account for both prior knowledge and the observed data. Such probabilistic predictions are essential for Def. 3.1 and applying proper scoring rules (see App. B.5).

The focus on Bayesian models as the first step is **reasonable** and **not unique to this work**. Bayesian models are also the focus of prior works (Chen et al., 2020; Zheng et al., 2024). In Sec. 3, we state that training non-Bayesian models involves an extra step of loss minimization rather than directly computing probabilities (based on the model specifications). Thus, it may be harder to prove Def. 3.1 in the next step. Additionally, considering the decentralized and iterative FL setting is also harder and hence left as the next step. In the last paragraph of App. B.1, we explain that

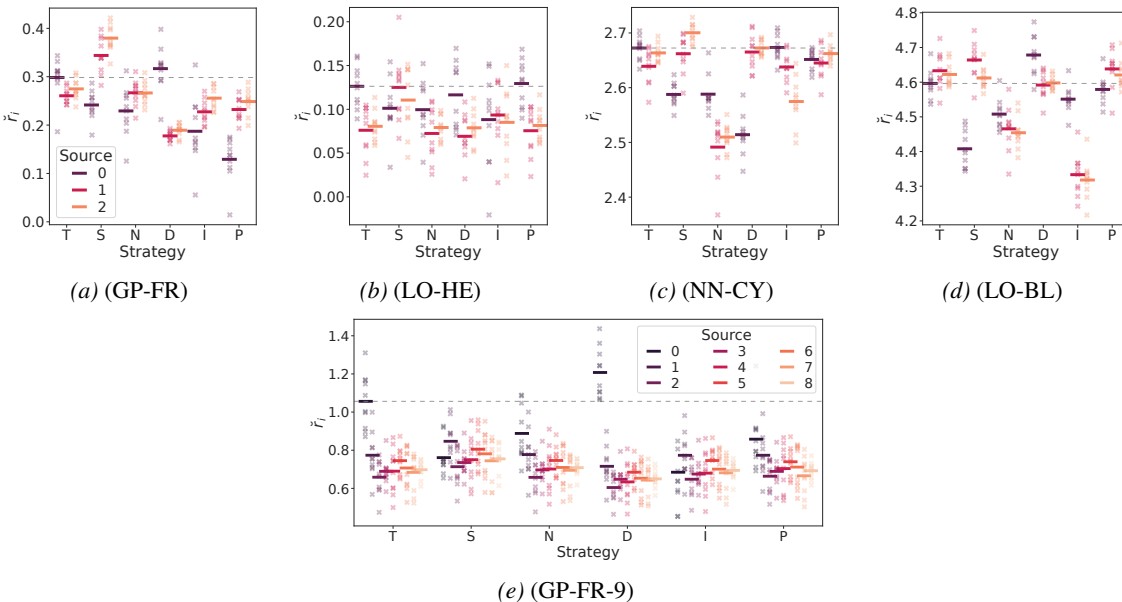

*Figure 18.* Graphs of all sources' reward values $\breve{r}_i = \sum_{j \in N \setminus \{i\}} \phi_i[\nu_{\mathfrak{D}}^{T_j}]$ (mean across 10 $50\% - 50\%$ training and validation splits plotted as straight bars) when source 0 uses different strategies for various datasets (a)-(e) and other sources are truthful. Source $i$ is not evaluated on its own validation set.

this is because FL reward mechanisms often involve giving out model rewards which conflict with strict truthfulness when there is a limited budget/performance.

Lastly, the assumption is **not too strict or restrictive** as Bayesian models can model complex relationships. We consider Gaussian processes and Bayesian neural networks in our experiments (Sec. 6). In App. B.4, we provide more background on Bayesian models.

**Requires a trusted mediator.** The valuation must be done by a trusted mediator who would not manipulate its validation set or collude with any source to increase the source's reward. The assumption of existence is **reasonable** as sources can use trusted third party platforms, legal contracts and verification checks. The assumption of existence is **not unique to this work** and shared by other data valuation works in Tab. 1.

**Computing exact semivalues may be prohibitively expensive in practice.** We consider semivalues to achieve the fairness conditions.

The scalability challenge is **not unique to this work**. It applies to any work (Ghorbani & Zou, 2019; Jia et al., 2019; Sim et al., 2020; 2023) on data values using Shapley value. The only difference is our DVF. In Sec. 3, we explain that our DVF is often used to evaluate the model performance of Bayesian models (in place of model accuracy). Thus, there are many works that have proposed efficient Monte Carlo approximations to obtain the Bayesian models and data values such as Hoffman & Gelman (2014). For the (NN-CY) experiment which involves NUTS sampling approximation on a neural network, it took 1 hour to compute the Shapley value (and utility of 8 coalitions).

**The challenge can be avoided or mitigated** due to two reasons. Firstly, in our collaborative ML setting (Sim et al., 2020), there are fewer data sources (e.g., hospitals holding larger datasets). Secondly, in the remark on efficiency in Sec. 4, we describe how our theoretical results would also hold for unbiased semivalue estimators. On average, the approximate semivalues will still incentivize truthfulness and fairness. Using approximate semivalues would significantly reduce the number of coalitions to evaluate to $O(n \log n)$ and scalability would also improve with the discovery of more efficient approximation techniques such as Kolpaczki et al. (2024); Li & Yu (2024). We evaluate on 20 sources in Tab. 3 using the approximation proposed by Kolpaczki et al. (2024).

**The empirical analysis shows that truthfulness is only still incentivized under minor model misspecification.** The limitation is **reasonable** as in practice, the misspecifications are likely to be minor as (1) the mediator and sources can use outlier detection techniques, such as residual analysis (Murphy, 2022), to remove data that do not fit the model specifications and (2) the mediator and sources will likely discuss and re-specify better model/distributions to improve

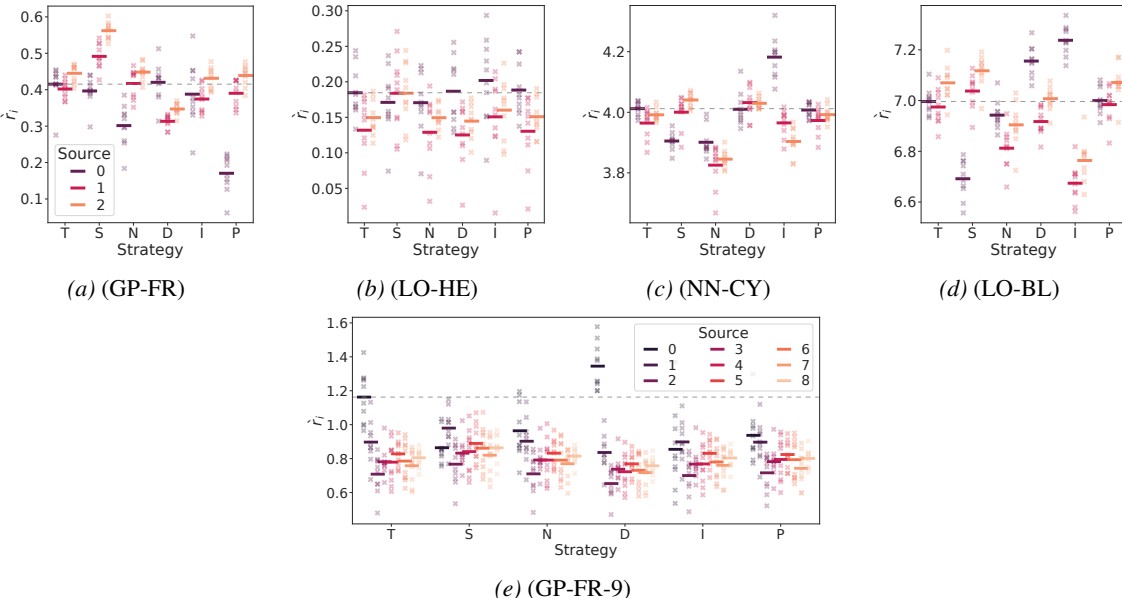

*Figure 19.* Graphs of all sources' reward values $\hat{r}_i = \sum_{j \in N} \phi_i[\nu_{\mathfrak{D}}^{T_j}]$ (mean across 10 $50\% - 50\%$ training and validation splits plotted as straight bars) when source 0 uses different strategies for various datasets (a)-(e) and other sources are truthful. Source $i$ is evaluated on its own validation set.

predictive performance. The analysis should be viewed as comprehensive instead of limited as other theoretical works (Chen et al., 2020) may not have discussed or evaluated assumption violations.

## H. Other Questions

1. **What is the novelty of the work since semivalues are already used to ensure fairness and pointwise mutual information is already used to incentivize truthfulness?**

   Both incentives have not been simultaneously considered. We explicitly identified the additional condition needed to extend the proof of truthfulness incentive to the value of coalitions and semivalues. Thus, we are able to derive theoretical guarantees. Our contributions also involve identifying pointwise mutual information/log-likelihood based on a validation set to ensure truthfulness is more *compatible* with fairness over alternative definitions, such as peer prediction (Chen et al., 2020) and other proper scoring rules. We also carefully justified the assumptions in App. A.

   Next, we novelly analyze why strict truthfulness may not be possible with a limited budget and without a validation set.

   Lastly, we novelly provide comprehensive experiments with multiple sources and probabilistic models and even tested cases where the probabilistic model might be misspecified. We find that truthfulness is still incentivized under minor misspecifications.

2. **At first glance, the work may seem like a simple extension of existing works. Can we clarify the technical contributions of the work?**

   The combination of pointwise mutual information and semivalues may not be that trivial and simple. To extend Zheng et al. (2024) and proper scoring rules to ensure fairness, there is a need to account for additional terms which depend on others' data, identify and justify the additional assumption A3 (which we have done in Sec. 3 and App. A) and define the propositions such as Prop. 3.2 precisely. For a complete treatment, we also consider potential violations through our experiments. We have also motivated why all sources would prefer the truthful Nash equilibrium, further justifying A3. All the above constitute part of the novel technical contributions.

   Next, we think that technical novelty may not only stem from complex or fundamentally new solutions. Technical novelty can also involve identifying an open problem and providing a clear and rigorous solution, which we believe is non-trivial. To elaborate, we have recognized that a common limitation of collaborative fairness works (Sim et al., 2020; 2023; Xu et al., 2021b) is that they assume truthfulness without incentivizing or verifying. **Then, we seek to**

**remove this common and strong assumption**. We identified the problem of provably ensuring both truthfulness and collaborative fairness before proposing the first mechanism that achieves it. Next, we have comprehensively evaluated our solution formally and empirically (which is not often seen in truthfulness works). These novel technical contributions are highlighted and differentiated from existing works in Tab. 1.

It may take multiple future works, each a significant contribution, to address and remove other assumptions in App. A.

3. **Why do we not consider federated learning and giving out rewards in each round? How does this work compare against Xu et al. (2021a); Wu et al. (2024)?**

   See App. B.1.

4. **How does the proposed mechanism handle the impact of differential privacy techniques (e.g., Gaussian noise addition) on truthfulness? Would such techniques violate the truthfulness guarantees?**

   The proposed mechanism seeks to ensure that each source $i$ submits the dataset $D_i$ drawn from the agreed on likelihood $p(D_i|\theta)$. If the likelihood does not account for differentially private techniques (e.g. Gaussian noise addition), truthfulness guarantees that use these techniques would decrease the valuation. If the likelihood accounts for them, truthfulness guarantees that use these techniques as described (e.g., specified level of noise) would maximize the valuation. See the last part of App. E.2.

5. **Can we comment about more complex and potentially untruthful strategies such as synthetic data generation with an ML model?**

   In Sec. 2, we define the true dataset $\bar{D}_i$ as the dataset that fully captures the knowledge of source $i$. We did not define truthfulness as the dataset actually collected. This is because the source may know that flipping the image or slightly modifying some pixels should still result in the same value or label. It is also reasonable that these uncollected augmentations increase the truthful data value instead of decreasing it. To resolve this issue, we *consider these helpful augmentations capturing knowledge as truthful*. Thus, synthetic data generation with an ML model *may not be untruthful* (i.e., there is no ground truth for us to evaluate the truthfulness incentive of the mechanism under this strategy).

   To better align our definition of truthfulness with an alternative version of submitting a collected dataset as-is, we believe the mediator should commit to doing the helpful data augmentation (such as image rotation, adding input noise of a limited magnitude, and training a generative model for each source) and value each source after the augmentation. Thus, there would be no incentive for each source to do it by themselves. Future work can also consider generative models instead of discriminative models.

6. **Does the reliance on semivalues make the mechanism unscalable?**

   High computational cost is needed to achieve the fairness conditions.

   We agree that scalability is a challenge, however, it would apply to **any work (Ghorbani & Zou, 2019; Jia et al., 2019; Sim et al., 2020; 2023) on data values using Shapley value** and semivalues and is less important in collaborative ML when there are fewer data sources (e.g., hospitals holding larger datasets). We hope that scalability would improve with the discovery of more efficient approximation techniques such as Li & Yu (2024).

   In the remark on efficiency in Sec. 4, we describe how our theoretical results would also hold for unbiased semivalue estimators. Thus, our method is still applicable to more sources.

   We evaluate on 20 sources in Tab. 3 using the approximation proposed by Kolpaczki et al. (2024).

7. **Comment on the real world applicability of this work.**

   As explained in App. B.1, we consider the collaboration between a few sources. Thus, scalability is less of an issue.

   In the reply to the previous question, we explain how semivalues can be approximated unbiasedly and efficiently. Our theoretical results still hold for these unbiased semivalue estimators.

   As existing data valuation works have also used these semivalue approximations, we focus on discussing the computation efficiency of our DVF here. In Sec. 3, we explain that our DVF is often used to evaluate the model performance of Bayesian models (in place of model accuracy). Thus, there are many works that have proposed efficient Monte Carlo approximations to obtain the Bayesian models and data values such as Hoffman & Gelman (2014). For the (NN-CY) experiment which involves NUTS sampling approximation on a neural network, it took 1 hour to compute the Shapley value (and utility of 8 coalitions).

8. **Is splitting out a validation set for each source necessary in Sec. 5.2?**

   Without dataset splitting, we can consider the case with $n$ games and in the $j$-th game, each source $j$'s full dataset is used to evaluate and incentivize truthfulness of all other sources such as $k$.

   This approach can also be interpreted as changing the validation and remaining set split in each game (to support a $0 - 100$ split for source $j$ in the $j$-th game). If so, the approach satisfies our modified fairness conditions.

   We chose the version in our paper for the ease of writing and because intuitively it feels fairer that in the $j$-th game, source $k$'s semivalue still depends on $n$ sources' datasets $\mathfrak{D}$ including source $j$'s data $D_j$ instead of only $n - 1$ sources (especially when they are the only 2 sources present). The dependence would require splitting source $j$'s data into the remaining and validation set.

   When only 2 sources are present, the approach without data splitting would always result in equal rewards despite differing contributions. With data splitting, their rewards would differ as we consider the semivalue in each game.

