# OpenReview forum: "Incentivizing Truthfulness and Collaborative Fairness in Bayesian Learning"
_ICML.cc/2026/Conference — ICML 2026 spotlight_

### Official Review · Reviewer_AwcL · 2026-03-06

**Soundness:** 3
**Presentation:** 3
**Significance:** 3
**Originality:** 3
**Overall Recommendation:** 5
**Confidence:** 3

**Summary:**

This paper studies how to simultaneously achieve truthfulness in data submission and collaborative fairness in cooperative Bayesian learning. The authors point out a structural divide in existing data valuation approaches at the mechanism level: semivalue-based methods (e.g., Shapley value) satisfy cooperative fairness axioms but provide no guarantee of truthful reporting; conversely, likelihood- or mutual information–based mechanisms can incentivize truthful submission but fail to ensure collaborative fairness. To bridge this gap, the paper proposes a Bayesian data valuation function (DVF) based on validation-set log-likelihood and constructs a cooperative game on top of it, where rewards are allocated using a semivalue (such as the Shapley value). Under Assumptions A1–A3, the paper proves that the proposed DVF is strictly truthful in expectation, and that the semivalue inherits this property, thereby guaranteeing the existence of a truthful Nash equilibrium.

**Compliance With Llm Reviewing Policy:**

Affirmed.

**Final Justification:**

We are very grateful for the author's reply. Based on the original manuscript and the author's reply, the above is our final score for this article.

**Key Questions For Authors:**

No

**Limitations:**

The theoretical results rely on several key assumptions, including: all participants share the same Bayesian generative model; the validation data are indeed generated from this model; and each participant believes that others’ data are generated by the same process. These assumptions may not hold in practical data markets or cross-institutional collaborations. In particular, when there is model misspecification or heterogeneous beliefs about data generation processes, truthfulness guarantees may fail. The paper provides limited discussion of such assumption violations and their implications.

**Strengths And Weaknesses:**

Strengths

1. The paper clearly distinguishes two core properties:
(T) Truthfulness, meaning that truthful submission maximizes expected utility;
(F) Collaborative fairness, meaning that rewards satisfy cooperative game axioms such as F0–F3.
The authors highlight that these two properties may be inherently in tension in multi-party collaborative settings and provide a unified formal framework to characterize this interaction. This problem reformulation clarifies the scope and limitations of existing methods and establishes a solid foundation for subsequent theoretical analysis.

2. The paper systematically integrates the Bayesian predictive model structure, a coalition value definition based on log-likelihood, and semivalue-based allocation rules into a cooperative game framework. While prior work typically analyzes the truthfulness of log-likelihood–based valuation functions at the single-source level, this paper extends the analysis to the coalition level. Under the stated assumptions, it proves that semivalue allocations preserve truthfulness, thereby implying the existence of a truthful Nash equilibrium.

3. The paper analyzes several key structural constraints of the mechanism, including the necessity of hiding validation information for maintaining truthfulness, the impact of budget constraints on the incentive structure, and the structural conflict between fairness and truthfulness when a global validation set is removed. These discussions strengthen the characterization of the mechanism’s applicability and theoretical boundaries.

4. Given the stated assumptions, the logical flow among the main propositions is clear and internally consistent. The proofs of truthfulness and fairness properties are structurally coherent and do not exhibit obvious logical gaps.

Weaknesses

1.	The main contribution of the paper is the systematic integration of Bayesian scoring rules with cooperative game allocation mechanisms and the extension of truthfulness results to the coalition level. However, the core mechanism components themselves are not fundamentally new structural designs. The novelty lies more in theoretical synthesis and generalization rather than in a fundamentally new mechanism form.

2. The method relies on a Bayesian predictive structure and a validation-based scoring mechanism. Its computational cost and scalability for non-Bayesian models or large-scale deep learning systems remain insufficiently validated. Moreover, the mechanism heavily depends on a hidden validation set. In realistic environments, validation data may be leaked or subject to distribution shift, which could undermine the robustness of the truthfulness guarantees. Chapter Two emphasizes that each user can only submit one dataset. How can we prevent a user from profiting by submitting split data through multiple accounts?

3. The theoretical results rely on several key assumptions, including: all participants share the same Bayesian generative model; the validation data are indeed generated from this model; and each participant believes that others’ data are generated by the same process. These assumptions may not hold in practical data markets or cross-institutional collaborations. In particular, when there is model misspecification or heterogeneous beliefs about data generation processes, truthfulness guarantees may fail. The paper provides limited discussion of such assumption violations and their implications.

---

> ### Author Rebuttal · Authors · 2026-03-31
>
> Thank you for your positive and detailed review that recognises the clarity and theoretical contributions of our work. We address your concerns below.
>
> > [W1]
>
> The reviewer is right to notice that the core mechanism components themselves (pointwise mutual information and semivalues) are not fundamentally new structural designs.
> Instead, the novelty lies more in the combination, which we believe is still non-trivial.
> * To extend (Zheng et al., 2024) and proper scoring rules to ensure fairness, there is a need to account for additional terms which depend on others’ data, identify and justify the additional assumption A3 (which we have done in Sec. 3 and App. A1) and define the propositions such as Prop. 4.1 precisely.
> * For a complete treatment, we also consider potential violations through our experiments. We have also motivated why all sources would prefer the truthful Nash equilibrium, further justifying A3.
>
> We believe that technical novelty may not require new structural designs but can also involve identifying an open problem and providing a clear and rigorous solution. To this end, we have recognized that a common limitation of collaborative fairness works is that they assume truthfulness without incentivizing or verifying. _Then, we seek to remove this common and strong assumption._ We identified the problem of provably ensuring both truthfulness and collaborative fairness before proposing the first mechanism that achieves it. Next, we have comprehensively evaluated our solution formally and empirically (which is not often seen in truthfulness works). These novel technical contributions are highlighted and differentiated from existing works in Tab. 1.
>
> > [W2] The method relies on a Bayesian predictive structure and a validation-based scoring mechanism. Its computational cost and scalability for non-Bayesian models or large-scale deep learning systems remain insufficiently validated. Moreover, the mechanism heavily depends on a hidden validation set...
>
> We view our work as a valuable first step that addresses one common and strong assumption (submission truthfulness). However, it may require multiple future works, each a significant contribution, to address and remove the other assumptions.  Future works can extend the truthfulness guarantees to other more complex models.
> We justify these assumptions in App. A. Specifically,
> * Bayesian models are also often used in machine learning tasks and have been made more efficient by recent advances in efficient posterior approximations such as new Monte Carlo (MC) sampling methods.
> * Using a validation set is a common practice in data valuation works (Jia et al., 2019; Ghorbani & Zou, 2019), and it serves as a foundation for ensuring truthfulness (Richardson et al., 2020). To our knowledge, works that remove the need for validation set and ground truth can only incentivize and ensure truthfulness in limited settings (e.g., multiple independent and homogeneous tasks, agents’ signals on a task being conditionally independent given the ground truth, discrete predictions, or knowing the misclassification rate for the noisy ground truth in binary classification (Liu et al., 2023)). Thus, removing this validation set assumption is highly non-trivial.
>
> > [W2]  How can we prevent a user from profiting by submitting split data through multiple accounts?
>
> We can prevent a user from profiting by submitting split data through multiple accounts by verifying the source's identity (e.g., through a business registration ID) or by adapting (Han et al., 2022) which proposes a robust variant of the Shapley value which ensures that the total reward of a source from cloning and multiple submissions is lower than the reward without cloning.
> This is left to future work and the assumption of no cloning/resubmissions is also used by (Chen et al., 2020; Zheng et al., 2024).
>
>
> > [W3]
>
> In Sec. 6 and App F.2., we have shown that minor model misspecifications still incentivize sources' truthfulness. In practice, the misspecifications are likely to be minor as the mediator and sources can use outlier detection techniques, such as residual analysis (Murphy, 2022), to remove data that do not fit the model specifications.
>
> Moreover, sources can resolve heterogeneous beliefs about the data generation process by discussing or voting on the model specifications and relationships based on historical, public, or similar data at the start of the collaboration.
>
> We believe that our empirical evaluation in Sec. 6 and App F.2 and justification of assumptions in App. A. may not be limited. Instead, they may be comprehensive given that other related theoretical works (Chen 2020) do not discuss or evaluate assumption violations.
>
> Thank you once again and we hope the clarifications address or reduce the weaknesses of our work!

---

> > ### Author Rebuttal · Reviewer_AwcL · 2026-04-01
> >
> > Thank you very much for the author's reply; it has resolved all my issues. I hope that a discussion of the above issues will be included in the new version to avoid any confusion for readers.

---

> > > ### Author Response · Authors · 2026-04-08
> > >
> > > Dear Reviewer AwcL,
> > >
> > > We are glad that our reply resolved your concerns. Thank you for your valuable feedback and we will include the clarifications in the revision. We highly appreciate your time and effort devoted in reviewing our paper, and we wish you all the best!
> > >
> > > Regards,
> > >
> > > The Authors.

---

### Official Review · Reviewer_oKcK · 2026-03-09

**Soundness:** 3
**Presentation:** 3
**Significance:** 3
**Originality:** 2
**Overall Recommendation:** 5
**Confidence:** 4

**Summary:**

This paper addresses the challenge of simultaneously ensuring truthfulness and collaborative fairness in Bayesian learning settings where multiple data sources contribute to a model. The authors propose a mechanism that combines a data valuation function (DVF) that incentivizes strict truthfulness with semivalue that satisfies key fairness properties. The paper also explores practical constraints such as limited budgets and the absence of a validation set, providing insights into how truthfulness guarantees can be maintained. Rigorous proof and comprehensive experiments demonstrate the effectiveness of their proposed mechanism.

**Compliance With Llm Reviewing Policy:**

Affirmed.

**Final Justification:**

The rebuttal addressed my main concerns, thus changing my evaluation.

**Key Questions For Authors:**

1. Regarding the content after "However, with only two data sources, peer prediction approaches" in the introduction: Can you add a numerical calculation example to demonstrate that Chen 2020 and Zheng 2024's methods indeed have the problem you mentioned, and calculate the results of your method (with semivalue added) to show that you have indeed solved this problem?
2. Regarding the definition of (F) on page three: Should monotonicity and strict monotonicity be defined for different data providers i and j? I don't think it should be defined for D and D', at least your condition w_C+w_ {C+1}>0 is for different data providers i and j to satisfy F3, rather than for the same data provider to satisfy F3 in D and D' (I think the condition for this is w_C>0).
3. Regarding Section 5.2: I believe that numerical examples should be added to provide a clearer description of the mechanism you proposed and its effects.

**Limitations:**

The author should consider the assumption of a trusted mediator as a limitation, as the mediator can collude with a data provider to manipulate its validation set and increase the data value of the data provider.

**Strengths And Weaknesses:**

Strengths:
1. The paper features a clear motivation, and the challenge of simultaneously ensuring truthfulness and collaborative fairness raised in this paper is important.
2. The mechanism proposed in this article is intuitive (directly combining the peer prediction method and semivalue) and effective, and the proof is also clear.
3. The paper presents a comprehensive evaluation of its proposed mechanism.

Weakness:
1. The article has a lot of redundant content: (1) both questions [I] and [II] in the introduction express the need to satisfy both truthfulness and fairness simultaneously; (2) Section 2 provides a rough definition of (F) before (T) and a detailed definition after (T), and the two parts should be merged; (3) The appendix repeatedly introduces why Bayesian learning and why validation sets are used.
2. Some parts of the article need to add numerical examples to make it clearer, see Key Questions for details.

---

> ### Author Rebuttal · Authors · 2026-03-31
>
> Thank you for recognizing the clarity, motivation, and comprehensiveness of our work and for your helpful suggestions! We will address your concerns below.
>
> > [W1] (1) both questions [I] and [II] in the introduction express the need to satisfy both truthfulness and fairness simultaneously; (2) Section 2 provides a rough definition of (F) before (T) and a detailed definition after (T), and the two parts should be merged; (3) The appendix repeatedly introduces why Bayesian learning and why validation sets are used.
>
> Thank you for raising the concern and we will try to reduce some repetition for (3) in our revision.
> For (1), Questions [I] and [II] differ
> * [I] focuses on truthfulness and considers model performance instead of fairness; [II] considers collaborative fairness;
> * [I] concerns the DVF while [II] concerns the subsequent post-processing of DVF. [I] is answered by Sec. 3 while [II] is answered by Sec. 4 where we use semivalues to ensure fairness but verify that its preserves the truthfulness of a truthful DVF.
>
> While we acknowledge that the definitions can be merged to reduce repetition for (2), our goal was to provide a simpler overview in Sec. 2 before providing the details that need more context and mathematical terms later in Sec 3.
>
> > [Q1] Can you add a numerical calculation example to demonstrate that Chen 2020 and Zheng 2024's methods indeed have the problem you mentioned, and calculate the results of your method (with semivalue added) to show that you have indeed solved this problem?
>
> With $2$ sources $i$ and $j$, the rewards based on Chen 2020 method will be
> $r\_i = \log p(\mathcal{D}\_j = D\_j \mid \mathcal{D}\_i = D\_i) - \log p(\mathcal{D}\_j = D\_j) = \log \frac{p(\mathcal{D}\_j = D\_j, \mathcal{D}\_i = D\_i)}{p(\mathcal{D}\_j = D\_j) p(\mathcal{D}\_{i} = D\_i)} = \log p(\mathcal{D}\_i = D\_i \mid \mathcal{D}\_j = D\_j) - \log p(\mathcal{D}\_i = D\_i) = r\_j$
> They are always equal regardless of what $\mathcal{D}$ and $D$ are used.
> We consider the GP Friedman dataset. where source $0$ and $1$ have 400 and 300 data points respectively. Our method solves the problem as source $0$ fairly has a Shapley value of $0.098$ that is larger than source $1$'s $0.066$.
>
> Zheng 2024 method is equivalent to rewarding source $i$ with our $v(D\_i)$ instead of the fair semivalue $𝜙\_i[\nu^v\_\mathcal{D}]$.
> We set up a numeric example with 3 sources using the GP Friedman dataset. Source $0$ has 400 data points, source 1 has another 300 unique data points while source $2$ has 250 data points that are a subset of $D_0$.
> We observe that based on Zheng 2024, source $2$ gets slightly higher valuation with $v(D_2) = .1346$ vs $v(D_1) = 0.1306$.
> However, source $2$ gets a slightly lower Shapley value $𝜙\_2[\nu^v\_\mathcal{D}] = 0.042$ vs $𝜙\_1[\nu^v\_\mathcal{D}] = 0.049$, thus solving the problem.
> This is because source $2$ contributes less to improving predictions when source $0$ is present, e.g., $v(D_{0,2}) = 0.1469$ vs  $v(D_{0,1}) = 0.1647$.
>
> > [Q2] Regarding the definition of (F) on page three: Should monotonicity and strict monotonicity be defined for different data providers i and j? I don't think it should be defined for D and D', at least your condition w_C+w_ {C+1}>0 is for different data providers i and j to satisfy F3, rather than for the same data provider to satisfy F3 in D and D' (I think the condition for this is w_C>0).
>
> Thank you for the correction! We realize that we have made a typo when referencing (Domenech, 2016) and the condition should be each $w_c>0$ instead of $w_c +w_ {c+1}>0$. We will revise it in the paper (in Sec. 2, remark in Sec. 4 and App. B.2.) and have verified that the other results still hold after fixing the typo.
>
> Note that monotonicity should hold for each data provider $i$ and $j$ and for any dataset $\mathfrak{D}$ and $\mathfrak{D}'$ that satisfy the pre-conditions. In App. B.2., the desirability property compares the semivalues across players in the same game instead.
>
> > [Q3] Section 5.2: numerical examples should be added to provide a clearer description of the mechanism you proposed and its effects.
>
> Thank you for the suggestion! We will add this [linked figure with notations](https://ibb.co/Qw9G6d2) to clearly describe the steps involved in the mechanism as the notations may help more than numeric values.
>
> >[L1] The author should consider the assumption of a trusted mediator as a limitation, as the mediator can collude with a data provider to manipulate its validation set and increase the data value of the data provider.
>
> We will add this as a limitation and suggest that it can be mitigated by using trusted third party platforms and verification checks. We will also note that a trusted mediator and validation set are used by other data valuation works in Table 1.
>
> Thank you once again and we hope that our clarifications and numerical examples address your concerns and improve your opinion of our work.

---

> > ### Author Rebuttal · Reviewer_oKcK · 2026-04-03
> >
> > Thank you for your responses. I'd like to raise the score to 5.

---

> > > ### Author Response · Authors · 2026-04-08
> > >
> > > Dear Reviewer oKcK,
> > >
> > > We are grateful to hear that our response addressed your concern and improved your opinion of our work. We appreciate your constructive feedback and will incorporate the clarifications in our revised version. Thank you for your support and engagement during the rebuttal period. We wish you all the best!
> > >
> > > Regards,
> > >
> > > The Authors.

---

### Official Review · Reviewer_LLXE · 2026-03-21

**Soundness:** 4
**Presentation:** 3
**Significance:** 4
**Originality:** 4
**Overall Recommendation:** 5
**Confidence:** 2

**Summary:**

This paper studies the data provider incentives and equilibrium in collaborative machine learning. It focuses on the truthfulness, fairness, and the possibility of waiving the validation set. The paper shows that unknown information is necessary for truthfulness; proposes a mechanism that provably ensures both truthfulness and collaborative fairness at equilibrium; and analyzes implications and relaxation when the mediator lacks a validation set.

**Compliance With Llm Reviewing Policy:**

Affirmed.

**Final Justification:**

I would like to maintain the evaluation score as my rebuttal acknowledgement. This paper with the authors rebuttal is overall valuable contribution to the community.

**Key Questions For Authors:**

This paper is well written and is valuable for the community. The following are some nice-to-have comments.
* In an equilibrium satisfying both T and F, how sensitive is this equilibrium to perturbations of the agents?
* Any complexity analysis regarding the computation of equilibrium or SVF?

**Limitations:**

Yes.

**Strengths And Weaknesses:**

**Strength**: The paper is technically sound and is significant to the community. It is the first work, to the best of the reviewer's knowledge, that establishes the theoretical connection between Bayesian incentives and group fairness. The theories are rigorously justified, and the experimentation is appropriate. The presentation is clear and organized, with the background and main goal of this work clearly delivered in the beginning.

**Weakness**:
* The paper assumes all data providers have the same prior belief of the system, including the model, mediator, and others' distribution. This might need justification for large-scale systems.
* The experiment correctly shows when truthfulness can dominate untruthfulness strategies. However, in regions where a truthful strategy is not dominant, it might be helpful to compare with other common DVF to determine whether an (empirical) improvement is possible.

---

> ### Author Rebuttal · Authors · 2026-03-31
>
> Thank you for the positive review and for appreciating the clarity, theoretical significance, and value of our work! We will address your questions below.
>
> > [W1] The paper assumes all data providers have the same prior belief of the system, including the model, mediator, and others' distribution. This might need justification for large-scale systems.
>
> Thank you for raising this concern.
> In a remark in Sec. 4, we mention that there are fewer data sources (e.g., hospitals holding larger datasets) in collaborative ML setting (Sim et al., 2020). Thus, we believe that agreement may still be possible among the few data providers, albeit with larger datasets each.
> We describe in App. A that in practice, before the collaboration begins, the data providers and mediator can discuss and agree on the model specifications and relationships based on historical, public, or similar data or models that are common knowledge or that they do not mind sharing.
> With more data providers, they can also use majority voting to select the best common prior belief.
> We note that the assumption of the same prior belief is also used by other Bayesian data valuation and collaborative machine learning methods (Chen et al., 2020; Sim et al., 2020).
>
> > [W2] The experiment correctly shows when truthfulness can dominate untruthfulness strategies. However, in regions where a truthful strategy is not dominant, it might be helpful to compare with other common DVF to determine whether an (empirical) improvement is possible.
>
> An empirical comparison with other common DVFs may be less helpful because:
> * A DVF is $i$-truthful if source $i$'s truthfulness leads to the highest value *in expectation*. This means that (a) for some realization of the datasets empirically, the truthful strategy may not lead to the highest value (i.e. be dominant) and (b) sources' knowledge of this theoretical guarantee is needed to incentivize their truthfulness.
> * Although other common DVF may lead to an empirical improvement, it (a) would not hold for all datasets as well and (b) may not incentivize each source $i$'s truthfulness as it is not theoretically guaranteed to maximize $i$'s reward on average.
>
> > [Q1] In an equilibrium satisfying both T and F, how sensitive is this equilibrium to perturbations of the agents?
>
> We are unsure what sensitivity and perturbations refer to or are with respect to, but we attempt to provide some clarifications below. Do let us know if further clarifications are needed.
> > The equilibrium in Sec. 4 would always satisfy (F) when fair semivalues are used. The equilibrium in Sec 4 would satisfy (T) whenever assumptions A1-3 hold, even when some agent $j$ truthfully has a different dataset $D_j$ or distribution $p(\mathcal{D}_j| \theta)$ that changes its data value and utilities.
> > Separately, in App. F.3. Fig. 14, we analyze the impact when source $1$ uses an untruthful strategy (add noise to dataset $D_1$). Source $1$ would have a lower value, and in this scenario, source $0$ would gain a higher value from deviating from its truthful strategy and duplicating its dataset to counteract the wrong $D_1$. Thus, when one source is significantly untruthful, other sources may benefit from deviating. However, as explained in Sec. 4 and Sec. 6.,  respectively, a source is unlikely to deviate unilaterally and truthfulness is still incentivized under minor model misspecifications.
>
> > [Q2] Any complexity analysis regarding the computation of equilibrium or SVF?
>
> We will add the following:
> > The computation of the equilibrium requires computing the semivalues for each of the $n$ sources.
> Thus, exact computation would requires $O(2^n)$ times the computation cost of the DVF and an efficient unbiased estimator requires $O(n log n)$ times the computation cost of the DVF as mentioned in Sec. 4 remark on efficiency.
>
> > The DVF can be computed exactly for exponential family models in time linear to the size of the dataset, e.g., $O(|D_i|)$ and by efficient Monte Carlo sampling for other models. The complexity of MC sampling is linear w.r.t the number of steps/samples and the cost of automatic differentiation which tend to be linear in the size of the dataset and number of model parameters.
>
> Thank you once again and we hope that our clarifications can address your concerns and improve your opinion of our work.

---

> > ### Author Rebuttal · Reviewer_LLXE · 2026-04-04
> >
> > Thanks the authors for the detailed rebuttal. The arguments are convincing to me. This paper with the clarifications meets the ICML criterion and I would like to keep the rating.

---

> > > ### Author Response · Authors · 2026-04-08
> > >
> > > Dear Reviewer LLXE,
> > >
> > > We are glad that our response addressed your concerns and you find it convincing. Your suggestions are invaluable to this work and we will include the additional explanations in our revision. Thank you for your recognition and we wish you all the best!
> > >
> > > Regards,
> > >
> > > The Authors.

---

### Decision · Program_Chairs · 2026-04-30

**Decision:**

Accept (spotlight)

**Comment:**

This paper studies the problem of incentivizing data providers in collaborative machine learning. The reviewers found the problem well motivated, theoretical results sound and the proposed mechanism intuitive.  All reviews are uniformly positive, some enhanced by the rebuttal process.